# Functional landscape of ubiquitin linkages couples K29-linked ubiquitylation to epigenome integrity

Javier Arroyo-Gomez [ID][1], Matthew J Murray [ID][1,2,7], Claire Guérillon [ID][1,2,7], Juanjuan Wang[1,7], Ekaterina Isaakova[3], Nazaret Reverón-Gómez [ID][1,2], Mikaela Koutrouli[1], Aldwin Suryo Rahmanto[3], Katrine Mitrofanov [ID][1], Andreas Ingham[1,2], Sofie Schovsbo[1], Katrine Weischenfeldt[1,2], Fabian Coscia [ID][1,4], Dimitris Typas[1], Moritz Völker-Albert[5], Victor Solis[5], Lars Juhl Jensen[1], Anja Groth[1,2,6], Andreas Mund [ID][1], Petra Beli[3], Robert F Shearer [ID][1✉] & Niels Mailand [ID][1,2,6✉]

## Abstract

**Linkage-specific ubiquitin chains govern the outcome of numerous critical ubiquitin-dependent signaling processes, but their targets and functional impacts remain incompletely understood due to a paucity of tools for their specific detection and manipulation. Here, we applied a cell-based ubiquitin replacement strategy enabling targeted conditional abrogation of each of the seven lysine-based ubiquitin linkages in human cells to profile system-wide impacts of disabling formation of individual chain types. This revealed proteins and processes regulated by each of these poly-ubiquitin topologies and indispensable roles of K48-, K63- and K27-linkages in cell proliferation. We show that K29-linked ubiquitylation is strongly associated with chromosome biology, and that the H3K9me3 methyltransferase SUV39H1 is a prominent cellular target of this modification. K29-linked ubiquitylation catalyzed by TRIP12 and reversed by TRABID constitutes the essential degradation signal for SUV39H1 and is primed and extended by Cullin-RING ubiquitin ligase activity. Preventing K29-linkage-dependent SUV39H1 turnover deregulates H3K9me3 homeostasis but not other histone modifications. Collectively, these data resources illuminate cellular functions of linkage-specific ubiquitin chains and establish a key role of K29-linked ubiquitylation in epigenome integrity.**

**Keywords** Ubiquitin; Linkage-specific ubiquitylation; Ubiquitin-dependent proteolysis; Epigenome maintenance; Heterochromatin
**Subject Categories** Chromatin, Transcription & Genomics; Post-translational Modifications & Proteolysis

## Introduction

Cellular homeostasis is maintained through a tightly regulated balance between protein synthesis and degradation, the latter of which is extensively mediated through the ubiquitin-proteasome system. Ubiquitin (Ub) is a highly abundant small modifier protein that is attached to substrate proteins through a hierarchical ubiquitylation cascade. Ub is activated in an ATP-dependent manner by an E1 Ub-activating enzyme and transferred to an E2 Ub-conjugating enzyme before it is covalently attached to an acceptor lysine on a substrate protein by an E3 Ub ligase (Swatek and Komander, 2016). Protein ubiquitylation is a reversible process that is antagonized by the action of deubiquitinases (DUBs), providing versatility to Ub signaling processes (Clague et al, 2019). Over 600 known E3 Ub ligase enzymes (Deshaies and Joazeiro, 2009) and ~100 DUBs (Clague et al, 2019) are encoded by human cells, conferring a high degree of specificity to the Ub system. Attachment of a single Ub moiety (mono-Ub) can elicit structural changes or modulate interaction surfaces of a substrate protein. However, the versatility of Ub signaling outcomes arises from chain extension via one of seven internal lysine residues (K6, K11, K27, K29, K33, K48, and K63), or the N-terminal methionine (M1), to form structurally distinct Ub polymers (poly-Ub). Chain assembly can also occur through multiple Ub-linkage sites simultaneously, giving rise to heterogeneous or branched chains, expanding Ub-chain complexity (Swatek and Komander, 2016). Furthermore, recent findings revealed that Ub can be attached to non-lysine amino acids and even to unconventional acceptor molecules such as lipids and sugars (Dikic and Schulman, 2023).

The most common outcome of Ub signaling is proteasomal degradation via the 26S proteasome, to which substrates are directed when modified with K48-linked Ub chains (Swatek and Komander, 2016). However, other Ub-linkages are associated with degradation in certain contexts. For example, K29-linked Ub chains are heavily upregulated during proteotoxic stress, colocalize with stress granule components, and enhance degradation signaling through facilitating p97/VCP-mediated unfolding (Yu et al, 2021b), which is primarily required to extract degradation substrates

[1]Novo Nordisk Foundation Center for Protein Research, Department of Cellular and Molecular Medicine, University of Copenhagen, 2200 Copenhagen, Denmark. [2]Danish Cancer Institute, Danish Cancer Society, 2100 Copenhagen, Denmark. [3]Institute of Molecular Biology, 55218 Mainz, Germany. [4]Max Delbrück Center for Molecular Medicine, Berlin, Germany. [5]EpiQMAx GmbH, Planegg, Germany. [6]Center for Epigenetic Cell Memory, Danish Cancer Society, 2100 Copenhagen, Denmark. [7]These authors contributed equally: Matthew J Murray, Claire Guérillon, Juanjuan Wang. ✉E-mail: robert.shearer@cpr.ku.dk; niels.mailand@cpr.ku.dk

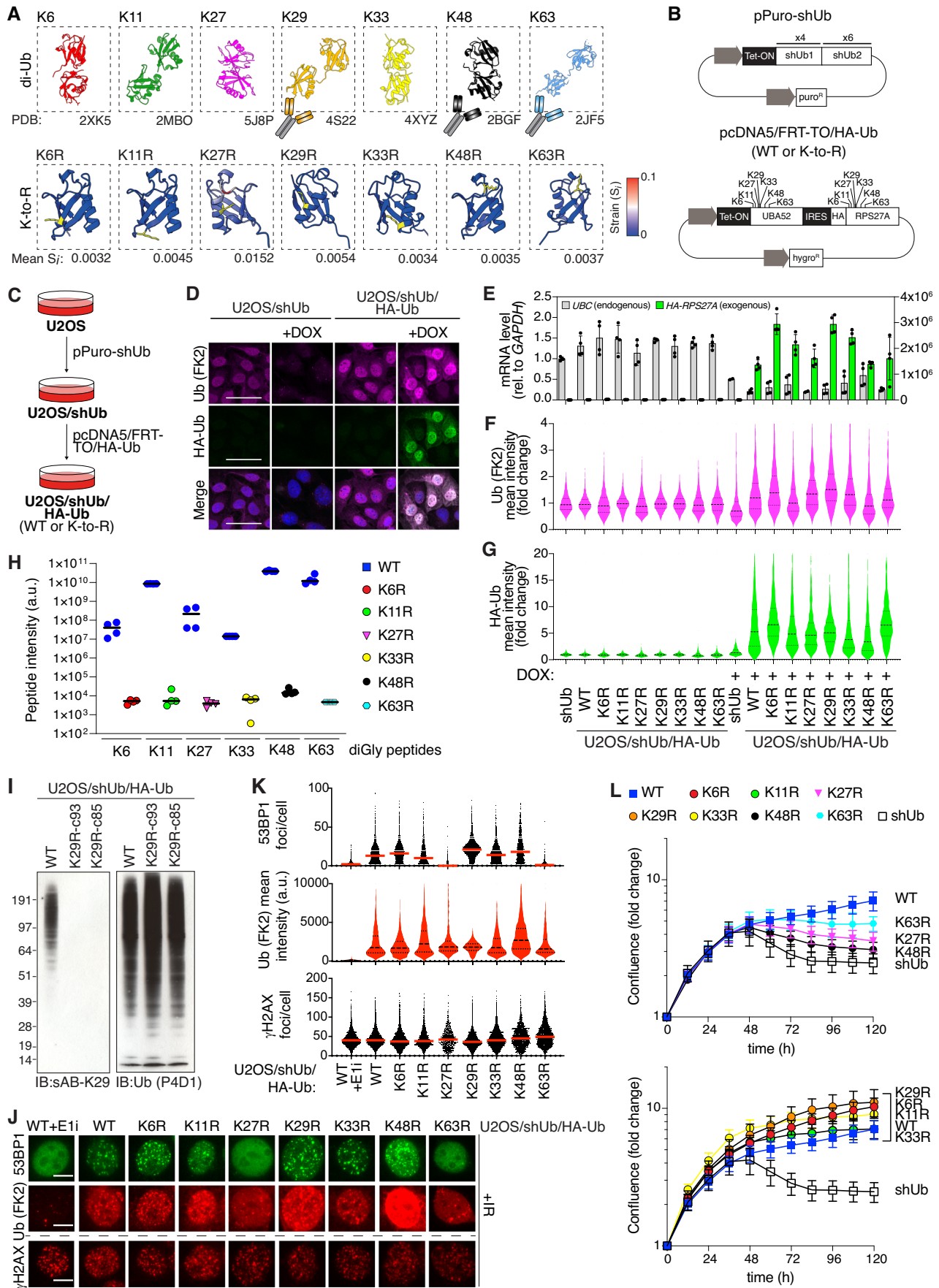

◄ **Figure 1. A Ub replacement cell line panel for inducible abrogation of lysine-based Ub topologies.**

(A) Schematic depicting structures of lysine linked di-Ub (upper panel). PDB identifiers are indicated. Summary of testing of commercially available Ub enrichment reagents with linkage specificity by immunoblot (IB) and immunofluorescence (IF) is indicated (see Fig. EV2). Alphafold2-based prediction (lower panel) of structural strain of Ub as a result of indicated K-to-R mutations (highlighted in yellow), expressed as Strain per residue ($S_i$). Strain on WT structure calculated from average of five output structures. (B) Schematic of doxycycline (DOX)-inducible pPuro-shUb and pcDNA5/FRT-TO/HA-Ub expression plasmids required for two-step Ub replacement (C). (C) Stepwise selection procedure for generation of DOX-inducible Ub replacement U2OS cell lines. (D) Representative images of U2OS/shUb and U2OS/shUb/HA-Ub(WT) derivative U2OS cell lines that were treated or not with DOX for 48 or 72 h, respectively, and immunostained with indicated antibodies. FK2 staining indicates conjugated Ub. Scale bar, 50 μm. (E) U2OS/shUb and derivative replacement cell lines were treated or not with DOX and mRNA levels were analyzed by RT-qPCR. Primers to *GAPDH* were used as a normalization control (data were technical duplicates of two independent experiments). Left axis corresponds to *UBC* detection and right axis corresponds to *HA-RPS27A* detection. (F, G). Quantitation of conjugated Ub (FK2) (F) and HA (G) immunostaining of U2OS/shUb and derivative replacement cell lines treated or not with DOX, determined by quantitative image-based cytometry (QIBC) (thick dashed line, median; dotted lines, quartiles). Data were a single representative replicate from three independent experiments, expressed as fold change normalized to untreated cells (>1000 cells analyzed per sample). (H) Intensity of indicated di-glycine (diGly) remnants derived from Ub peptides following diGly antibody-based enrichment from indicated Ub(WT) and Ub(K-to-R) replacement cell lines treated with DOX for 72 h (black bars, median; $n = 4$ technical replicates). See Dataset EV1 for full results. (I) Immunoblot analysis of U2OS/shUb/HA-Ub(WT) and independent derivative U2OS/shUb/HA-Ub(K29R) cell lines (c93 and c85) treated with DOX for 72 h. (J) Representative images of DOX-treated Ub replacement cell lines exposed to ionizing radiation (IR, 2 Gy) 1 h prior to CSK pre-extraction, fixation and immunostaining with indicated antibodies. Where indicated, the Ub E1 enzyme inhibitor MLN-7243 (E1i) was added 1 h before exposure to IR. Scale bar, 10 μm. (K) Quantitation of 53BP1 (upper), FK2 (middle) and γH2AX (lower panel) staining localized to IR-induced foci in cells in (J), using QIBC (thick dashed line, median; dotted lines, quartiles). Data were a single representative replicate from three independent experiments (>1500 cells analyzed per condition). (L) Short-term proliferation of U2OS/shUb and derivative Ub replacement cell lines after DOX induction for the indicated times. Cell lines that proliferate in a manner comparable to WT include K29R, K6R, K11R, and K33R replaced U2OS cells (lower), while K63R, K27R, K48R, and shUb-only replaced cell lines proliferated sub-optimally (upper panel) after DOX addition. Normalized logarithmic proliferation was determined using Incucyte image-based confluence assay (mean ± s.e.m.; $n = 5$ independent experiments). Source data are available online for this figure.

embedded in macromolecular structures or membranes (van den Boom and Meyer, 2018). More recently, K6-linked Ub chains driven by RNF14 were implicated in proteasome- and p97-dependent resolution of RNA-protein crosslinks (Suryo Rahmanto et al, 2023; Zhao et al, 2023). Ub chains can mediate receptor interactions rather than signal degradation by forming a binding surface for so-called Ub "readers". Although many Ub interactions are likely linkage-independent, it is expected that most Ub-linkage types have specific interaction partners, as a proteome-wide study probing with synthetic di-Ub found many linkage-specific interactions (Zhang et al, 2017). Ub-mediated interactions are conferred through mostly alpha-helical structures forming a diverse range of Ub-binding domains with various degrees of interaction affinity or Ub-linkage specificity (Randles and Walters, 2012), a classic example of which can be seen in DNA repair. Upon the formation of DNA double-strand breaks (DSBs), K63- and K27-linked Ub chains, catalyzed by RNF8 and RNF168, respectively, form at DSB sites, providing a scaffold for the recruitment of downstream DSB repair factors (Gatti et al, 2015; Schwertman et al, 2016).

The heterogeneous and dynamic nature of Ub signaling forms an important cellular code; however, not all Ub-linkage types are well characterized. Under normal cycling conditions, K6-, K27-, and K33-linked Ub chains are found in low abundance in mammalian cells (usually undetectable or <0.5%) (Dammer et al, 2011). The function of these atypical Ub-linkage types has been explored; however, due to technical limitations, broad biological functions have yet to be resolved. Furthermore, current understanding of low-abundance Ub-linkages often relies upon artifact-prone Ub mutant overexpression that may alter endogenous Ub signaling. Proteomic approaches allow quantitative measurement of Ub-linkage composition on individual substrates; however, methodologies for the high-throughput study of Ub-linkages in a complex mixture are lacking. This is largely due to the difficulty in developing robust enrichment reagents for multiple Ub-linkage types.

To address this limitation, we leveraged the Ub replacement strategy, a conditional cell-based system for depletion of the endogenous Ub pool that is rescued to near-endogenous levels by coexpression of exogenous Ub harboring specific K-to-R mutations

(Xu et al, 2009). Exemplifying its utility, we previously applied this Ub replacement approach to show that K27-linked Ub chains whose functions have been challenging to address are critical for the fitness of mammalian cells, predominantly nuclear and associated with p97 activity in the nucleus (Shearer et al, 2022). Here, expanding upon this approach, we profiled the global landscape of protein ubiquitylation changes induced by abrogating the formation of all individual lysine-based Ub-linkage types in human cells, providing unique system-wide insights into factors and processes impacted by specific Ub-linkages. Among numerous regulated proteins, this unveiled the histone methyltransferase SUV39H1 as a prominent substrate of proteolytic K29-linked ubiquitylation, which we show is tightly linked to chromosome-associated processes and indispensable for proteasomal degradation of SUV39H1 and other key chromatin components. We identify the enzymatic machinery responsible for K29-linked ubiquitylation of SUV39H1, revealing an intricate collaboration between TRIP12, the Ub ligase catalyzing K29-linked ubiquitylation of SUV39H1, and Cullin-RING E3 ligase (CRL) activity priming and extending these modifications. Stabilization of SUV39H1 by disrupting its regulation by K29-linked ubiquitylation deregulates the control of SUV39H1-dependent histone H3K9 trimethylation (H3K9me3) marks that govern heterochromatin formation (Lachner et al, 2001; Rea et al, 2000). Collectively, our findings provide an atlas of proteins and processes impacted by linkage-specific Ub modifications in human cells and highlight a key role for K29-linked ubiquitylation in preserving epigenome integrity.

## Results

### A Ub replacement cell line panel for conditional abrogation of all lysine-based Ub chains

We expanded upon the previously described Ub replacement strategy (Xu et al, 2009) to include all seven structurally diverse lysine-based Ub-linkages (Fig. 1A). In silico folding prediction of full-length Ub was used to

determine structural strain ($S_i$) resulting from individual lysine to arginine mutations (K-to-R), which was negligible (Fig. 1A). The $S_i$ for the Ub K27R mutation, although low, was relatively higher than other Ub K-to-R mutations (Fig. 1A), however we previously showed that the Ub K27R mutation is fully permissive for Ub conjugation in cells (Shearer et al, 2022). We next conducted a series of two-step stable transfections of a U2OS human osteosarcoma cell line to generate a panel of doxycycline-inducible Ub replacement cell lines for conditional abrogation of each individual linkage type (Fig. 1B,C). We had previously generated a U2OS cell line harboring a cassette encoding inducible shRNAs targeting the four human loci containing Ub-coding genes (herein U2OS/shUb) (Shearer et al, 2022). We utilized the U2OS/shUb base to generate a series of derivative cell lines expressing the human Ub fusion proteins UBA52 and RPS27A, with Ub either in wild-type (WT) or K-to-R configuration (herein U2OS/shUb/HA-Ub(WT) or U2OS/shUb/HA-Ub(K-to-R)) (Fig. 1B,C). Successful induction of Ub replacement was determined by detection of conjugated Ub after doxycycline treatment via immuno-fluorescence and immunoblot analysis. Stable clones were carefully selected for optimal Ub expression levels, both in uniformity and similarity to endogenous Ub levels, and doxycycline-induced Ub replacement in these cell lines was validated by qPCR analysis (Figs. 1D–G and EV1A–C). A characteristic Ub smear was visible by immunoblot for all Ub-replaced cell lines, indicating functional Ub-polymer formation (Fig. EV1A), and proteasomal degradation was blocked only in the U2OS/shUb/HA-Ub(K48R) replaced cell line when induced with a proteolysis targeting chimera (PROTAC) driving VHL-dependent degradation of BRD4 (Zengerle et al, 2015), indicating proteolytic function of the Ub K-to-R mutants was mostly unimpaired except for the expected impact of abrogating K48-linked Ub chain formation (Fig. EV1D). Importantly, quantitative diGly proteomics analysis verified strongly reduced abundance of the Ub-linkages whose formation is directly impacted by the Ub K-to-R mutations in the Ub(K6R), Ub(K11R), Ub(K27R), Ub(K33R), Ub(K48R), and Ub(K63R) replacement cell lines after 72 h of doxycycline treatment (48 h in the case of Ub(K48R) cells) (Fig. 1H; Dataset EV1). Although K29-linkages could not be consistently quantified by this proteomic approach due to technical limitations, we confirmed quantitative loss of K29-linked ubiquitylation in Ub(K29R) cell lines using a previously established highly specific synthetic antibody for K29-linkages (sAB-K29) (Yu et al, 2021b) (Fig. 1I).

To validate the utility of the Ub replacement cell line panel for probing Ub-linkage requirements of specific Ub-dependent processes, we monitored recruitment of the DNA repair factor 53BP1 to sites of ionizing radiation (IR)-induced DSBs driven by the Ub ligases RNF8 and RNF168 (Schwertman et al, 2016). Accumulation of 53BP1 at DNA damage sites was unimpaired except in Ub(K63R)- and Ub(K27R)-replaced cells or when Ub signaling was inhibited with an inhibitor of the Ub E1 enzyme (E1i) (Fig. 1J,K), consistent with the known requirement for both K63- and K27-linked Ub for efficient 53BP1 recruitment to DSBs (Gatti et al, 2015; Thorslund et al, 2015). Moreover, we used Ub(K6R) replacement cells and a K6-Ub-specific binder (LotA) to verify recent findings showing a strong increase in K6-linked ubiquitylation upon RNA-protein crosslink induction by combined UV-A and 4-thiouridine (S4U) treatment (Suryo Rahmanto et al, 2023) (Fig. EV1E,F).

We next utilized the complete Ub replacement panel to determine the relative importance of all individual lysine-based Ub topologies for the fitness of human cells. As expected, abrogation of high abundance Ub topologies (K48- and K63-linked Ub) resulted in poor short-term viability, as assessed by relative cellular confluence (Fig. 1L). Reduced cellular viability observed in Ub(K48R)- and Ub(K63R)-replaced cells was accompanied by an increased proportion of cells in G2/M phase and a reduced fraction of mitotic cells, indicative of G2 arrest (Fig. EV1G). Loss of several low-abundance Ub topologies (K6-, K11-, and K33-linked Ub) had minor impacts on cell proliferation in short-term confluence assays (Fig. 1L). As previously reported (Shearer et al, 2022), cells unable to form K27-linked Ub chains showed impaired viability, while abrogation of K29-linked ubiquitylation was well tolerated. Thus, K48-, K63-, and K27-linkages are instrumental for the proliferation of human cells, whereas K6-, K11-, K29-, and K33-linked Ub polymers are less critical for the overall fitness of unperturbed cells.

The relatively poor characterization of roles for low-abundance Ub chains is in part due to the limited range of robust linkage-specific Ub detection reagents. For this reason, the Ub replacement cell line panel was used to assess the performance and specificity of commercially available Ub-linkage-selective enrichment reagents for immunoblotting and immunofluorescence. A range of such affinity reagents displayed suboptimal linkage specificity in our hands when employed in a standard immunofluorescence protocol, except for the K29-linkage antibody sAB-K29 and the APU2 antibody recognizing K48-linked chains, which also specifically recognized these linkages by immunoblotting (Fig. EV2). An antibody recognizing K63-linked Ub chains performed well by immunoblot but showed nonspecific staining by immunofluorescence (Fig. EV2). Thus, the panel of conditional Ub(K-to-R) replacement cell lines provides a unique tool for interrogating the Ub-linkage requirements of Ub-mediated processes of interest and the performance of linkage-selective Ub affinity reagents in human cells.

## A proteome-wide map of ubiquitylation changes resulting from targeted disruption of individual Ub chain types

Given the limited functional understanding of low-abundance lysine-based Ub topologies in cell biology, we next used our panel of Ub replacement cell lines to profile global changes in the Ub-substrate landscape caused by loss of individual Ub chain types. To this end, we subjected HA-Ub conjugates isolated from Ub replacement cell lines under denaturing conditions and corresponding whole cell extracts following 72 h of Ub replacement to quantitative label-free MS proteomic analysis, and individual Ub(K-to-R) mutant samples were compared to samples from Ub(WT)-replaced cells (Fig. 2A). Among all Ub-replaced cell lines, the most pronounced impacts on overall proteome status were observed for abrogation of K48- and K63-linked ubiquitylation, consistent with these chains comprising the most abundant Ub-linkage types in cells, while disrupting other Ub chain types resulted in more modest global proteome changes (Appendix Fig. S1A; Dataset EV2). Due to the particularly strong inhibitory impact of Ub(K48R) replacement on cell proliferation (Fig. 1K), we examined whole proteome impacts of Ub(K48R)-replaced cell lines at both 48 and 72 h time points, the latter of which displayed more extensive proteomic changes (Appendix Fig. S1A). We then performed pairwise comparisons between HA-Ub conjugates isolated from Ub(WT)- and Ub(K-to-R)-replaced cells to comprehensively illuminate protein ubiquitylation changes resulting from abrogation of individual Ub-linkage types (Fig. 2B–H; Dataset EV3). Principal component analysis showed expected clustering of proteomic samples (Appendix Fig. S1B), and we used an immunoblotting approach to confirm selected major ubiquitylation

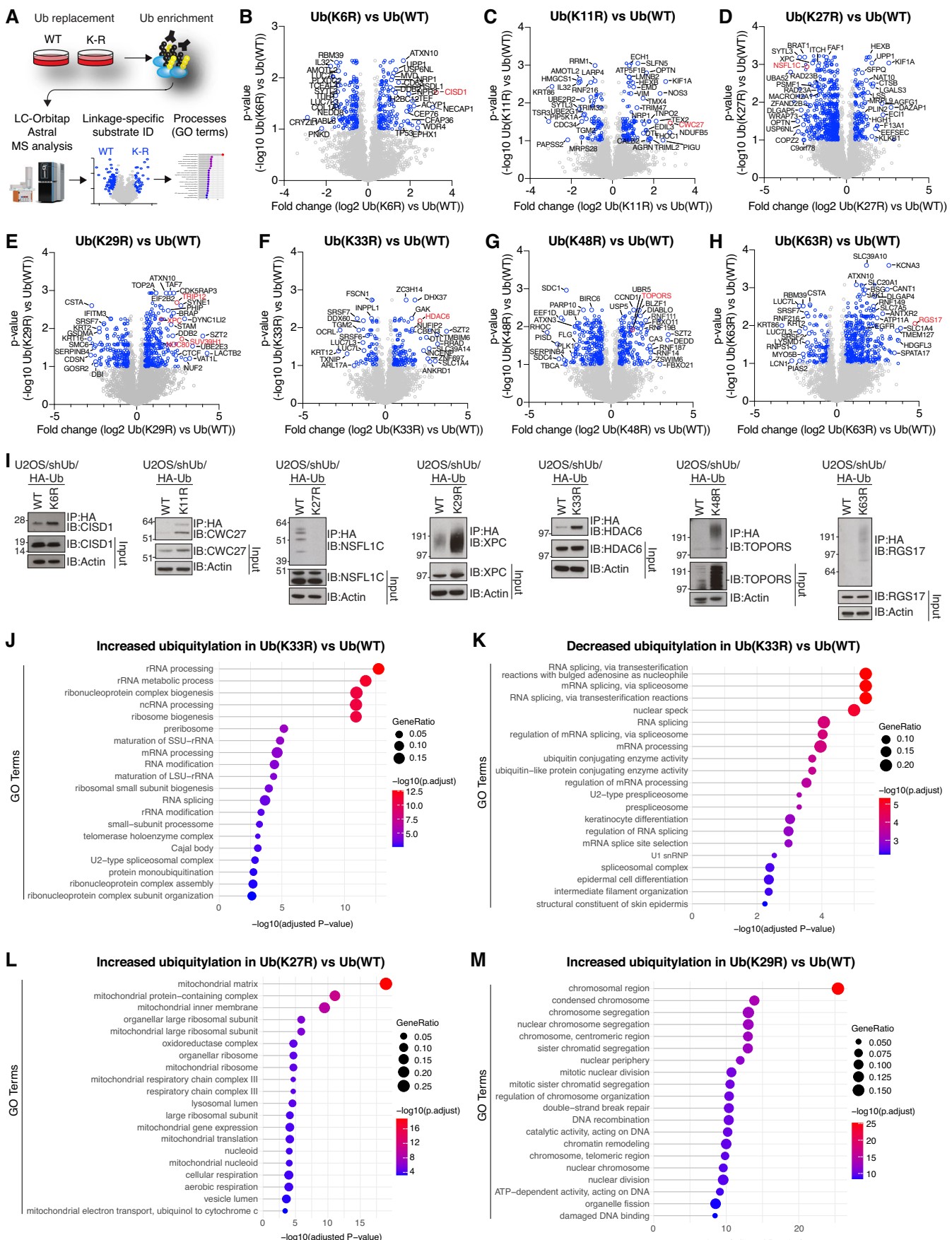

**Figure 2.   System-wide profiling of ubiquitylation changes induced by disrupting the formation of individual Ub chain types.**

(A) Schematic depicting experimental setup for proteomic analysis of HA-Ub conjugates enriched from whole cell lysates from Ub(WT) and Ub(K-to-R) replacement cell lines after DOX treatment for 48 or 72 h. (B–H) Ubiquitylation changes resulting from the replacement of endogenous Ub with indicated Ub(K-to-R) mutants by DOX treatment for 72 h, analyzed by quantitative label-free MS as shown in (A). Volcano plots depict protein abundance changes after denaturing HA-enrichment in HA-Ub(K-to-R)-replaced cells relative to HA-Ub(WT)-replaced cells. Blue dots indicate proteins with at least 1.5-fold enrichment and passing a significance threshold of $p < 0.1$ ($n = 3$ technical replicates; unpaired two-tailed $t$-test, Benjamini–Hochberg corrected). The most strongly regulated proteins are labeled. See Dataset EV2 for full results. (I) Immunoblot analysis validating individual hits from the MS datasets (B–H) through HA-Ub enrichment from indicated Ub(WT) and Ub(K-to-R) replacement cell lines after DOX treatment for 72 h. (J, K) GO term analysis of cellular compartments enriched among proteins showing significantly up- or downregulated ubiquitylation in Ub(K33R)-replaced cells relative to Ub(WT)-replaced cells (F) ($n = 3$ technical replicates; unpaired two-tailed $t$-test, Benjamini–Hochberg corrected). (L) GO term analysis of cellular compartments enriched among proteins showing significantly upregulated ubiquitylation in Ub(K27R)-replaced cells relative to Ub(WT)-replaced cells (D) ($n = 3$ technical replicates; unpaired two-tailed $t$-test, Benjamini–Hochberg corrected). (M) GO term analysis of cellular compartments enriched among proteins showing significantly upregulated ubiquitylation in Ub(K29R)-replaced cells relative to Ub(WT)-replaced cells (E) ($n = 3$ technical replicates; unpaired two-tailed $t$-test, Benjamini–Hochberg corrected). Source data are available online for this figure.

changes arising from disruption of each Ub-linkage type (Fig. 2I), supporting the validity of the proteomic data. Furthermore, the ubiquitylation profile of K27R-replaced cells recapitulated that of our previously described results (Shearer et al, 2022), including decreased overall ubiquitylation of multiple p97/VCP cofactors, including UFD1, NSFL1C/p47, and FAF1.

We used gene ontology (GO) term analysis applied to the ubiquitylation datasets to pinpoint cellular processes responsive to disruption of specific Ub-linkage topologies. For instance, Ub(K63R)-replaced cells uniquely showed significant enrichment of terms related to cell membranes (Fig. EV3), consistent with the well-established role of K63-linked ubiquitylation in membrane trafficking (Erpapazoglou et al, 2014). Strikingly, abrogation of K33-linked ubiquitylation, but not other Ub chain types, impacted the ubiquitylation status of proteins involved in multiple aspects of RNA processing, while disrupting K27-linked ubiquitylation was specifically associated with enhanced ubiquitylation of mitochondrial proteins (Figs. 2J–L and EV3). Thus, the ubiquitylation changes arising from abolishing the formation of specific Ub chain types shed light on the nature of cellular processes regulated by these Ub-linkage topologies. Interestingly, Ub(K29R)-replaced samples contrasted to other Ub(K-to-R) mutants in that enriching proteins were often nuclear localizing. Consistently, GO term analysis revealed a prominent enrichment of chromosome-associated processes that was selective to this Ub-linkage type (Figs. 2M and EV3). Among chromatin-associated proteins whose ubiquitylation status was markedly impacted by abrogating K29-Ub-linkage formation was SUV39H1, a histone methyltransferase that catalyzes the formation of histone H3 lysine 9 trimethylation (H3K9me3) enriched in repressive chromatin domains (Rea et al, 2000) and the kinetochore complex component NDC80, a central component of the NDC80 complex that forms at the contact point between kinetochores and microtubules (Ciferri et al, 2008) (Fig. 2E). Another notable protein enriched in Ub(K29R)-replaced samples was the E3 Ub ligase TRIP12, which catalyzes K29-linked Ub chain formation in vitro and targets several chromatin-associated proteins including RNF168 (Gudjonsson et al, 2012) (Fig. 2E). We therefore focused on these candidate K29-Ub-regulated proteins in subsequent studies of the function of K29-linked ubiquitylation in chromatin biology. Strikingly, when Ub conjugates enriched from Ub-replaced cells were immunoblotted using antibodies to TRIP12, SUV39H1, NDC80 and RNF168, all proteins displayed a high intensity molecular weight shift consistent with covalent poly-Ub modification exclusively in Ub(K29R)-replaced cells (Fig. 3A). This suggests that the turnover of Ub-modified forms of these proteins becomes defective when formation of K29-linkages is blocked. Indeed, SUV39H1, TRIP12, RNF168 and

NDC80 all displayed markedly elevated total abundance upon abrogation of K29-linked Ub chains, but not other linkage types (Fig. 3A). Collectively, these analyses provide a comprehensive picture of the impact of abrogating specific Ub-linkage types on global ubiquitylation dynamics in human cells, informing on proteins and cellular processes regulated by these Ub modifications.

## SUV39H1 is a substrate of proteasomal degradation via K29-linked ubiquitylation

SUV39H1 plays a central role in epigenome maintenance (Lachner et al, 2001; Rea et al, 2000). However, how histone methyltransferases and other epigenetic modifiers are regulated remains poorly understood. We therefore sought to functionally understand the emerging role of K29-linked Ub chains in regulating SUV39H1. Using Ub(WT)- and Ub(K29R)-replaced cells (Fig. EV4A), we first verified that the accumulation of Ub conjugates on SUV39H1 required Ub(K29R) replacement induction, and that the Ub(K29R) mutation had marginal impact on Ub stability and overall ubiquitylation/deubiquitylation dynamics, as evidenced by the similar kinetics of Ub conjugate clearance and accrual upon treatment with inhibitors of translation (Cycloheximide), the proteasome (MG132) and the Ub E1 enzyme (MLN-7243; E1i) (Fig. EV4B–E). Moreover, for validation purposes, an independent Ub(K29R) replacement clone (c85) was generated that showed Ub replacement kinetics and near-complete abrogation of K29-linked ubiquitylation comparable to the Ub(K29R)-c93 clone used above (Figs. 1I and EV4F,G). Additionally, both clones were confirmed to be undergoing correct transcription and translation using EU and L-AHA incorporation, respectively (Fig. EV4H). To confirm the topology of Ub chains formed on SUV39H1, a FLAG-SUV39H1 construct was transfected into Ub(K29R)-replaced cells to enable SUV39H1 isolation by FLAG immunoprecipitation under denaturing conditions. This showed that K29-linked Ub chains were readily detectable on FLAG-SUV39H1 by immunoblotting (Fig. 3B). Interestingly, K48-linked ubiquitylation of FLAG-SUV39H1 was notably increased when K29-linked ubiquitylation was prevented by Ub(K29R) replacement, in line with the strong increase in overall SUV39H1 ubiquitylation in Ub(K29R)-replaced cells (Fig. 3A,B). Similar to SUV39H1, ectopically expressed RNF168 was also modified by both K29- and K48-linkages (Fig. EV4I). Cycloheximide pulse-chase experiments to assay protein stability showed that endogenous SUV39H1 is a substrate of proteasomal degradation with a relatively short half-life in line with previous findings (Li et al, 2021) (Fig. 3C), revealing that SUV39H1 turnover is fully dependent on K29-linked

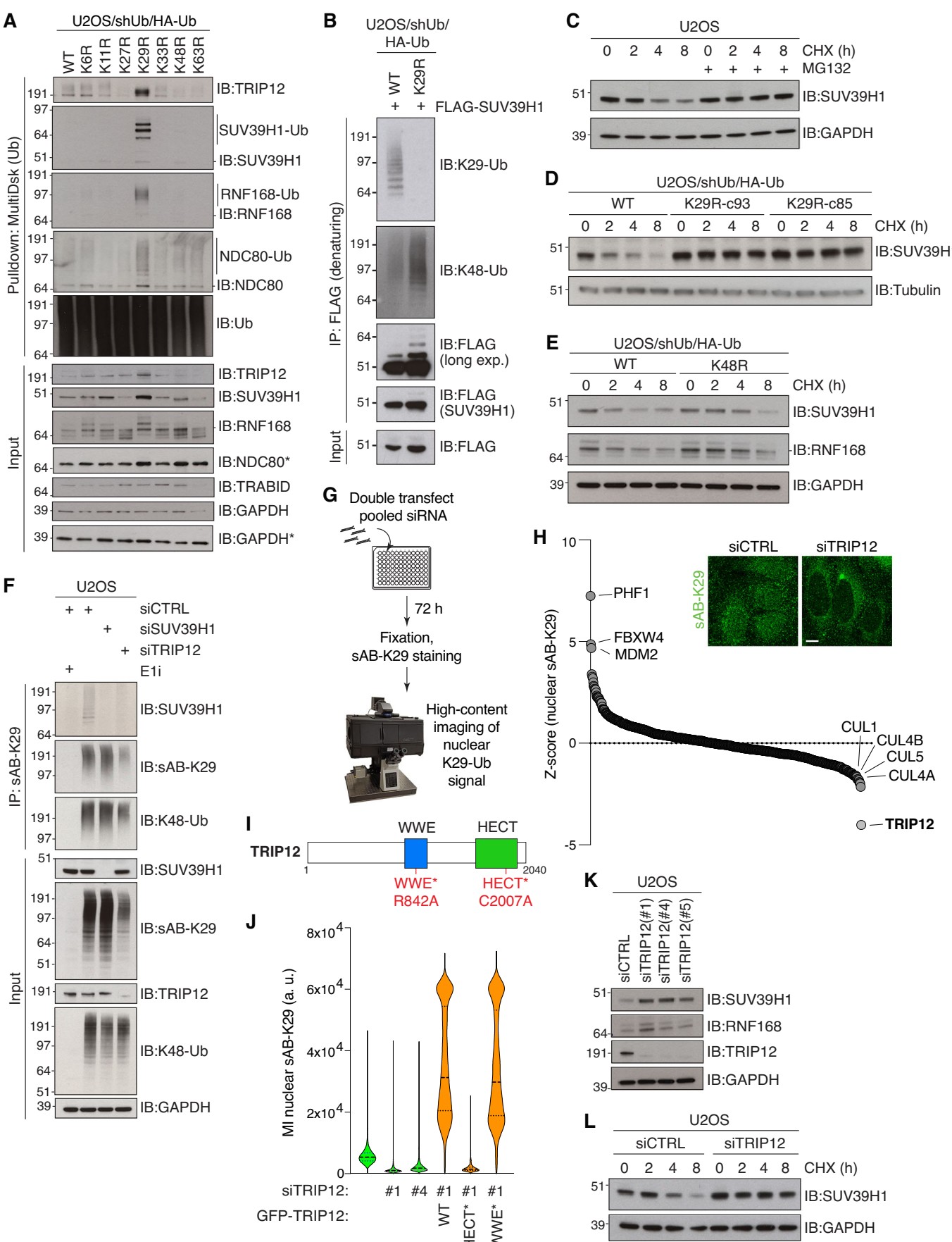

**Figure 3.   K29-linked ubiquitylation catalyzed by TRIP12 is essential for the degradation of the H3K9me3 methyltransferase SUV39H1.**

(A) Immunoblot analysis of Ub conjugates from indicated Ub-replaced cell lines isolated via MultiDsk pulldown (* indicates samples run on a separate gel). (B) DOX-treated Ub(WT) and Ub(K29R) replacement cell lines transfected with FLAG-SUV39H1 expression construct were subjected to FLAG IP under denaturing conditions and immunoblotted with the indicated antibodies. (C) Immunoblot analysis of SUV39H1 in U2OS cells treated or not with cycloheximide (CHX) for the indicated times in the absence or presence of the proteasome inhibitor MG132. (D, E) As in (C), using the indicated DOX-treated Ub replacement cell lines. (F) Immunoblot analysis of U2OS cells transfected with indicated siRNAs and treated or not with E1i were subjected to immunoprecipitation (IP) with sAB-K29 and immunoblotted with indicated antibodies. (G) Schematic depicting the workflow of the siRNA screen for regulators of K29-linked ubiquitylation in the nucleus. (H) Waterfall plot summarizing siRNA screen for regulators of nuclear K29-linked ubiquitylation in U2OS cells (G). The library contains multiplexed siRNAs to 575 E3 Ub ligase enzymes. Data shows the Z-score of sAB-K29 immunostaining relative to the population by QIBC. See Dataset EV3 for full results. Inset shows representative images of cells transfected with indicated siRNAs and immunostained with sAB-K29. Scale bar, 10 μm. (I) Schematic of human TRIP12 protein showing functional domains and mutations introduced to generate the indicated mutants. (J) QIBC analysis of nuclear sAB-K29 immunostaining in U2OS cells sequentially transfected with indicated siRNA and siRNA-resistant GFP-TRIP12 expression plasmids. Samples transfected with GFP-TRIP12 plasmids (orange) were gated on GFP-positive cells. Data were a single representative replicate from three independent experiments (thick dashed lines, median; dotted lines, quartiles; >1000 cells analyzed per condition). (K) Immunoblot analysis of U2OS cells transfected with non-targeting control (CTRL) or TRIP12 siRNAs. (L) Immunoblot analysis of siRNA-transfected U2OS cells treated or not with cycloheximide (CHX) for the indicated times. Source data are available online for this figure.

ubiquitylation as evidenced by its complete stabilization in two independent clones of Ub(K29R)-replaced cells (Figs. 3D and EV4J). Similar K29-Ub-dependent degradation kinetics were observed for RNF168 (Fig. EV4K). By contrast, abrogation of K48-linked ubiquitylation in Ub(K48R)-replaced cells only partially stabilized SUV39H1 and RNF168 (Fig. 3E), suggesting that K29-linked Ub chains represent an essential degradation signal for these proteins, whereas K48-linkages are less critical. The presence of K29-linked Ub chains on endogenous SUV39H1 was confirmed by immunoblotting of K29-linked Ub immunoprecipitates (Fig. 3F). Collectively, these results highlight an important role for K29-linked ubiquitylation in Ub-dependent regulation of chromatin-associated proteins and establish SUV39H1 as a novel substrate of proteasomal degradation via K29-linked Ub chains.

## TRIP12 is required for K29-linked ubiquitylation of SUV39H1

Due to the enrichment of chromosome-associated proteins affected by K29-linked ubiquitylation, we next sought to explore Ub writer machineries involved in nuclear K29-linked ubiquitylation in an unbiased manner. To this end, we performed a high-content imaging-based siRNA screen for E3 Ub ligases impacting K29-Ub abundance using sAB-K29 binder positivity in the nucleus as a readout. The specificity of sAB-K29 in detecting K29-Ub signals by immunostaining was confirmed in two independent Ub(K29R) clones (Fig. EV4F). U2OS cells were transfected with an siRNA library comprehensively covering known E3 Ub ligase enzymes and immunostained with sAB-K29 (Fig. 3G; Dataset EV4). Notably, depletion of TRIP12 resulted in the strongest decrease in nuclear K29-Ub detection (Fig. 3H), with an approximate 50% decrease in nuclear K29-Ub signal relative to a non-targeting siRNA control (Dataset EV4) and a marked reduction in overall cellular K29-linked ubiquitylation (Fig. 3F). Interestingly, a moderate decrease in detectable K29-Ub signal was also observed when depleting several Cullin proteins (CUL1, CUL4A, CUL4B, or CUL5), scaffold proteins for Cullin-RING-type Ub ligases (CRLs) (Fig. 3H). Reduced detection of nuclear K29-Ub and stabilization of SUV39H1 and RNF168 after TRIP12 depletion was also observed in HeLa cells (Fig. EV5A,B). TRIP12 is an E3 Ub ligase containing a catalytic HECT domain required for Ub ligase activity and a WWE domain recognizing poly(ADP-ribosyl)ation (PARylation) modifications required for PAR-targeted Ub-ligase (PTUbL) activity

(Gatti et al, 2020). To validate the involvement of TRIP12 in generating nuclear K29-Ub chains, we obtained TRIP12 expression constructs encoding WT, catalytic-dead (C2007A, herein HECT*) and PAR-binding deficient (R842A, herein WWE*) mutants (Fig. 3I), and introduced silent coding mutations providing resistance to an siRNA targeting TRIP12 (Fig. EV5C). Expression of siRNA-resistant GFP-TRIP12 restored nuclear K29-linked ubiquitylation in cells depleted of endogenous TRIP12 in a manner that was dependent on E3 ligase activity, as expected, but independent of PAR-binding (Figs. 3J and EV5D). Further suggesting a preference for TRIP12 in catalyzing K29-linked Ub chains, our MS data showed a clear accumulation of ubiquitylated TRIP12 along with SUV39H1, RNF168 and other proteins after Ub(K29R) replacement (Fig. 2E). Many E3 Ub ligase enzymes have been suggested to self-regulate by auto-ubiquitylation either in cis or trans via oligomerization (de Bie and Ciechanover, 2011). Consistently, TRIP12 accumulates in Ub(K29R)-replaced cells and displays a mobility shift similar to other K29-Ub substrates (Figs. 3A and EV5E), suggesting that TRIP12 auto-ubiquitylation via K29-Ub chains promotes its proteasomal turnover.

Given the strong evidence for TRIP12 as a major E3 ligase responsible for nuclear K29-linked ubiquitylation, we next asked whether TRIP12 is required for proteolytic turnover of SUV39H1. Supporting this, depletion of endogenous TRIP12 in U2OS cells resulted in clear stabilization of SUV39H1 similar to the impact of Ub(K29R) replacement (Fig. 3D,K,L), and depletion of TRIP12 removed the K29-Ub chains detected directly on SUV39H1 (Fig. 3F). Consistent with our data showing dispensable PAR-binding activity of TRIP12 for nuclear K29-Ub formation, PARylation was not required for the turnover of SUV39H1, which was degraded normally in the presence of the PARP inhibitor Olaparib (Fig. EV5F). We conclude that TRIP12 is the dominant writer of nuclear K29-linked Ub in human cells, responsible for K29-linkage-dependent ubiquitylation and degradation of SUV39H1.

## TRABID antagonizes K29-linked Ub chains on SUV39H1 and other nuclear substrates

Ub chains are tightly regulated through the action of DUBs (Clague et al, 2019). Having established TRIP12 as the major writer of nuclear K29-linked ubiquitylation, we next investigated whether the proteostasis of nuclear K29-Ub substrates is regulated by DUB activity. TRABID (ZRANB1) is a nuclear DUB antagonizing K29- and K33-linked Ub chains (Michel et al, 2015), and as such, we obtained a TRABID

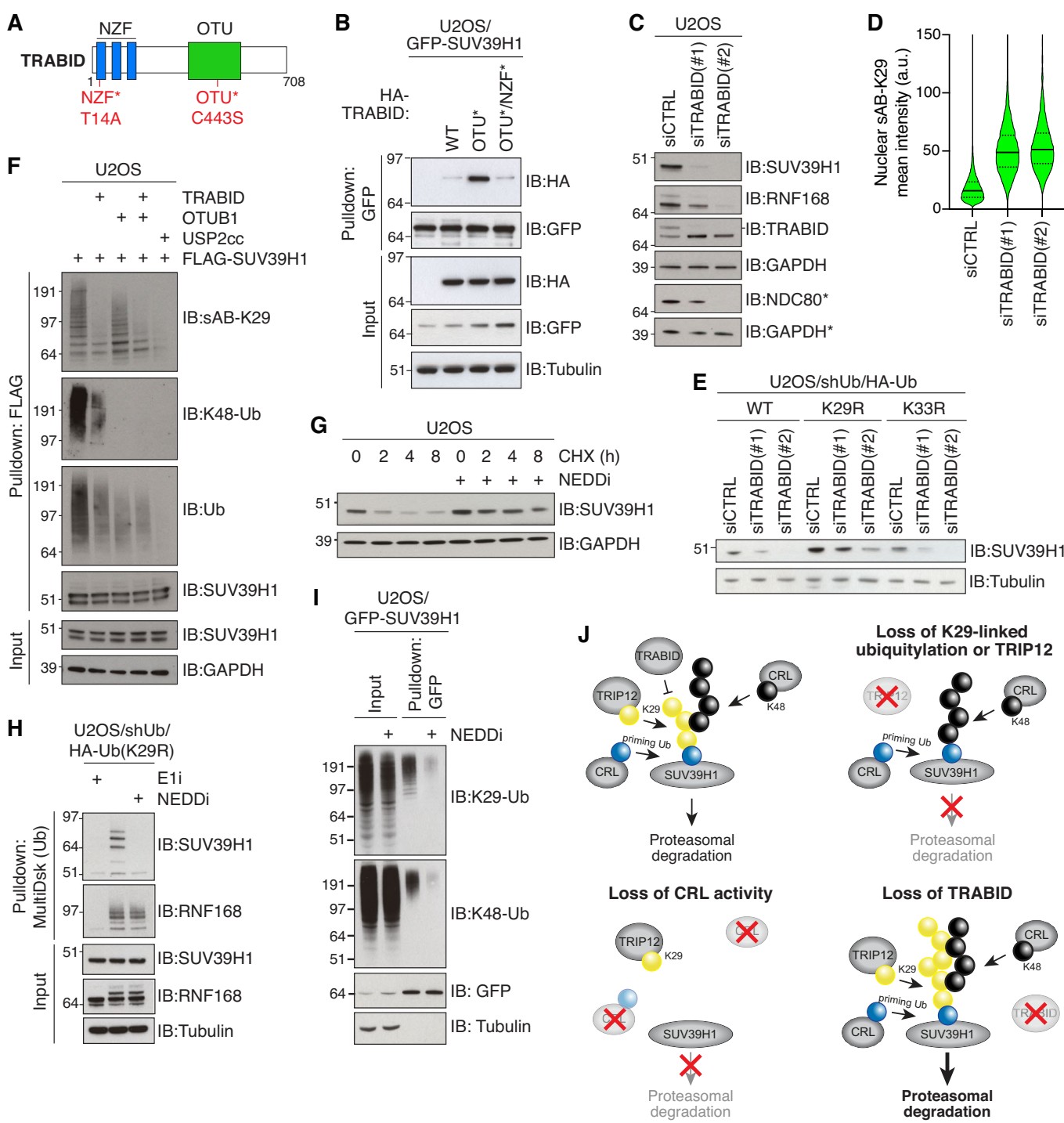

expression vector for binding studies. We introduced mutations (Fig. 4A) to generate a catalytic-dead (C443S, herein OTU*) variant of TRABID, which has been described as a trapping mutant that binds Ub-modified substrates with increased affinity (Harris et al, 2021). We further generated a Ub-binding-deficient variant of TRABID OTU* by introducing a T14A mutation into the NZF domain (T14A/C443S, herein NZF*) that has been shown to abolish Ub-binding of TRABID (Kristariyanto et al, 2015). Immunoprecipitation analysis revealed a weak interaction between TRABID WT and SUV39H1, which was

greatly enhanced by the inactivating OTU* mutation of TRABID and reduced back to WT levels by simultaneous mutation of the NZF (Fig. 4B), suggesting that TRABID recognizes and targets Ub modifications on SUV39H1. TRABID overexpression was accompanied by a marked increase in SUV39H1 abundance (Fig. EV6A). Conversely, depletion of TRABID resulted in destabilization of endogenous SUV39H1, RNF168 and NDC80 in U2OS cells (Fig. 4C), likely due to the acceleration of their K29-linked ubiquitylation in the absence of TRABID. Furthermore, the strong effect of TRABID depletion in reducing SUV39H1 abundance

**Figure 4. K29-linked ubiquitylation of SUV39H1 is reversed by TRABID and primed by CRL activity.**

(A) Schematic of human TRABID protein showing functional domains and mutations introduced to generate indicated mutants. (B) U2OS cells stably expressing GFP-SUV39H1 were transfected with indicated HA-TRABID expression constructs and subjected to GFP pulldown, followed by immunoblotting with indicated antibodies. (C) Immunoblot analysis of U2OS cells transfected with indicated siRNAs. Note that siTRABID(#1) has reduced knockdown efficiency relative to siTRABID(#2) (* indicates samples run on a separate gel). (D) QIBC analysis of nuclear sAB-K29 immunostaining in U2OS cells treated with indicated siRNAs. Data were a single representative replicate from three independent experiments (red bar, median; dotted lines, quartiles; >5000 cells analyzed per condition). (E) Immunoblot analysis of DOX-treated Ub(WT), Ub(K29R), and Ub(K33R) cells transfected with indicated siRNAs. (F) Immunoblot analysis of FLAG-SUV39H1 IPs isolated via FLAG IP and incubated with indicated DUBs for 1 h at 37 °C. (G) Immunoblot analysis of SUV39H1 in U2OS cells treated or not with cycloheximide (CHX) for the indicated times in the absence or presence of a NEDDylation inhibitor (NEDDi). (H) Immunoblot analysis of Ub conjugates isolated via MultiDsk pulldown from Ub(K29R)-replaced cell lines treated with Ub E1 inhibitor or NEDDylation inhibitor. (I) U2OS/GFP-SUV39H1 cells treated or not with NEDDi for 4 h were subjected to GFP pulldown under denaturing conditions and immunoblotted with indicated antibodies. (J) Schematic model depicting regulation of linkage-specific SUV39H1 ubiquitylation and stability by TRIP12, TRABID, and CRL activity. Source data are available online for this figure.

was replicated in other mammalian cell lines (Fig. EV6B). In line with this, TRABID-depleted cells showed a robust increase in nuclear K29-Ub chains when immunostained using sAB-K29 (Figs. 4D and EV6C), suggesting that TRABID is a major DUB responsible for hydrolyzing K29-linkages in the nucleus. Destabilization of SUV39H1 in TRABID-depleted cells was rescued by Ub(K29R) but not Ub(K33R) replacement (Fig. 4E), indicating that TRABID activity towards SUV39H1 is K29-Ub-dependent. Consistently, recombinant TRABID efficiently removed K29-linked Ub conjugates on SUV39H1 in vitro (Fig. 4F). Together, these results show that SUV39H1 proteolytic turnover via TRIP12-mediated K29-linked ubiquitylation is antagonized by the DUB activity of TRABID.

## TRIP12 and CRLs cooperate to promote SUV39H1 ubiquitylation and degradation

We next explored the role of CRLs in regulating SUV39H1 turnover, given that depletion of Cullin proteins also reduced nuclear K29-Ub abundance albeit to a lesser extent than TRIP12 (Fig. 3H). The appearance of multiple Cullins (CUL1, CUL4A, CUL4B, and CUL5) in our siRNA screen led us to examine the role of Cullins as a redundant grouping. To this end, we utilized a potent inhibitor of Neddylation (herein NEDDi) that selectively inhibits the NEDD8-activating enzyme E1 regulatory subunit (NAE1) (Soucy et al, 2009), a core component of the UBA3-NAE1 dimer required for CRL activation by covalent attachment of the ubiquitin-like modifier NEDD8 (Saha and Deshaies, 2008). Strikingly, treatment with NEDDi impaired degradation of SUV39H1 in the presence of cycloheximide, implying a requirement for CRL activity for SUV39H1 degradation (Fig. 4G). The observation that high-molecular-weight Ub smears accumulated on SUV39H1 in Ub(K29R)-replaced or TRIP12-depleted cells (Figs. 3A and EV6D,E) suggested that these non-K29-linked Ub modifications might be CRL-dependent. Indeed, NEDDi treatment completely abolished Ub modifications on SUV39H1 in Ub(K29R)-replaced cells in a manner comparable to treatment with the Ub E1 inhibitor (Figs. 4H and EV6F). Moreover, NEDDi treatment diminished both K48- and K29-linked ubiquitylation of SUV39H1 (Fig. 4I). Interestingly, in vitro deubiquitylation experiments using TRABID and the K48-linkage-specific DUB OTUB1 revealed that hydrolysis of K29-linked Ub chains from SUV39H1 also removes the bulk of K48-Ub modifications on SUV39H1 but not vice versa (Fig. 4F), suggesting that K48-linked ubiquitylation of SUV39H1 mostly occurs distal to K29-linked ubiquitylation. Consistently, K48-linked Ub chains on SUV39H1 were present in higher

molecular weight smears than the corresponding K29-linkages (Fig. 4F). However, the observation that CRL activity is required for K29-linked ubiquitylation of SUV39H1 (Fig. 4I) suggests that CRL complexes not only extend K29-Ub conjugates on SUV39H1 with non-K29-linkages but are also required for initiating ubiquitylation of SUV39H1 to prime its subsequent TRIP12-dependent K29-Ub modification. Interestingly, NEDDi treatment had no effect on Ub chains accumulating on RNF168 in Ub(K29R)-replaced cells (Fig. 4H), in agreement with the known role of the NEDDi-insensitive HECT E3 ligase UBR5 in regulating RNF168 stability (Gudjonsson et al, 2012; Kaiho-Soma et al, 2021). Collectively, these findings suggest an interconnected role of TRIP12 and other Ub ligases whose identities may differ between substrates in promoting proteasomal degradation of SUV39H1 and other chromatin-associated targets, with the latter E3s priming and branching TRIP12-dependent K29-linked ubiquitylation that provides the critical degradation signal (Fig. 4J).

## K29-Ub-dependent regulation of SUV39H1 is required for H3K9me3 homeostasis

Organization of mammalian chromatin into transcriptionally repressed heterochromatin is controlled by the addition of epigenetic modifications of DNA and histones. Constitutive heterochromatin is modified by H3K9me3 (Allshire and Madhani, 2018), with SUV39H1 acting as the major histone methyltransferase catalyzing H3K9me3 formation in these chromatin domains (Rea et al, 2000). Given the critical role of nuclear K29-Ub in regulating SUV39H1 stability, we therefore explored whether and how altered SUV39H1 stability affects the epigenetic landscape in human cells. To this end, we performed unbiased MS-based profiling of more than 80 histone post-translational modifications (PTMs) in Ub(WT) and Ub(K29R) replacement cell lines treated or not with doxycycline. Consistent with increased SUV39H1 activity, K29R-Ub replacement corresponded with a marked increase in global H3K9me3 levels (Fig. 5A). Strikingly, however, other changes in the complex PTM landscape on core histones were negligible, apart from a minor decrease in H3K9me1 detection that most likely reflects the corresponding increase in H3K9me3 (Fig. 5A; Dataset EV5). We confirmed that Ub(K29R)-replaced U2OS cells do not show increased stabilization of other known H3K9me3 writers and erasers (Fig. 5B), arguing that the increased level of H3K9me3 is largely due to the stabilization of SUV39H1. Consistently, the excess SUV39H1 pool in Ub(K29R)-replaced cells exclusively accumulated on chromatin, and the marked increase in global H3K9me3 abundance induced by Ub(K29R) replacement was curtailed by SUV39H1 knockdown (Figs. 5C and EV7A). Moreover, quantitative proteomic analysis

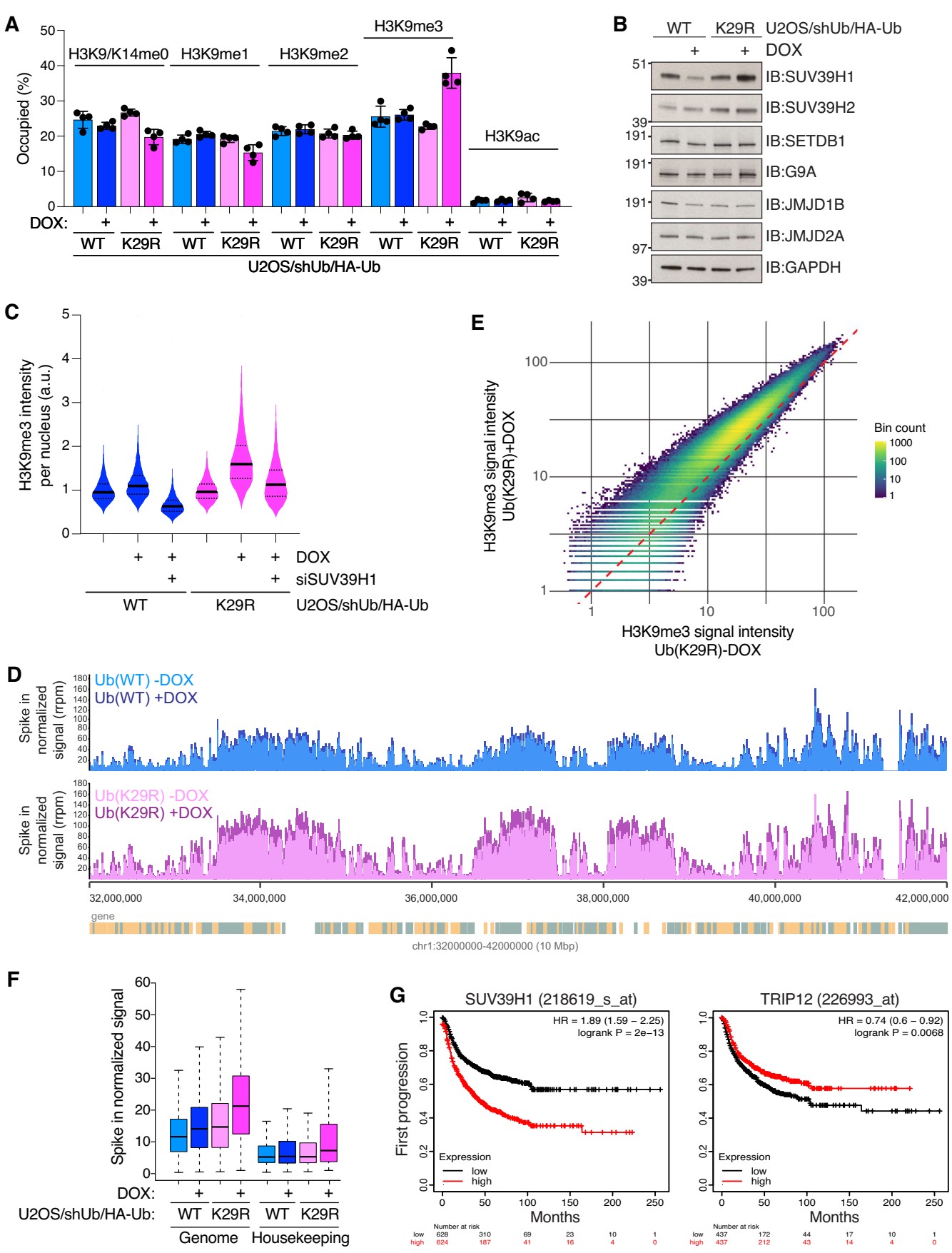

◀ **Figure 5. Abrogation of K29-linked ubiquitylation deregulates H3K9me3 homeostasis.**

(A) Quantitative proteomic analysis of H3K9 modifications in Ub(WT) and Ub(K29R) replacement cell lines treated or not with DOX ($n = 4$ biological replicates). (B) Immunoblot analysis of Ub(WT) and Ub(K29R) replacement cell lines treated or not with DOX, using indicated antibodies. (C) QIBC analysis of nuclear H3K9me3 immunostaining in Ub(WT) and Ub(K29R) replacement cell lines treated or not with DOX and transfected with SUV39H1 siRNA where indicated. Data were a single representative replicate from three independent experiments (black bar, median; dotted lines, quartiles; >5000 cells analyzed per condition). (D) Snapshot of chromosome 1 depicting H3K9me3 ChIP-seq signal distribution in Ub(WT) and Ub(K29R) replacement cell lines treated or not with DOX. (E) Scatter plot comparing H3K9me3 ChIP-seq signal intensities between Ub(K29R) replacement cells induced or not with DOX. Each point represents a 10 kb genomic region common to both datasets. Signal values are spike-in normalized. (F) ChIP-Seq quantification of H3K9me3 in Ub(WT) and Ub(K29R) replacement cell lines treated or not with DOX stratified by genomic location. Reads per million normalized to *Drosophila* spike-in (center line, median; box limits, first and third quartiles; whiskers, lowest, and highest data points within 1,5x the interquartile range from the box; $n = 4$ biological replicates). (G) Kaplan–Meier analysis of first progression-free survival in non-small cell lung cancer (NSCLC) stratified on SUV39H1 (left) or TRIP12 (right) expression. Analysis was performed using KMPlotter tool. Hazard ratios and $p$ values were calculated through Cox regression. Source data are available online for this figure.

revealed a significant increase in H3K9me3 abundance upon TRIP12 knockdown that was rescued by simultaneous depletion of SUV39H1 (Fig. EV7B), further suggesting that loss of TRIP12- and K29-Ub-dependent SUV39H1 turnover undermines regulatory control of H3K9me3 formation. We next subjected Ub(WT) and Ub(K29R) replacement cell lines treated or not with doxycycline to quantitative ChIP-seq analysis of H3K9me3, in order to profile genome-wide alterations in the H3K9me3 landscape in the absence of K29-linked Ub. Consistent with our proteomic data (Fig. 5A), the total H3K9me3 ChIP-seq signal was increased in Ub(K29R)-replaced cells (Fig. 5C–E). Notably, the increase in H3K9me3 levels upon abrogation of K29-linked ubiquitylation was highly uniform across the genome (Figs. 5C–E and EV7C). However, H3K9me3 domains remained positionally stable in Ub(K29R)-replaced cells and did not expand substantially to neighboring coding regions, despite the global increase in H3K9me3 abundance (Figs. 5D–F and EV7C,D). These findings reveal an important role of K29-linked ubiquitylation in maintaining H3K9me3 homeostasis by restricting SUV39H1 levels and activity in heterochromatin.

## TRIP12 and SUV39H1 expression levels anticorrelate in cancer

Heterochromatin establishment and maintenance is central to cell identity. Aberrant H3K9me3 occupancy is associated with loss of cell identity, such as premature aging or cancer (Padeken et al, 2022). As such, we next explored whether TRIP12-dependent regulation of SUV39H1 is disrupted in a disease context. Kaplan–Meier analysis of survival across a dataset of 33 types of cancer (GEPIA2 (Tang et al, 2019)) showed that higher expression of SUV39H1 correlates with poorer disease-free survival (Fig. EV7E). Conversely, the same cohort showed the opposite relationship for TRIP12, with poorer disease-free survival correlating with lower TRIP12 expression (Fig. EV7E). Although higher expression at the mRNA level is likely independent of Ub-mediated degradation, poorer disease-free survival with increased SUV39H1 may indicate a generally adverse effect of high SUV39H1 in cancer progression, highlighting the importance of regulating SUV39H1 abundance and activity.

We next explored the relationship between TRIP12 and SUV39H1 protein levels in various cancer types. Lung adenocarcinoma (LUAD) is one of the most common subtypes of non-small cell lung cancer (NSCLC), which makes up 85% of cases worldwide (Herbst et al, 2018). Stratification of LUAD samples on TRIP12 protein detection showed lower SUV39H1 protein detection in "high" TRIP12-expressing samples (Fig. EV7F), a relationship not

seen when samples were stratified on TRIP12 mRNA levels (Fig. EV7G). Consistently, Kaplan–Meier analysis of survival in a cohort of NSCLC patients (Gyorffy, 2024) showed a favorable outcome for patients with either low SUV39H1 or high TRIP12 expression in both overall survival and first progression, emphasizing the anticorrelation of these proteins in survival (Figs. 5G and EV7H).

# Discussion

Low-abundance Ub chains are challenging to isolate for study due to a paucity in high-performance linkage-specific Ub detection reagents. To remedy this barrier, we leveraged a robust cell-based strategy for the conditional abrogation of all seven individual lysine-based Ub-chain topologies in human cells, expanding our previous work identifying a role for K27-linked Ub chains in p97-dependent nuclear proteostasis (Shearer et al, 2022), and carefully benchmarked performance using biochemical and cell biological validation. The Ub replacement cell line panel described here complements an array of tools to study the Ub-proteasome system at multiple levels, such as absolute quantitation of Ub linkage abundance (Ub-AQUA) (Kirkpatrick et al, 2006) and Ub chain restriction (UbiCRest) analysis (Hospenthal et al, 2015) for the respective analysis of Ub chain composition and architecture, the DegronID database (Zhang et al, 2023) for mapping of E3 Ub ligase specificity, Ub interactor Affinity enrichment-MS (UbiA-MS) (Zhang et al, 2017) for the profiling of Ub chain interacting proteins, and a proteome half-life resource (Li et al, 2021). Our Ub replacement cell lines were rigorously selected to ensure uniform conditional expression of exogenous Ub at levels similar to that of endogenous Ub and thus maximal compliance with endogenous ubiquitylation dynamics and cell fitness upon Ub replacement. This cell line panel represents a readily available and highly accessible system for the investigation of low-abundance Ub chains and their cellular functions in the absence of appropriate enrichment reagents, which often display cross-reactivity with more abundant Ub-linkages, or suboptimal performance in vivo, as we show for a range of commercially available linkage-specific Ub affinity reagents. The system is maintained in standard cell culture conditions, requiring minimal specialized equipment or expertise to implement.

A key advantage of the Ub replacement strategy is the retention of cellular context, both in terms of abrogating a specified Ub-linkage in live cells and in keeping other Ub-substrate linkages intact. Compatibility of the Ub replacement system with live-cell

assays allowed us to explore, for the first time, the requirement of all Ub-linkages for viability of human cells, showing essentiality of K48-, K63-, and K27-linkages, but not K6-, K11-, K29-, and K33-linked Ub, for short-term proliferation of non-stressed U2OS cells. However, it is likely that a stronger dependency on these nonessential Ub-linkage-types would be observed under appropriate perturbation conditions. K6-linked ubiquitylation, for example, was recently shown to promote the resolution of RNA-protein crosslinks (RPCs) (Suryo Rahmanto et al, 2023; Zhao et al, 2023). Indeed, using Ub(K6R) replacement cells, we validated a strong RPC-dependent increase in K6-linked ubiquitylation. In general, our panel of Ub replacement cell lines enables rapid and efficient probing of the requirement of specific Ub-linkages for cellular processes of interest. Illustrating this, we show that K63- and K27-linked Ub chains, but not other Ub-linkages, are required for Ub-dependent recruitment of 53BP1 to DNA damage sites, in line with previous findings (Gatti et al, 2015; Thorslund et al, 2015). However, some care must be taken when interpreting data obtained with Ub mutants containing K-R substitutions due to potential impacts on their recognition and processing by Ub system components, and where possible, results should be validated by orthogonal approaches. In the present study, we show that the Ub(K29R) mutation has no discernible impact on Ub stability and overall cellular ubiquitylation/deubiquitylation kinetics, and we recapitulate the key role of K29-linked ubiquitylation in promoting SUV39H1 degradation revealed by Ub(K29R) replacement using a K29-Ub-specific antibody and manipulation of the enzymatic machinery controlling its K29-linked ubiquitylation status.

A main utility of the Ub replacement strategy lies in its unique ability to inform on how disrupting the formation of specific Ub-linkages impacts cellular ubiquitylation processes globally. This offers insights into the scope of proteins impacted by a particular Ub-linkage type and thus the cellular functions of this Ub chain topology. Here, we provide system-wide proteomic inventories of ubiquitylome and whole proteome changes induced by abrogating each lysine-based Ub chain type under otherwise unperturbed conditions in human U2OS cells, providing a valuable resource to further explore cellular roles of linkage-specific ubiquitylation processes. For instance, this revealed prospective roles of K33- and K27-linked ubiquitylation in regulating RNA processing and mitochondrial proteins, respectively. Moreover, we discovered a distinct enrichment of nuclear and chromatin-associated factors among ubiquitylated proteins responsive to disruption of K29-linked ubiquitylation, suggesting a key role of K29-Ub in regulating chromatin status and dynamics. We demonstrate that SUV39H1, a major writer of the repressive histone mark H3K9me3, is a prominent novel target of regulation by K29-linked Ub chains, which we found are essential for proteasomal degradation of SUV39H1. The Ub replacement system proved particularly useful for the identification of targets of K29-linked ubiquitylation, due to a striking tendency of substrates to accumulate non-K29-linked Ub conjugates in the absence of K29-Ub. This behavior indicates that, at least for some K29-Ub targets such as SUV39H1, K48-linked ubiquitylation is insufficient for proteasomal degradation in line with recent findings (Kaiho-Soma et al, 2021). In fact, unlike K29-linked ubiquitylation, our data suggest that K48-Ub-linkages are not strictly required for SUV39H1 degradation, as Ub(K48R) replacement only led to partial SUV39H1 stabilization. We identified TRIP12 and TRABID as the major writer and eraser of

nuclear K29-linked Ub modifications, respectively, and we establish that these enzymes are the key effectors controlling K29-linked ubiquitylation of SUV39H1. Furthermore, our data suggest a highly interconnected role of TRIP12-dependent K29-linked ubiquitylation and non-K29-linkages catalyzed by CRL activity in driving proteasomal degradation of SUV39H1 (Fig. 4J). Unexpectedly, in vitro deubiquitylation assays with linkage-specific DUBs showed that most K48-linked Ub conjugates on SUV39H1 are assembled on pre-existing K29-linkages, suggesting that K29-Ub modifications on SUV39H1 undergo CRL-dependent branching that may facilitate efficient proteasomal degradation. However, the observation that CRL activity is, at the same time, prerequisite for K29-linked ubiquitylation of SUV39H1 suggests that CRL activity also acts upstream to initiate SUV39H1 ubiquitylation, priming its subsequent TRIP12-dependent modification with K29-linked chains, the critical signal for SUV39H1 proteolysis. Hence, the degradation of SUV39H1 is governed by an intricate interplay between TRIP12 and CRL activity, extending previous findings implicating cooperative TRIP12 and CRL involvement in PROTAC-mediated degradation of neo-substrates (Kaiho-Soma et al, 2021). When K29-linked ubiquitylation is blocked by Ub(K29R) replacement or TRIP12 depletion, the CRL-dependent priming ubiquitylations on SUV39H1 and other substrates may instead undergo extension by non-K29-linkages that are insufficient for driving proteasomal degradation, thus leading to accumulation of ubiquitin-modified forms of these targets, as we observed. The CRL-dependent priming and extension of K29-linked Ub polymers on SUV39H1 may involve distinct CRL complexes, in line with several CRL adapters having previously been implicated in SUV39H1 degradation (Shan et al, 2024; Zhang et al, 2018). Interestingly, the CRL-type CUL5^ASB7 E3 ligase complex was recently shown to be required for SUV39H1 turnover (Zhou et al, 2025), suggesting that it may represent the Ub ligase activity priming TRIP12-dependent K29-linked ubiquitylation of SUV39H1 that drives its degradation.

Histone modifications are a central component of epigenetic control of gene expression; however, how epigenetic writers are themselves controlled is an area that remains understudied. We identified an important role for K29-linked Ub chains in the maintenance of the cellular epigenetic landscape that selectively impinges on H3K9 methylation status. Contrary to the impact on SUV39H1, we found no evidence of altered abundance of a range of other known H3K9me3 writers and erasers with K29-Ub abrogation, suggesting that the role of this Ub-linkage type in regulating H3K9me3 status may be predominantly exerted via TRIP12-dependent stimulation of SUV39H1 turnover. Notably, while disrupting K29-Ub-dependent degradation of SUV39H1 significantly increased total H3K9me3 abundance, this predominantly manifested as hyper-accumulation of this mark within existing H3K9me3 domains, indicative of deregulation of enzyme activity rather than targeting mechanism. Why SUV39H1 is uniquely controlled by K29-linked ubiquitylation among H3K9me3 regulatory factors remains to be addressed. During cell division, H3K9me3 is established with slow kinetics on new histones, limiting accumulation of this repressive modification (Alabert et al, 2015). Tight control of SUV39H1 levels might be needed to control H3K9me3 propagation, matching maintenance via SUV39H1 read-write function (Allshire and Madhani, 2018; Reinberg and Vales, 2018) with cell cycle speed. It will be

interesting to explore whether K29-Ub-mediated control of SUV39H1 plays a role in the reorganization of H3K9me3 patterns during development and/or erasure of H3K9me3 during cellular reprogramming. Aberrant epigenetic regulation is a prevalent feature of multiple cancer types (Berdasco and Esteller, 2010). We show here a putative disparity in survival outcomes for cancer patients when stratified by expression of both TRIP12 and SUV39H1. This was particularly evident for NSCLC patients, a subgroup of which also showed lower SUV39H1 protein levels in cancer samples with relatively higher TRIP12 that was associated with a favorable prognosis. These observations emphasize the strong anti-correlative relationship between SUV39H1 and TRIP12 and its potential importance for fitness and survival.

In summary, we describe here a robust panel of cell lines for conditional abrogation of all lysine-based Ub topologies with broad applicability across cell biology, exemplified by the discovery of a novel role for K29-linked Ub in promoting epigenome integrity via regulation of SUV39H1 stability, with implications for cell physiology and disease.

# Methods

### Reagents and tools table

| Reagent/ resource | Reference or source | Identifier or catalog Number |
|---|---|---|
| **Experimental models** | | |
| HeLa | ATCC | CCL-2 |
| RPE-1 | ATCC | CRL-4000 |
| TIG3 | (Alabert et al, 2015) | n/a |
| U2OS | ATCC | HTB-96 |
| U2OS/shUb | (Shearer et al, 2022) | n/a |
| U2OS/shUb/HA-Ub(WT) | (Shearer et al, 2022) | n/a |
| U2OS/shUb/HA-Ub(K6R) | This study | n/a |
| U2OS/shUb/HA-Ub(K11R) | This study | n/a |
| U2OS/shUb/HA-Ub(K27R) | (Shearer et al, 2022) | n/a |
| U2OS/shUb/HA-Ub(K29R) | (Shearer et al, 2022) | n/a |
| U2OS/shUb/HA-Ub(K33R) | This study | n/a |
| U2OS/shUb/HA-Ub(K48R) | This study | n/a |
| U2OS/shUb/HA-Ub(K63R) | (Shearer et al, 2022) | n/a |
| U2OS/GFP-RNF168 | (Doil et al, 2009) | n/a |
| U2OS/GFP-SUV39H1 | This study | n/a |
| **Recombinant DNA** | | |
| pPuro-shUb | (Xu et al, 2009) | n/a |

| Reagent/ resource | Reference or source | Identifier or catalog Number |
|---|---|---|
| pcDNA5/FRT/TO-HA-Ub WT | (Shearer et al, 2022) | n/a |
| pcDNA5/FRT/TO-HA-Ub K6R | This study | n/a |
| pcDNA5/FRT/TO-HA-Ub K11R | This study | n/a |
| pcDNA5/FRT/TO-HA-Ub K27R | (Shearer et al, 2022) | n/a |
| pcDNA5/FRT/TO-HA-Ub K29R | (Shearer et al, 2022) | n/a |
| pcDNA5/FRT/TO-HA-Ub K33R | This study | n/a |
| pcDNA5/FRT/TO-HA-Ub K48R | This study | n/a |
| pcDNA5/FRT/TO-HA-Ub K63R | (Shearer et al, 2022) | n/a |
| pFN18A-HaloTag-MultiDsk-6xHis | This study | n/a |
| GFP-SUV39H1 | This study | n/a |
| FLAG-SUV39H1 | This study | n/a |
| HA-TRABID | This study | n/a |
| HA-TRABID C443S | This study | n/a |
| HA-TRABID T14A/C443S | This study | n/a |
| GFP-TRIP12 (siR) | This study | n/a |
| GFP-TRIP12 HECT* (siR) | This study | n/a |
| GFP-TRIP12 WWE* (siR) | This study | n/a |
| **Antibodies** | | |
| Actin | Millipore | MAB1501, RRID:AB_2223041 |
| GAPDH | Santa Cruz | sc-20357, RRID:AB_641107 |
| GFP | Merck | 11814460001, RRID:AB_390913 |
| HA | Roche | 11867423001, RRID;AB_390918 |
| FLAG | Sigma-Aldrich | F1804, RRID:AB_262044 |
| RPS27A | Abcam | ab111598, RRID:AB_10863285 |
| Tubulin | Sigma-Aldrich | T9026, RRID:AB_2541185 |
| UBA52 | Thermo Fisher | PA5-23685, RRID:AB_2541185 |
| Ub (FK2) | Enzo | BML-PW8810, RRID:AB_2820835 |
| Ub (P4D1) | Santa Cruz | sc-8017, RRID:AB_2762364 |
| Ub K11-linked | Sigma-Aldrich | SAB5701121, Sigma |
| Ub K27-linked | Abcam | ab181537, RRID:AB_2713902 |
| Ub K33-linked | Thermo Fisher | PA5-120623, RRID:AB_2914195 |
| Ub K48-linked (Apu2) | Sigma-Aldrich | 05-1307, RRID:AB_11213655 |
| Ub K63-linked | Enzo | BML-PW0600-0025, RRID:AB_2052278 |
| SUV39H1 | Cell Signaling | 8729 T, RRID:AB_10829612 |
| SUV39H2 | Ls Bio | LS-C116360, RRID:AB_10914491 |

| Reagent/ resource | Reference or source | Identifier or catalog Number |
|---|---|---|
| TRABID | Abcam | ab262879 |
| TRIP12 | Bethyl Labs | A301-814A, RRID:AB_1264344 |
| JMJD1B | Cell Signaling | 2621S, RRID:AB_915946 |
| JMJD2A | Novus | NB110-40585, RRID:AB_669535 |
| RNF168 | Gift from Daniel Durocher | (Stewart et al, 2009) |
| NDC80 | Gift from Jakob Nilsson; raised against NDC80 aa1-230 by Moravian Biotechnology | n/a |
| H3K9me3 | Abcam | ab176916, RRID:AB_2797591 |
| H3K9me3 | Thermo Fisher | 61013, RRID:AB_2687870 |
| SETDB1 | Abcam | ab107225, RRID:AB_10861045 |
| BRD4 | Cell Signaling | 13440, RRID:AB_2687578 |
| Histone H3-pS10 | Merck | 06-570, RRID:AB_310177 |
| 53BP1 | Millipore | MAB3802, RRID:AB_11212586 |
| γH2AX | Millipore | 05-636, RRID:AB_309864 |
| CISD1 | Novus | NBP2-92696, RRID:AB_3462596 |
| CWC27 | Novus | NBP1-82508, RRID:AB_11030884 |
| NSFL1C | Novus | NBP2-13677, RRID:AB_3261294 |
| XPC | Bethyl Labs | A301-122 A, RRID: AB_2288476 |
| HDAC6 | Cell Signaling | 7558, RRID:AB_10891804 |
| TOPORS | (Liu et al, 2024) | n/a |
| RGS17 | Proteintech | 12549-1-AP, RRID:AB_2179954 |
| **Oligonucleotides and other sequence-based reagents** | | |
| siCTRL | Sigma-Aldrich | 5'-GGGAUACCUAGACGUUCUA-3' |
| siTRIP12-1 | Sigma-Aldrich | 5'-GCAAUUUGAUUCGUUCAGAtt-3' |
| siTRIP12-2 | Sigma-Aldrich | 5'-GCACCUAGAUUGGAUAGAA-3' |
| siTRIP12-4 | Sigma-Aldrich | 5'-AAAGUAAUAAAGAUUGUGU-3' |
| siTRIP12-5 | Sigma-Aldrich | 5'-GCAAUUUGAUUCGUUCAGA-3' |
| siTRABID-1 | Sigma-Aldrich | 5'-GAAUCGUCCUUCUGCCUUUtt-3' |
| siTRABID-2 | Sigma-Aldrich | 5'-GUGAUCAUCCCAGACCUAAtt-3' |
| siSUV39H1-1 | Sigma-Aldrich | 5'-AGAACAGCUUCGUCAUGGAtt-3' |
| siSUV39H1-2 | Sigma-Aldrich | 5'-GGCCUUCGUGUACAUCAAUtt-3' |
| siSUV39H1-3 | Sigma-Aldrich | 5'-CAAAUCGUGUGGUACAGAAtt-3' |
| Human on-target ubiquitin conjugation siRNA libraries subset 1 | Horizon | DHARG-105615 |
| Human on-target ubiquitin conjugation siRNA libraries subset 2 | Horizon | DHARG-105625 |
| Human on-target ubiquitin conjugation siRNA libraries subset 3 | Horizon | DHARG-105635 |

| Reagent/ resource | Reference or source | Identifier or catalog Number |
|---|---|---|
| **Chemicals, enzymes and other reagents** | | |
| Doxycycline | Sigma-Aldrich | D3072 |
| MLN-7243 (Ub E1i) | Active Biochem | MLN-7243 |
| NMS-873 (p97i) | Sigma-Aldrich | NMS-873 |
| MG132 | Sigma-Aldrich | MG132 |
| Actinomycin D | Sigma-Aldrich | A9415 |
| MLN-4924 (NEDDi) | Selleckchem | MLN-4924 |
| Cycloheximide | Sigma-Aldrich | C1988 |
| MZ1 | R&D Systems | MZ1 |
| Olaparib | Selleckchem | S1060 |
| GFP-Trap | Chromotek | GTA |
| DYKDDDDK Fab-Trap | Chromotek | FFA |
| **Software** | | |
| Prism v9.3.0 | GraphPad | n/a |

## Plasmids and siRNAs

Plasmids encoding Ub depletion (pPuro-shUb) and Ub replacement (pcDNA5/FRT/TO-HA-Ub WT, K27R, K29R, K33R, and K63R) were described previously (Shearer et al, 2022; Xu et al, 2009). In brief, shUb encodes siRNA sequences 5'-ACACCATTGAGAATGTCAA-3' targeting *UBC* and *UBA52*, and 5'-AGGCCAAGATCCAGGATAA-3' targeting *UBB* and *RPS27A*. Ub replacement cassettes to achieve K6R, K11R and K48R replacement were generated via sequential mutagenesis of pcDNA5/FRT/TO-HA-Ub(WT). All mutagenesis was performed using the Q5 Site-Directed Mutagenesis kit (New England Biolabs) and the following primer sets: UBA52 K6R: 5'-ATCTTTGTGAGGACCCT-CACTGG-3' and 5'-CTGCATGGTGTACCAGCC-3'; RPS27A K6R: 5'-ATTTTCGTGAGAACCCTTACG-3' and 5'-CTGCATAGCGTAATCT GG-3'; UBA52 K11R: 5'-CTCACTGGCAGAACCATCACC-3' and 5'-G GTCTTCACAAAGATCTGC-3'; RPS27A K11R: 5'-CTTACGGGGAG GACCATCACC-3' and 5'-GGTTTTCACGAAAATCTGCATAG-3'; UBA52 K48R: 5'-TTTGCCGGCAGACAGCTGGAG-3' and 5'-TATCA GACGCTGCTGGTC-3'; RPS27A K48R: 5'-TTTGCTGGCAGGCAGC TGGAA-3' and 5'-GATCAGTCTCTGCTGATCAGG-3'.

A synthetic cDNA sequence (Thermo Fisher) encoding the MultiDsk (Wilson et al, 2012) tandem Ub binding entity (TUBE) was cloned into pFN18A-HaloTag (Promega) to create pFN18A-HaloTag-MultiDsk-6xHis. GFP-SUV39H1 and FLAG-SUV39H1 were generated by Gateway recombination between pENTR221-SUV39H1 (Invitrogen Ultimate ORF IOH6289) and pDEST-53 (Thermo Fisher) and pDEST/FRT/N1-FLAG, respectively (Thermo Fisher). HA-TRABID was generated by cloning TRABID cDNA into pHM6 vector (Roche). Catalytic-dead HA-TRABID C443S as per (Harris et al, 2021) was generated from HA-TRABID using the primer set: 5'-GCAGGA GACTCCCTACTTGATTC-3' and 5'- AGTCCGGTTCCAAAGTG C-3'. Ub-binding deficient HA-TRABID T14A/C443S as per (Kristar-iyanto et al, 2015) was generated from HA-TRABID C443S using the primer set: 5'- TGAATATTGTGCGTATGAAAACTG-3' and 5'- CA

AGCCCACTTAATTCCAC-3'. GFP-TRIP12 and variants encoding HECT* and WWE* (C2034A and R869A, respectively) (Gatti et al, 2020) were a kind gift from Matthias Altmeyer and were altered with silent mutations for siRNA resistance using the following primer set: 5'- ACAGTTTTAGAGATGGATTTGAATCAG -3' and 5'-CGAACT GCCTAGAAACGCCTTC -3'.

The following siRNAs from Sigma were used: siMM1 (5'-GGGAU ACCUAGACGUUCUA-3'); siTRIP12-1 (5'-GCAAUUUGAUUCGU UCAGAtt-3'); siTRIP12-2 (5'-GCACCUAGAUUGGAUAGAA-3'); siTRIP12-4 (5'-AAAGUAAUAAAGAUUGUGU-3'); siTRIP12-5 (5'-G CAAUUUGAUUCGUUCAGA-3'); siTRABID-1 (5'-GAAUCGUCC UUCUGCCUUUtt-3'); siTRABID-2 (5'-GUGAUCAUCCCAGACC UAAtt-3'); siSUV39H1-1 (5'-AGAACAGCUUCGUCAUGGAtt-3'); siSUV39H1-2 (5'-GGCCUUCGUGUACAUCAAUtt-3'); siSUV39H1-3 (5'-CAAAUCGUGUGGUACAGAAtt-3'). All siRNAs were double-transfected at 20 nM using Lipofectamine RNAiMAX reagent (Thermo Fisher) according to the manufacturer's instructions. All siRNA transfections were harvested at 72 h post initial transfection unless otherwise indicated.

E3 ligase screening was performed using the human on-target ubiquitin conjugation siRNA libraries subset 1 (DHARG-105615, Horizon), subset 2 (DHARG-105625, Horizon), and subset 3 (DHARG-105635, Horizon). Cells were plated in 96-well black-walled optical imaging culture vessels 24 h prior to double-transfection at 50 nM, fixed after 72 h, before staining as indicated and high-content image acquisition (see immunofluorescence). Sample mean intensities were normalized to siMM1 within each 96-well plate. Z-score was calculated as:

$$Z = \frac{x - \mu}{\sigma}$$

where x is the sample intensity, μ is the population mean, and σ is the population standard deviation.

## Cell culture

Human U2OS (HTB-96), HeLa (CCL-2) and RPE-1 (CRL-4000) cells were obtained from ATCC, grown under standard conditions at 37 °C and 5% $CO_2$ in DMEM (Gibco) supplemented with 10% FBS (v/v) and Penicillin-Streptomycin (Thermo Fisher). TIG3 cells were previously described (Alabert et al, 2015). U2OS cell lines stably expressing the Ub replacement expression cassette (U2OS/shUb/HA-Ub) were generated by co-transfection of U2OS/shUb cells (Shearer et al, 2022) with pcDNA5/FRT/TO-UBA52/HA-RPS27A and pOG44 (Thermo Fisher) plasmids, followed by selection with Hygromycin B (Thermo Fisher). Stable clones were carefully screened for uniform expression of ectopic Ub at levels comparable to those of endogenous Ub and maintained under selection with Puromycin (Sigma-Aldrich), Blasticidin and Hygromycin B (Thermo Fisher). Where indicated, Ub replacement was achieved by doxycycline (DOX) (0.5 μg/ml) treatment for 72 h, unless otherwise indicated. All key findings using Ub(K29R) replacement cells were validated in two independent clones (c85 and c93). Unless otherwise indicated, the Ub(K29R)-c93 clone was used in experiments shown in this study. All cells were regularly tested negative for *Mycoplasma* infection. The cell lines were not authenticated.

U2OS cells stably expressing GFP-RNF168 (U2OS/GFP-RNF168) were described previously (Doil et al, 2009). U2OS/GFP-SUV39H1 cells were generated by transfection and selection with G418 (Invivogen) at 400 μg/ml. Drosophila S2-DRSC suspension cells were cultivated at 25 °C with 5% $CO_2$ in spinners using M3 + BPYE media: comprising Shields and Sang M3 Insect Medium (Sigma-Aldrich), $KHCO_3$ (Sigma-Aldrich), yeast extract (Sigma-Aldrich), bactopeptone (BD Biosciences), 10% heat-inactivated FBS (GE Hyclone), and 1X penicillin/streptomycin (Gibco). hTERT TIG3 fibroblasts from fetal lung were additionally supplemented in culture with 10% nonessential amino acids.

Incucyte confluence assays were performed as described (Shearer et al, 2022). In brief, cellular confluence was assessed using the Incucyte S3 Live-Cell Analysis System (Sartorius). Unless otherwise indicated, the following standard treatment concentrations were used: doxycycline (1 μg/ml, Sigma-Aldrich), MLN-7243 (E1i; 5 μM, Active Biochem), NMS-873 (5 μM, Sigma-Aldrich), MG132 (20 μM; Sigma-Aldrich), Actinomycin D (ActD; 2 μg/ml, Sigma), MLN-4924 (NEDDi; 1 μM, Selleckchem), Cycloheximide (CHX; 20 μg/ml, Sigma-Aldrich), MZ1 (500 nM, R&D Systems), and Olaparib (1 μM, AstraZeneca). DNA double-strand breaks (DSBs) were induced where indicated using the YXLON Smart X-ray system.

## RT-qPCR

Total RNA was extracted from cells following the manufacturer's protocol (RNeasy kit, Qiagen). cDNA was synthesized from RNA by reverse transcription PCR (iScript cDNA Synthesis Kit, Bio-Rad). Real-time quantitative PCR was performed using the Stratagene Mx3005P System and Brilliant III Ultra-Fast SYBR Green QPCR Master Mix (Agilent). GAPDH mRNA levels were used as a control for normalization. The following primers were used for amplification of the respective cDNAs: GAPDH (forward): 5'-CAGAACATCATCCC TGCCTCTAC-3'; GAPDH (reverse): 5'- TTGAAGTCAGAGGAGAC CACCTG-3'; HA-RPS27A (forward): 5'-TACCCTTACGATGTACCA GA-3'; HA-RPS27A (reverse): 5'-GAGGGTTCAACCTCGAGGGT-3'; RPS27A (forward): 5'-TCGTGGTGGTGCTAAGAAAAGG-3'; RPS 27A (reverse): 5'-TTCAGGACAGCCAGCTTAACCT-3; UBB (forward): 5'-CTTTGTTGGGTGAGCTTGTTTGT-3'; UBB (reverse): 5'-GACCTGTTAGCGGATACCAGGAT-3'; UBA52 (forward): 5'-CTG CGAGGTGGCATTATTGAG-3'; UBA52 (reverse): 5'-GTTGACAG-CACGAGGGTGAAG-3'; UBC (forward): 5'-GTGTCTAAGTTTCCC CTTTTAAGG-3'; UBC (reverse): 5'-TTGGGAATGCAACAACTTTA TTG-3'.

## In silico local deformation prediction of a single amino acid mutation

The amino acid sequence for WT and single K-to-R mutants of human Ub (P0CG48, Uniprot) were input to AlphaFold2_mmseqs2 via ColabFoldv1.5.5 (Mirdita et al, 2022) with standard settings and no template mode. Output structures were input to PDAnalysis (McBride et al, 2023). Effective strain between the WT structure and individual mutations were calculated using an average of five output structures from AlphaFold2. Effective strain was then superimposed onto the default structure using the b-factor column via a custom Python3 script. All protein structures were visualized with ChimeraX v1.7.

## Antibodies

The following primary antibodies were used: Actin (MAB1501, Millipore (RRID:AB_2223041)); GAPDH (sc-20357, Santa Cruz (RRID:AB_641107)); GFP (11814460001, Merck (RRID:AB_390913)); HA (11867423001, Roche (RRID;AB_390918)), FLAG (F1804, Sigma (RRID:AB_262044)); RPS27A (ab111598, Abcam (RRID:AB_10863285)); Tubulin (T9026, Sigma (RRID:AB_2541185)); UBA52 (PA5-23685, Thermo (RRID:AB_2541185)); Ub (FK2) (BML-PW8810, Enzo (RRID:AB_2820835)); Ub (P4D1) (sc-8017, Santa Cruz (RRID:AB_2762364)); Ub K11-linked (SAB5701121, Sigma); Ub K27-linked (ab181537, Abcam (RRID:AB_2713902)); Ub K33-linked (PA5-120623, Thermo (RRID:AB_2914195)); Ub K48-linked (Apu2) (05-1307, Sigma (RRID:AB_11213655)); Ub K63-linked (BML-PW0600-0025, Enzo (RRID:AB_2052278)); SUV39H1 (8729 T, Cell Signaling (RRID:AB_10829612)); SUV39H2 (LS-C116360, Ls Bio (RRID:AB_10914491)); TRABID (ab262879, Abcam); TRIP12 (A301-814A, Bethyl Labs (RRID:AB_1264344)); JMJD1B (2621S, CST (RRID:AB_915946)); JMJD2A (NB110-40585, Novus (RRID:AB_669535)); RNF168 (kind gift from Daniel Durocher (Stewart et al, 2009)); NDC80 (kind gift from Jakob Nilsson; raised against NDC80 aa1-230 by Moravian Biotechnology); H3K9me3 (ab176916, Abcam (RRID:AB_2797591) and 61013, Thermo (RRID:AB_2687870)); SETDB1 (ab107225, Abcam (RRID:AB_10861045)); BRD4 (13440, CST (RRID:AB_2687578)); Histone H3-pS10 (06-570, Merck (RRID:AB_310177)); 53BP1 (MAB3802, Millipore (RRID:AB_11212586)); γH2AX (05-636, Millipore (RRID:AB_309864)); CISD1 (NBP2-92696, Novus (RRID:AB_3462596)); CWC27 (NBP1-82508, Novus (RRID:AB_11030884)); NSFL1C (NBP2-13677, Novus (RRID:AB_3261294)); XPC (A301-122 A, Bethyl (RRID: AB_2288476)); HDAC6 (7558, CST (RRID:AB_10891804)); TOPORS (Liu et al, 2024); RGS17 (12549-1-AP, Proteintech (RRID:AB_2179954)).

## Cell lysis, immunoblotting, and immunoprecipitation

Immunoprecipitation (IP) of HA-tagged proteins was performed using Anti-HA Affinity matrix (1181501600, Roche) under denaturing conditions as described (Shearer et al, 2022). Cell lysis was performed using RIPA buffer (140 mM NaCl; 10 mM Tris-HCl (pH 8.0); 0.1% sodium deoxycholate (w/v); 1% Triton X-100 (v/v); 0.1% SDS (w/v); 1 mM EDTA; 0.5 mM EGTA). GFP-trap and DYKDDDDK Fab-Trap (Chromotek) pulldown was performed according to the manufacturer's instructions. Unless indicated, all whole lysate and immunoprecipitation samples were prepared and washed in non-denaturing buffer (150 mM NaCl; 50 mM Tris-HCl, pH 8.0; 0.5 mM EDTA). All samples obtained under denaturing conditions were lysed and washed in denaturing buffer (50 mM NaCl; 20 mM Tris-HCl, pH 7.5; 0.5% sodium deoxycholate (w/v); 0.5% igepal (v/v); 0.5% SDS (w/v); 1 mM EDTA). All samples indicated as subject to MultiDsk pulldown were lysed and washed in a stringent buffer (50 mM Tris-HCl, pH 8.0; 1 M NaCl; 1% Igepal v/v; 0.1% SDS w/v). Halo-MultiDsk was conjugated to HaloLink Resin (Promega) prior to IP for 1 h at RT in wash buffer (100 mM Tris-HCl, pH 7.5; 150 mM NaCl; 0.05% Igepal v/v). All lysis and wash buffers were supplemented with PMSF Protease Inhibitor (1 mM, Thermo Fisher), EDTA-free protease inhibitor Cocktail Tablets (Roche), *N*-ethylmaleimide (1.25 mM, Sigma-Aldrich) and DUB inhibitor PR-619 (50 μM, Calbiochem).

Met-Gly-Ser-His6-3C-codon-optimized LotA-N (catalytically inactive C13A mutant) with a C-terminal (Ala-Ser)3 linker followed by an AviTag (Avidity) was expressed from a pET vector in *E. coli* (ArcticExpress (DE3), Agilent), and purified as previously described (Suryo Rahmanto et al, 2023). Following cell lysis in modified RIPA buffer, pre-cleared protein lysate was incubated with 1 μM biotin-conjugated LotA-N for 2 h at 4 °C. High-capacity Neutravidin beads (15 μl per sample) were added and incubated for 1 h at 4 °C on a rotating wheel to precipitate biotin-conjugated probes. Beads were washed four times with modified RIPA buffer supplemented with protease inhibitors and 10 mM *N*-ethylmaleimide to remove nonspecific binders prior to elution in NuPAGE LDS 2×Sample Buffer (Life Technologies).

## Chromatin fractionation

U2OS cells were collected by scraping in ice-cold PBS and subsequent centrifugation for 5 min at 400 g. The pellets were lysed with ice-cold cell lysis buffer (10 mM Tris, pH 8.0; 10 mM KCl; 1.5 mM MgCl$_2$; 0.34 M sucrose; 10% glycerol; 0.1% Triton X-100) supplemented with protease, phosphatase and DUB inhibitors. After centrifugation for 5 min at 2000×g, the soluble fraction was recovered. The pellets were carefully washed once in lysis buffer and centrifuged as in the previous step. Pellets were resuspended in RIPA buffer supplemented with benzonase and protease, phosphatase and DUB inhibitors and incubated in a thermomixer for 15 min at 37 °C. A last centrifugation step (16,000×g for 10 min at 4 °C) allowed the recovery of the solubilized chromatin-bound proteins.

## Immunofluorescence

Coverslips were fixed in 10% formalin buffer (VWR) for 15 min at room temperature, permeabilized with 0.5% Triton X for 5 min and blocked with 5% BSA (Sigma-Aldrich). Cells were stained with primary antibody for 1 h at RT, washed with PBS and stained with a combination of Alexa Fluor secondary antibodies (Thermo Fisher) and 4',6-diamidino-2-phenylindole (DAPI; molecular probes) for 30 min at RT. Coverslips were then washed, dried and mounted using Mowiol (Sigma-Aldrich). Where indicated, cells were pre-extracted prior to fixation using CSK buffer (100 mM NaCl; 10 mM HEPES; 3 mM MgCl$_2$; 300 mM sucrose; 0.25% Triton X; 1 mM PMSF). Where indicated, nascent DNA synthesis was estimated by incubation with 10 μM 5-ethynyl-2'-deoxyuridine (EdU; Thermo Fisher) for 1 h prior to fixation. EdU incorporation was labeled using Click-iT Plus EdU Alexa Fluor 647 Imaging Kit (Thermo Fisher) according to the manufacturer's instructions. Transcription and translation efficiency was assessed as described (Shearer et al, 2022). Quantitative image-based cytometry (QIBC) was performed as described (Shearer et al, 2022). In brief, images were acquired and analyzed using the ScanR high-content screening platform (Olympus). Cell cycle gating was performed using DAPI and EdU staining, with mitotic figures determined as a percentage of the whole population based on histone H3-pS10 staining. Representative images were acquired with a confocal microscope (LSM 880; Carl Zeiss), mounted on a confocal laser-scanning microscope (Zeiss AxioObserver.Z1; Carl Zeiss) equipped with a Plan-Apochromat 40×/1.3 NA oil immersion objective. Image acquisition was performed with ZEN 2.1 software (Carl Zeiss). Raw images were exported as TIFF files, and identical settings were used on all images of a given experiment when adjustments in image brightness were applied.

## Deubiquitylation assays

FLAG-SUV39H1 was immobilized on DYKDDDDK Fab-Trap beads (Chromotek) in RIPA buffer supplemented with EDTA-free protease inhibitor Cocktail Tablets (Roche), *N*-ethylmaleimide (1.25 mM; Sigma-Aldrich) and DUB inhibitor PR-619 (50 μM; Calbiochem), and beads were washed twice in RIPA and twice in TBS (50 mM Tris, pH 7.4; 150 mM NaCl). DUBs (TRABID, OTUB1, and USP2cc; K-400, Boston Biochem) were diluted to 1x according to the manufacturer's instructions and added in a total volume of 30 μl of TBS, and reactions were incubated for 1 h at 37 °C with shaking, eluted in 2xLaemmli sample buffer and analyzed by immunoblotting.

## Mass spectrometry (whole proteome analysis)

Protein samples were collected in modified RIPA buffer (50 mM Tris, pH 7.5; 150 mM NaCl; 1 mM EDTA; 1% NP-40; 0.1% sodium deoxycholate) supplemented with protease inhibitors (Complete Protease Inhibitor Cocktail Tablets, Roche), 1 mM sodium orthovanadate, 5 mM β-glycerophosphate, 5 mM sodium fluoride, and 10 mM *N*-ethylmaleimide. Chromatin-bound proteins were extracted by the addition of NaCl to a final concentration of 500 mM and pulse sonication at 4 °C for 10 min. Total protein concentrations were estimated using the QuickStart Bradford Protein assay (Bio-Rad) from the pre-cleared lysates. Proteins were precipitated in fourfold excess of ice-cold acetone and subsequently re-dissolved in denaturation buffer (6 M urea; 2 M thiourea; 10 mM HEPES, pH 8.0). Cysteines were reduced with 1 mM dithiothreitol and alkylated with 5.5 mM chloroacetamide. Proteins were digested with MS-approved Trypsin (Sigma-Aldrich). Protease digestion was stopped by the addition of trifluoroacetic acid to 0.5%, and precipitates were removed by centrifugation. Peptides were purified and desalted using reversed-phase Sep-Pak C18 cartridges (Waters) and eluted in 50% acetonitrile. For full proteome profiling, the peptides were supplemented with trifluoroacetic acid up to 1% and analysed on a quadrupole Orbitrap mass spectrometer (Astral, Thermo Fisher) equipped with a UHPLC system (Vanquish, Thermo Fisher).

Raw data files were converted to mzML format using MSConvert and analyzed using FragPipe (development v.21.1) LFQ-MBR workflow (Kong et al, 2017; Teo et al, 2021; Yu et al, 2021a). Parent ion and MS2 spectra were searched against a database containing 98,566 human protein sequences obtained from UniProtKB (April 2018 release) using the MSFragger search engine. Spectra were searched with a mass tolerance of 6 ppm in MS mode, 20 ppm in HCD MS2 mode and strict trypsin specificity, allowing up to two miscleavages. Cysteine carbamidomethylation was searched as a fixed modification. Peptide probability was determined by the highest supporting PSM probability. Results for each experiment were aggregated and filtered at 1% peptide FDR.

Data analysis and visualization were performed using RStudio in the R 4.4.1 environment.

## Mass spectrometry (HA-tag affinity purification)

HA-tag affinity purification was performed using Anti-HA Affinity Matrix (Roche). Partial on-bead digest was performed with buffer (2 M urea; 2 mM DTT; 20 μg/ml Trypsin; 50 mM Tris, pH 7.5) incubated at 37 °C for 30 min at 1400 rpm. Supernatants were transferred to new tubes, alkylated with 25 mM chloroacetamide (CAA) and further digested overnight at RT and 1000 rpm. Digestion was terminated with 1% trifluoroacetic acid (TFA). Peptides were desalted and purified using styrenedivinylbenzene-reversed-phase sulfonate (SDB-RPS) StageTips prepared in 0.2% TFA. Peptides were washed and eluted with elution buffer (80% acetonitrile (ACN); 1% ammonia) prior to vacuum-drying. Dried peptides were reconstituted in 2% ACN and 0.1% TFA.

All samples were loaded onto Evotips Pure and measured with a data-independent acquisition (DIA) method. 200 ng of peptides were partially eluted from Evotips with <35% acetonitrile and analyzed with an Evosep One LC system (Evosep Biosystems) coupled online to an Orbitrap mass spectrometer (Orbitrap Astral, Thermo Fisher) (Bache et al, 2018; Guzman et al, 2024; Heil et al, 2023). Eluted peptides were separated on an 8-cm-long PepSep column (150 μm inner diameter packed with 1.5 μm of Reprosil-Pur C18 beads (Dr Maisch)) in a standard preset gradient method (21 min, 60 samples per day) with a stainless emitter (30-μm inner diameter). The mobile phases were 0.1% formic acid in liquid chromatography (LC)–MS-grade water (buffer A) and 0.1% formic acid in acetonitrile (buffer B). Data were acquired in DIA mode. Each acquisition cycle consisted of a survey scan at a resolution of 240,000 (normalized automatic gain control target (AGC) of 500% and a maximum injection time of 100 ms. Fragment ion scans were recorded with a maximum injection time of 5 ms and with 200 windows of 3Th scanning from 380−980 m/z. Higher-energy collisional dissociation (HCD) fragmentation was set to a normalized collision energy of 25%.

Raw files were analyzed with the directDIA workflow in Spectronaut v.18.6 (Bruderer et al, 2015) using Default settings. Data filtering was set to "Qvalue". "Cross-run normalization" was enabled with the strategy of "local normalization" based on rows with "Qvalue complete". FDR was set to 1% at both the protein and peptide precursor levels. Raw data were searched against the human proteome reference database, including isoform information (Uniprot March 2023). Data were filtered for 60% valid values across WT or Mutant samples (protein groups with >40% missing values were excluded from downstream statistical analysis). Protein intensities were then log2-transformed for downstream statistical and bioinformatics analysis. Imputation of missing data was performed by random numbers drawn from a normal distribution with a width of 0.3 and a downshift of 1.8 applied to each sample. Significance was determined through an unpaired two-tailed *t*-test corrected by Benjamini–Hochberg at FDR of 0.1 and fold change of 1.5 (Dowell et al, 2021; Lin et al, 2022). GO-term enrichment was performed using ClusterProfiler (Yu et al, 2012). Data analysis and visualization were performed using Jupyter Notebook (Python 3.9) and RStudio (v2023.03.1 + 446) with R (v4.3.0) for ClusterProfiler-based analysis.

## Mass spectrometry (Ub-modified proteome (diGly) analysis)

Cells were collected in modified RIPA buffer (50 mM Tris pH 7.5, 150 mM NaCl, 1 mM EDTA, 1% NP-40, and 0.1% sodium deoxycholate) supplemented with protease inhibitors (Complete protease inhibitor cocktail tablets, Roche Diagnostics), 1 mM sodium orthovanadate, 5 mM β-glycerophosphate, 5 mM sodium fluoride, and 10 mM *N*-ethylmaleimide. Chromatin-bound proteins were extracted by the addition of NaCl to a final concentration of 500 mM and pulse sonication at 4 °C for 10 min. Total protein concentrations were estimated using the QuickStart Bradford

Protein assay (Bio-Rad) from the pre-cleared lysates, and a minimum of 1 mg of protein from each samples was subjected to OtUBD pulldown to isolate total cellular Ub conjugates (Zhang et al, 2022). Typically, 15 μl of OtUBD resin (bed volume) was used for each 1 mg of lysate protein, and the mixture was incubated at 4 °C for 2 h with rotation. The resin was allowed to settle by gentle centrifugation at 1000×g for 1 min. The unbound solution was drained and the pelleted OtUBD resin was washed three times with 8 M Urea and once with 1% SDS-PBS prior to elution. Bound proteins were eluted by incubating the resin with 2x resin volumes of 2x LDS sample buffer for 15 min at RT with rotation. The eluted Ub-modified proteins were reduced using 5 mM DTT and then alkylated with the addition of 11 mM chloroacetamide in the dark at room temperature for 30 min. Samples were subsequently cleaned up by SP3 and digested in solution using sequencing-grade modified trypsin (Sigma-Aldrich) at 37 °C for 16 h. Digested peptides were desalted on reverse-phase C18-StageTips prior to MS analysis.

For Ub remnant profiling, cells were lysed in modified RIPA buffer supplemented with protease inhibitors (Complete protease inhibitor cocktail tablets, Roche Diagnostics), 1 mM sodium orthovanadate, 5 mM β-glycerophosphate, 5 mM sodium fluoride, and 10 mM N-ethylmaleimide. Proteins were digested with sequencing-grade modified trypsin (Sigma-Aldrich), and di-glycine-modified peptides were enriched using di-glycine-lysine antibody resin (Cell Signaling Technology). Peptides were analyzed on an Orbitrap mass spectrometer (Orbitrap Astral, Thermo Fisher) equipped with a UHPLC system (Vanquish Neo, Thermo Fisher). Raw data files were analyzed using MaxQuant (development version 1.5.2.8). Parent ion and MS2 spectra were searched against a database containing 98,566 human protein sequences obtained from UniProtKB (April 2018 release) using the Andromeda search engine. Spectra were searched with a mass tolerance of 6 ppm in MS mode, 20 ppm in HCD MS2 mode and strict trypsin specificity, allowing up to three miscleavages. Cysteine carbamidomethylation was searched as a fixed modification, whereas protein N-terminal acetylation and methionine oxidation were searched as variable modifications. The dataset was filtered based on posterior error probability (PEP) to arrive at a false discovery rate (FDR) of less than 1% estimated using a target-decoy approach.

## Mass spectrometry analysis of histone PTMs (EpiQMAx)

U2OS/shUb/HA-Ub(WT) and U2OS/shUb/HA-Ub(K29R) cells were seeded in quadruplicate 15-cm dishes at a density of 1 million cells per dish and treated or not with DOX for 72 h. Cells were trypsinized and washed twice with PBS prior to collection in media. From this suspension, 10 million cells were pelleted at 300×g for 5 min and washed twice with PBS. The cell pellets were snap-frozen and shipped on dry ice to EpiQMAx GmbH for further processing. Acid-extracted histones were resuspended in Laemmli sample buffer, separated by a 14–20% gradient SDS–PAGE, and stained with Coomassie (Brilliant Blue G-250, 35081.01). Protein bands within the 15–23 kDa molecular weight range were excised as single bands, destained in a 50% acetonitrile/50 mM ammonium bicarbonate solution, and chemically modified by propionylation for 30 min at room temperature with 2.5% propionic anhydride (Sigma-Aldrich) in ammonium bicarbonate, pH 7.5. Proteins were then digested overnight with 200 ng of trypsin (Promega) in 50 mM

ammonium bicarbonate. The resulting supernatant was desalted using C18-Stagetips (reversed-phase resin) and carbon Top-Tips (Glygen) according to the manufacturer's instructions, speed vacuumed until dry, and stored at −20 °C until mass spectrometry analysis.

Peptides were injected in an Ultimate 3000 RSLCnano system (Thermo Fisher) and separated on a 15-cm analytical column (75 μm ID with ReproSil-Pur C18-AQ 2.4 μm (Dr. Maisch)) using a gradient from 4% B to 90% B (solvent A 0.1% FA in water, solvent B 80% ACN, 0.1% FA in water) over 90 min at a flow rate of 300 nl/min. The effluent from the HPLC was directly electro-sprayed into an Exploris 240 mass spectrometer (Thermo Fisher). The mass spectrometer was operated in data-dependent mode to automatically switch between full scan MS and MS/MS acquisition. Survey full scan MS spectra (from m/z 375 to 1600) were acquired with resolution R = 60,000 at m/z 400 (AGC target of $3 \times 10^6$). The ten most intense peptide ions with charge states between 2 and 5 were sequentially isolated to a target value of $1 \times 10^5$ and fragmented at 27% normalized collision energy. Typical mass spectrometric conditions were as follows: spray voltage, 1.5 kV; no sheath and auxiliary gas flow; heated capillary temperature, 250 °C; and ion selection threshold, 33,000 counts.

Raw files were analyzed using Skyline software against histone H3 and H4 peptides and their respective PTMs, with a precursor mass tolerance of 5 ppm. The chromatogram boundaries of +2 and +3 charged peaks were validated, and the total area MS1 under the first four isotopomers was extracted for relative quantification and comparison between experimental groups. For co-eluting isobaric peptides (e.g., H3K36me3 and H3K27me2K36me1), the Total Area MS1 was resolved using their unique MS2 fragment ions. The average ratio of analogous ions (e.g., y7 vs. y7) was used to calculate the respective contribution of the precursors to the isobaric MS1 peak.

Relative abundances (percentages) were calculated as follows, using H3K18 acetylation as an example (where "ac" denotes acetylation and "un" denotes unmodified):

$$\%H3K18ac = \frac{(H3K18ac\_K23un + H3K18ac\_K23ac)}{(H3K18un\_K23un + H3K18ac\_K23unmod + H3K18un\_K23ac + H3K18ac\_K23ac)}$$

## Quantitative ChIP-seq

U2OS/shUb/HA-Ub(WT) and U2OS/shUb/HA-Ub(K29R) cells were seeded in a 15-cm plate (3 plates per cell line). Cells were treated or not with DOX for 72 h. Fixation buffer A (truChIP Chromatin Shearing Kit, Covaris) was added to each plate, followed by fresh formaldehyde to a final concentration of 1% for fixation at room temperature with constant movement for 10 min. Glycine was then added to quench the reaction for 5 min at room temperature with constant movement. The fixed cells were washed with ice-cold PBS, collected and centrifuged for 5 min at 500×g at 4 °C, transferred to Eppendorf tubes, and centrifuged again for an additional 5 min at 500×g at 4 °C. The cell pellets were snap-frozen in liquid nitrogen and stored at −80 °C until lysis.

Nuclei isolation was performed on fixed cells using the truChIP Chromatin Shearing Kit (Covaris) according to the manufacturer's instructions. Cells were sonicated using a Covaris E220 Evolution with settings: 5% duty cycle, 200 cycles/burst, 20 min processing time, 7 °C bath temperature, and full water level. Sonicated

chromatin was centrifuged at 14,000 rpm at 4 °C for 10 min, and the supernatant was retained. Drosophila S2 cells were processed in parallel using the same procedure. For each ChIP reaction, 30 µg of U2OS chromatin was mixed with 0.5% sonicated Drosophila S2 chromatin and diluted up to 500 µl with dialysis buffer (4% glycerol; 10 mM Tris-HCl, pH 8.0; 1 mM EDTA; 0.5 mM EGTA) and 400 µl of incubation buffer (2.5% Triton X-100; 0.25% sodium deoxycholate; 0.25% SDS; 0.35 M NaCl; 10 mM Tris-HCl, pH 8.0) supplemented with leupeptin, aprotinin, pepstatin, and PMSF. Chromatin was pre-cleared with Protein A/G Agarose beads (Thermo Fisher) for 1 h at 4 °C. After preclearing, 10 µl were set aside as input, and the remaining chromatin was incubated overnight at 4 °C with 2 µl H3K9me3 antibody. Protein A/G Agarose beads were pre-blocked with 1 mg/ml BSA in RIPA buffer overnight at 4 °C and then incubated for 3 h with the chromatin-antibody mix.

ChIPs were washed three times in ice-cold RIPA buffer (140 mM NaCl; 10 mM Tris-HCl, pH 8.0; 1 mM EDTA; 1% Triton X-100; 0.1% SDS; 0.1% sodium deoxycholate; 1 mM PMSF), three times in RIPA buffer with 0.5 M NaCl, once in LiCl buffer (250 mM LiCl; 10 mM Tris-HCl, pH 8.0; 1 mM EDTA; 0.5% igepal CA-630; 0.5% sodium deoxycholate), and twice in TE (10 mM Tris-HCl, pH 8.0; 1 mM EDTA). Inputs and washed beads were incubated with 50 µg RNase A (Sigma-Aldrich) for 30 min at 37 °C, followed by the addition of SDS and NaCl to final concentrations of 0.5% and 100 mM, respectively. Samples were incubated with proteinase K (10 µg) for 10 h at 37 °C, followed by 6 h at 65 °C for de-crosslinking. DNA was purified using the MinElute PCR purification kit (QIAGEN) and quantified using a Qubit fluorometer.

The input material and immunoprecipitated DNA were prepared for sequencing using the KAPA Hyperprep kit protocol (Roche) with Illumina-compatible indexed adapters (IDT). The DNA underwent end repair, A-tailing, and amplification, with 6–7 PCR cycles. DNA fragments between 200-700 bp were selected using Agencourt AMPure XP beads (Beckman Coulter) and sequenced paired-end on a NextSeq 2000 (Illumina).

Reads underwent filtering for adapters and low quality using cutadapt (v.4.4). Subsequently, reads were aligned to a hybrid mouse (mm10) and fly (dm6) genome using bowtie2 (v.2.3.4). Duplicate reads were then removed with umi-tools (v.1.1.4), and ENCODE hg38 blacklist regions were masked. Only reads with a mapping quality above 30 were retained for subsequent downstream analyses. QC stats were produced from the bam files according to the ENCODE ChIP-seq pipeline (v.2.2.1).

For spike-in normalization, downsampling factors were calculated for each sample as per (Fursova et al, 2019). These factors were determined based on the ratio of ChIP dm6 reads to Input dm6 reads, normalized by the corresponding Input mm10 reads:

$$\text{Downsampling factor} = \alpha * \frac{1}{\text{ChIP dm6 reads}} * \frac{\text{Input (dm6 reads)}}{\text{Input (mm10 reads)}}$$

The largest downsampling factor for each protein was set to 1, and other factors were adjusted accordingly. To obtain reference-adjusted reads per million (RRPMs), the total number of unique reads (uniquely and multi-mapping) were divided by their corresponding downsampling factors.

Visualization tracks for ChIP-seq data were generated using Seqmonk (v.1.47.1), with 5 kb non-overlapping bins quantified as indicated for each figure.

Custom R scripts were utilized for creating Hilbert curves, and box plots. These scripts were based on various packages including HilbertCurve and ggplot2.

## Survival analysis

Kaplan–Meier survival analyses were performed using two online tools. Gene Expression Profiling Interactive Analysis (GEPIA2) was used to assess disease-free survival stratified by SUV39H1 and TRIP12 expression across 33 cancer types (Tang et al, 2019). Log-rank test was used for hypothesis evaluation. Group cut-off was set at the median, and a confidence interval of 95% was used. Hazard ratios were calculated following the Cox proportional-hazard model.

The Kaplan–Meier plotter was used to assess overall survival and first progression survival in NSCLC stratified on expression of SUV39H1 and TRIP12 (Gyorffy, 2024). Patient cut-off was set at the median. Univariate analysis, performing Cox regression was used to compute hazard ratios and $p$ value.

## Academia Sinica LUAD-100 Proteome Study

Data from the LUAD-100 cohort were downloaded from the Proteomics Data Commons and processed using the following files: Academia_Sinica_LUAD100_Proteome_v2.tmt10.tsv for proteomics, Academia_Sinica_LUAD100_Proteome_v2.summary.tsv for spectral counts, and

Academia_Sinica_LUAD100_Proteome_v2.peptides.tsv for peptide-level information.

Clinical data were retrieved from Academia Sinica LUAD-100_Clinical.xlsx. To advance the proteogenomic understanding of LUAD, the CPTAC program analyzed 111 tumors, of which 102 were paired with adjacent normal tissue samples, subjecting them to global proteome and phosphoproteome analysis. An optimized mass spectrometry workflow using tandem mass tags (TMT-10) was employed (Mertins et al, 2018). The LUAD discovery cohort's proteome, phosphoproteome, and acetylome data are available along with peptide spectrum matches (PSMs) and protein summary reports from the common data analysis pipeline (CDAP) (Gillette et al, 2020).

RNA sequencing data for the lung adenocarcinoma (LUAD) cohort was obtained from the Morpheus platform hosted by the Broad Institute. The specific files utilized were for the breast cancer RNA-seq data: rna_seq_breast.gct and for the LUAD RNA-seq data: rna_seq_luad.gct.

Data analysis was conducted using Python, using pandas for data manipulation, seaborn for visualization, and matplotlib for plotting. Proteomics data were preprocessed by filtering for protein-coding genes and normalizing log-transformed reporter ion intensity values (Log2 Ratio). Data from the Academia_Sinica_LUAD100_Proteome_v2.tmt10.tsv file were used, focusing on the expression data across 94 samples. Transcriptomic data were processed by extracting expression values (tpm, fpkm) averaged across the respective aliquots for LUAD cohorts. Expression data were further filtered to focus on protein-coding genes, and subsequent analyses were performed on normalized expression values.

Violin plots were created to visualize the distribution of expression levels for the TRIP12 and SUV39H1 genes across different conditions. These conditions were stratified based on

median expression levels of TRIP12, separating the data into "low TRIP12" and "high TRIP12" groups for both proteomics and transcriptomics datasets. For each gene and condition, violin plots were generated using seaborn to compare expression levels across the two groups. This was done separately for proteomics and transcriptomics datasets for both the breast cancer and LUAD cohorts. Additionally, heatmaps were generated using the correlation matrices of the expression data to explore potential relationships between genes.

All scripts and code used in this analysis are provided as a Jupyter notebook, which can be accessed along with the raw data files. The notebook includes step-by-step instructions for reproducing the analysis, along with detailed comments to facilitate understanding.

## Quantification and statistical analysis

All statistical analyses were performed using Prism 9.3.0 (GraphPad Software). Statistical details, including the number of independent experiments (*n*), definition of significance and measurements, are defined in figure legends. No statistical method was used to predetermine sample size, and no data were excluded from the analyses, except for standard gating during high-content imaging analysis. Samples were not randomized, and investigators were not blinded to group allocation during data collection and analysis.

## Data availability

The mass spectrometry proteomics data have been deposited to the ProteomeXchange Consortium (Perez-Riverol et al, 2019) via the Proteomics Identification (PRIDE) partner repository (http://www.ebi.ac.uk/pride) under dataset IDs PXD056493 (Dataset EV2), PXD056564 (Dataset EV3), and PXD056295 (Dataset EV5). Sequencing data produced in this study have been deposited in NCBI's Gene Expression Omnibus (GEO) (Edgar et al, 2002) with accession code GSE279218. All other data supporting the findings of this study are available within the article and supplementary information.

The source data of this paper are collected in the following database record: biostudies:S-SCDT-10_1038-S44318-025-00599-7.

## Peer review information

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

## Acknowledgements

We thank Zhijian Chen, Matthias Altmeyer, Minglei Zhao, Daniel Durocher, and Jakob Nilsson for providing reagents, members of the Mailand lab for helpful discussions, and Martin Möckel and the IMB Mainz protein production facility for purification of OtUBD and LotA-N proteins. This work was supported by grants from Novo Nordisk Foundation (grants no. NNF14CC0001 and NNF24SA0098829), Lundbeck Foundation (grants no. R223-2016-281 and R402-2022-1321), Independent Research Fund Denmark (grant no. 9040-00038B), European Union's Horizon 2020 research and innovation program (Marie Skłodowska-Curie grant agreement no. 860517 (UBIMOTIF) and no. 846795), Danish Cancer Society (grant no. R316-A19079) and Danish National Research Foundation (DNRF195). All experimental procedures were conducted according to the highest sustainability practices, as part of the laboratory efficiency assessment framework (LEAF).

## Author contributions

**Javier Arroyo-Gomez**: Conceptualization; Investigation; Methodology; Writing—original draft; Writing—review and editing. **Matthew J Murray**: Funding acquisition; Investigation; Writing—review and editing. **Claire Guérillon**: Funding acquisition; Investigation; Writing—review and editing. **Juanjuan Wang**: Investigation; Writing—review and editing. **Ekaterina Isaakova**: Investigation; Writing—review and editing. **Nazaret Reverón-Gómez**: Investigation; Writing—review and editing. **Mikaela Koutrouli**: Investigation; Writing—review and editing. **Aldwin Suryo Rahmanto**: Investigation; Writing—review and editing. **Katrine Mitrofanov**: Investigation; Writing—review and editing. **Andreas Ingham**: Investigation; Writing—review and editing. **Sofie Schovsbo**: Investigation; Writing—review and editing. **Katrine Weischenfeldt**: Methodology; Writing—review and editing. **Fabian Coscia**: Methodology; Writing—review and editing. **Dimitris Typas**: Methodology; Writing—review and editing. **Moritz Völker-Albert**: Methodology; Writing—review and editing. **Victor Solis**: Methodology; Writing—review and editing. **Lars Juhl Jensen**: Supervision; Writing—review and editing. **Anja Groth**: Supervision; Writing—review and editing. **Andreas Mund**: Investigation; Writing—review and editing. **Petra Beli**: Supervision; Writing—review and editing. **Robert F Shearer**: Conceptualization; Supervision; Investigation; Methodology; Writing—original draft; Writing—review and editing. **Niels Mailand**: Conceptualization; Supervision; Funding acquisition; Writing—original draft; Project administration; Writing—review and editing.

Source data underlying figure panels in this paper may have individual authorship assigned. Where available, figure panel/source data authorship is listed in the following database record: biostudies:S-SCDT-10_1038-S44318-025-00599-7.

## Disclosure and competing interests statement

The authors declare no competing interests.

# Expanded View Figures

**Figure EV1.  Extended data related to Fig. 1.**

(**A**) Immunoblot (IB) analysis of U2OS/shUb and derivative U2OS/shUb/HA-Ub cell lines treated or not with DOX for the indicated times. (**B**) Immunoblot analysis of parental U2OS and derivative U2OS/shUb cell lines. (**C**) U2OS/shUb and derivative replacement cell lines were treated or not with DOX and mRNA levels were analyzed by RT-qPCR. Primers to GAPDH were used as a normalization control (data are technical duplicates of two independent experiments). (**D**) Immunoblot analysis of Ub replacement cell lines treated or not with the BRD4 PROTAC MZ1 (500 nM) overnight. (**E**) Immunoblot analysis of K6-linked Ub chains isolated by pulldown with the K6-Ub binder Biotin-LotA-N in the indicated DOX-treated Ub replacement cell lines exposed to UV-A (500 mJ/cm$^2$) and/or 4-thiouridine (S4U, 25 mM), combined treatment of which leads to RNA-protein crosslink formation. (**F**) Abundance of K6 diGly peptide after OtUBD (total Ub) enrichment of indicated DOX-treated Ub replacement cell lines in (**E**) (mean ± s.d.; $n = 2$ technical replicates). (**G**) Cell cycle analysis of indicated Ub replacement cell lines by QIBC. All Ub replacement cell lines were treated with DOX for 72 h, except Ub(K48R) cells that were treated for 48 h. Upper panel shows cell cycle fraction (mean ± s.e.m.; $n = 4$ independent experiments; >1000 cells analyzed per condition). Lower panel shows mitotic fraction as determined by QIBC of histone H3-pS10 immunostaining (mean ± s.e.m.; $n = 7$ independent experiments; >1000 cells analyzed per condition). Source data are available online for this figure.

▶

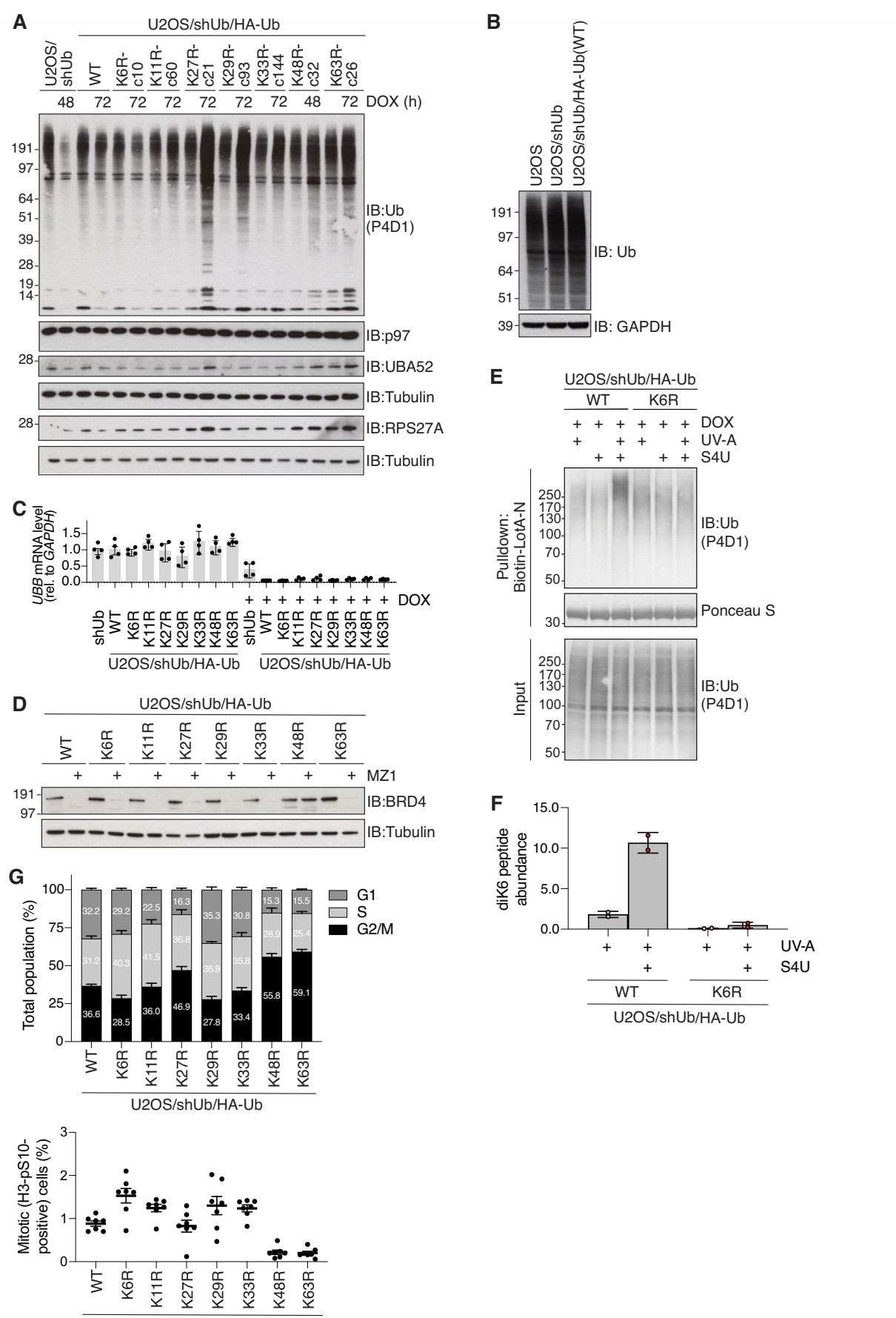

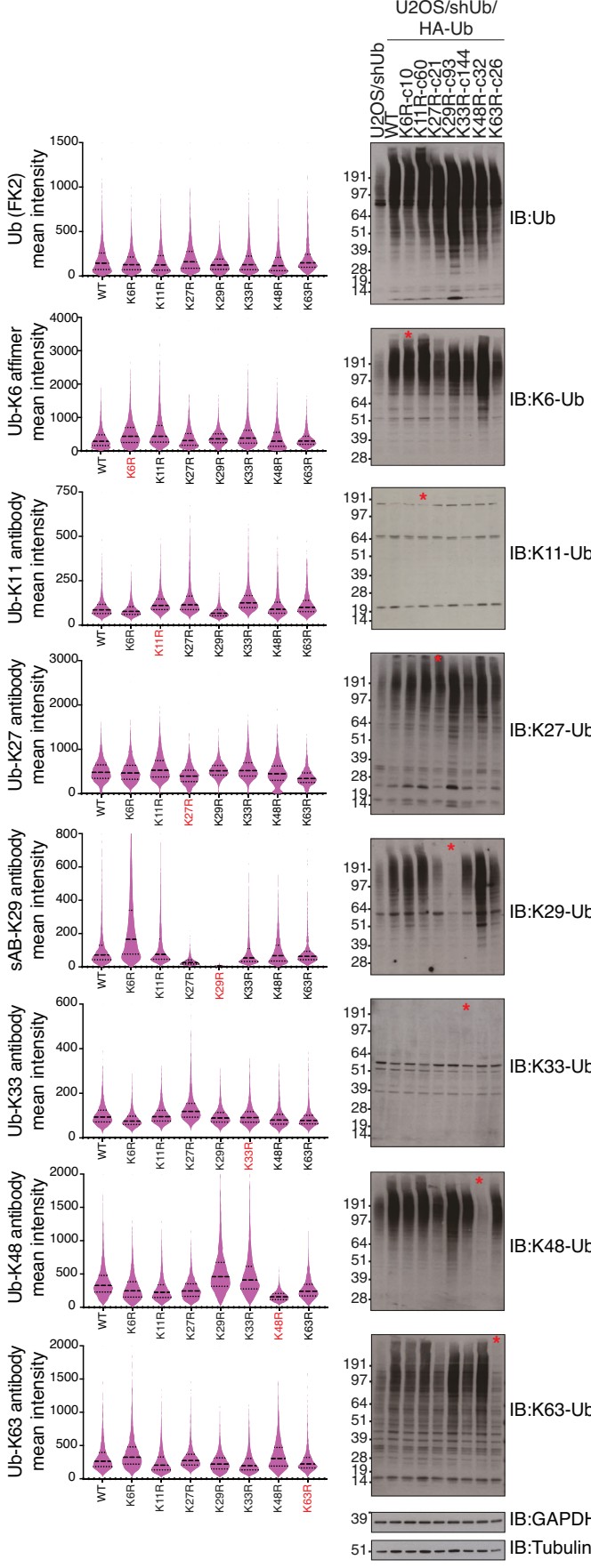

◀   **Figure EV2.   Extended data related to Fig. 2.**

QIBC analysis (left) and immunoblot analysis (right) of DOX-treated Ub replacement cell lines using the indicated linkage-specific antibodies or binders (thick dashed lines, median; dotted lines, quartiles). Data were a single representative replicate from three independent experiments. Source data are available online for this figure.

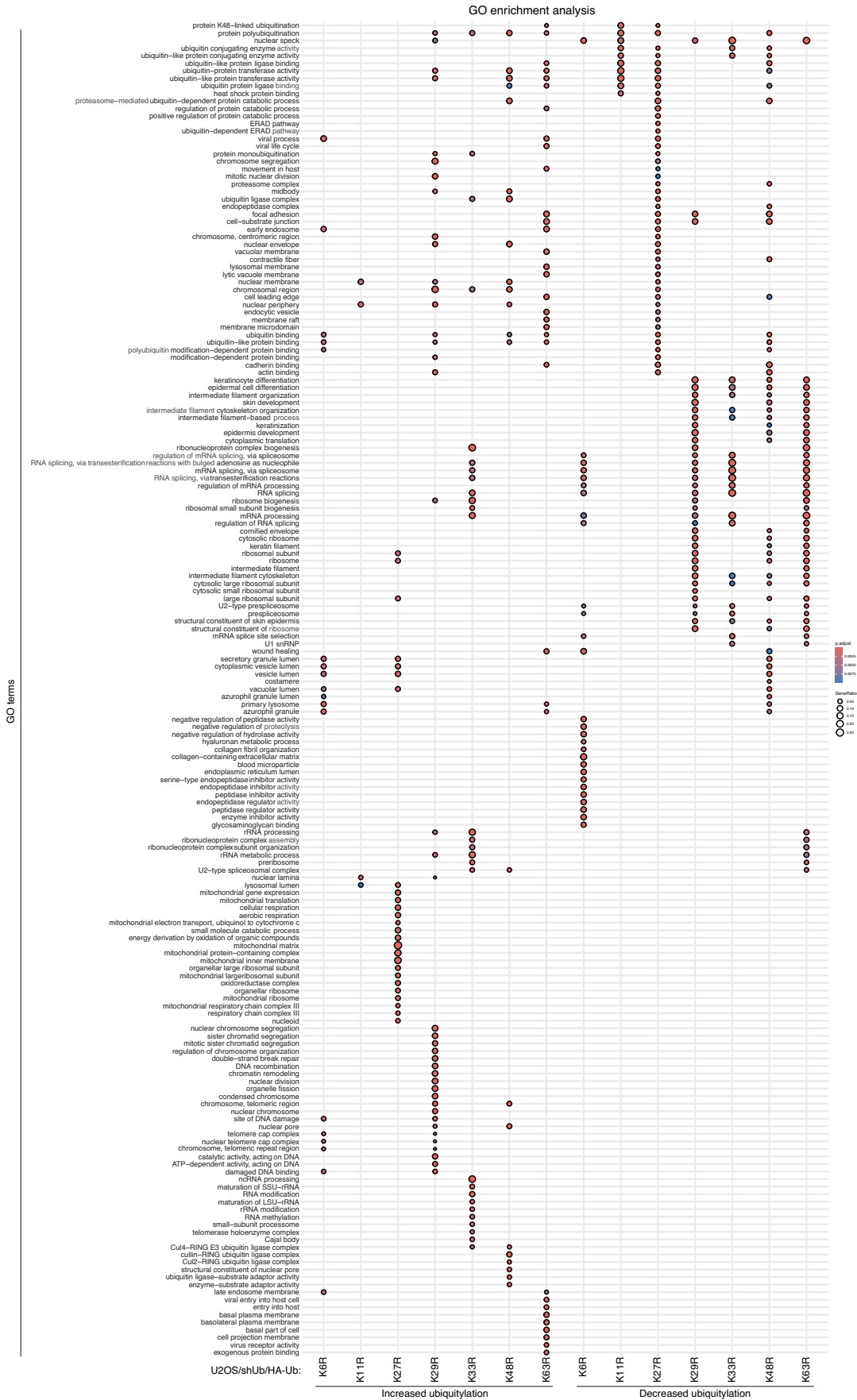

GO enrichment analysis

◄ **Figure EV3. Extended data related to Fig. 2.**

GO term analysis of cellular compartments enriched among proteins showing significantly up- or downregulated ubiquitylation in Ub(K-to-R)-replaced cells relative to Ub(WT)-replaced cells (Fig. 2B–H) ($n = 3$ technical replicates; unpaired two-tailed $t$-test, Benjamini–Hochberg corrected).

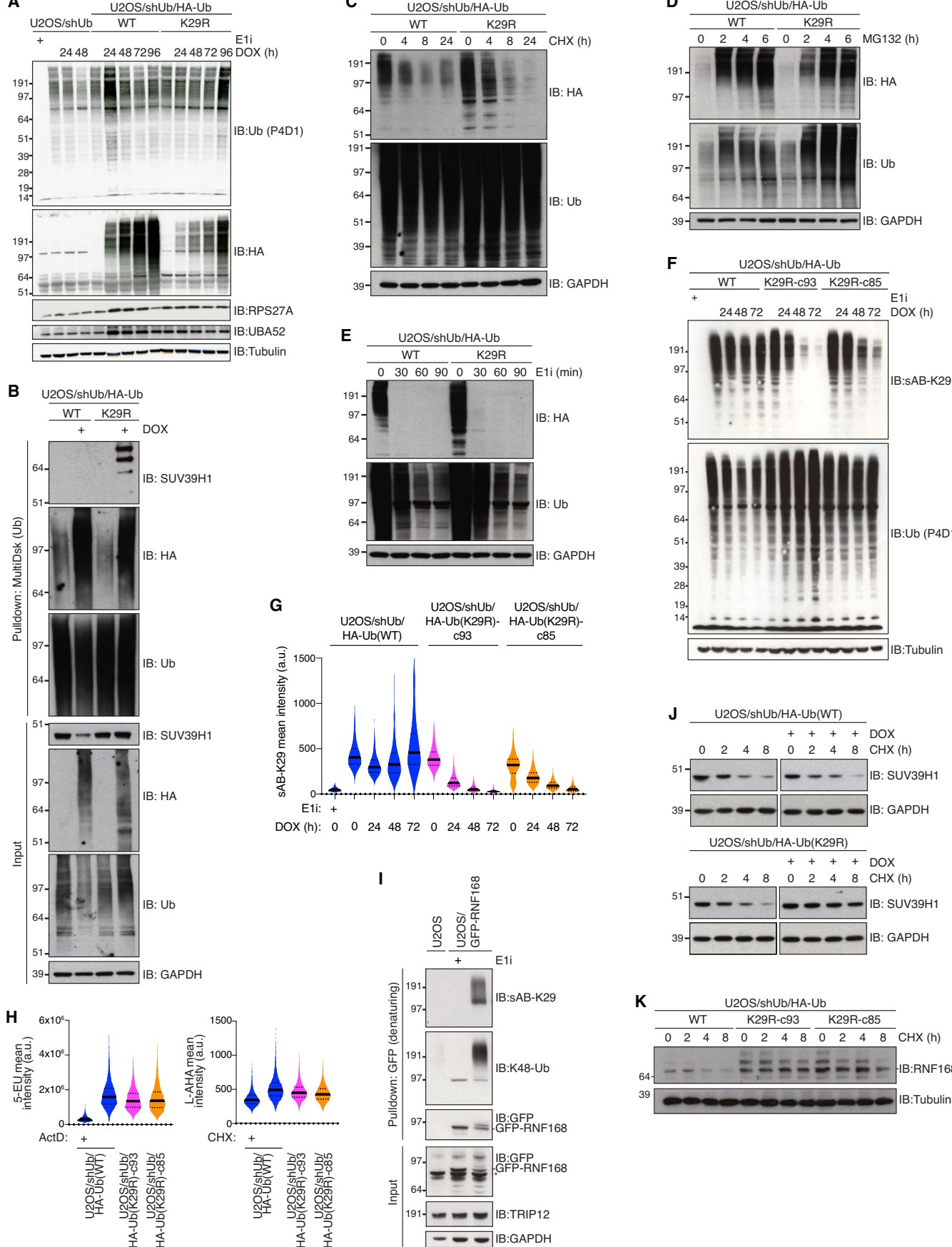

◀ **Figure EV4.  Extended data related to Fig. 3.**

(A) Immunoblot analysis of U2OS/shUb, U2OS/shUb/HA-Ub(WT) and U2OS/shUb/HA-Ub(K29R)-c93 cell lines treated or not with DOX for the indicated times. Where indicated, cells were treated with Ub E1 inhibitor (E1i) for 1 h. (B) Immunoblot analysis of Ub conjugates isolated via MultiDsk pulldown from indicated Ub replacement cell lines treated or not with DOX. (C) Immunoblot analysis of DOX-treated Ub(WT) and Ub(K29R) cell lines treated or not with cycloheximide (CHX) for the indicated times. (D) As in (C), except that cells were treated with MG132 for the indicated times. (E) As in (C), except that cells were treated with MLN-7243 (E1i) for the indicated times. (F) As in (A), but using Ub(WT), Ub(K29R)-c93 and Ub(K29R)-c85 replacement cell lines. (G) QIBC analysis of Ub(WT) and Ub(K29R) replacement cell lines treated with DOX for the indicated times and immunostained with sAB-K29 (thick dashed line, median; dotted lines, quartiles). Data were a single representative replicate from three independent experiments (>1000 cells analyzed per sample). (H) QIBC analysis of DOX-treated Ub(WT) and Ub(K29R) replacement cell lines treated or not with the transcription inhibitor Actinomycin D (ActD) or the protein synthesis inhibitor cycloheximide (CHX), where indicated and stained for nascent RNA (upper) or nascent protein (lower panel) by detection of incorporated EU and AHA, respectively (thick lines, median; dotted lines, quartiles). Data were a single representative replicate from three independent experiments. (I) U2OS cells or a derivative cell line stably expressing GFP-RNF168 were treated or not with Ub E1i, subjected to GFP pulldown under denaturing conditions and immunoblotted with the indicated antibodies. (J) Immunoblot analysis of Ub(WT) and Ub(K29R) cell lines incubated or not with DOX for 72 h and treated with cycloheximide (CHX) for the indicated times. (K) Immunoblot analysis of DOX-treated Ub(WT) and Ub(K29R) cell lines treated or not with cycloheximide (CHX) for the indicated times. Source data are available online for this figure.

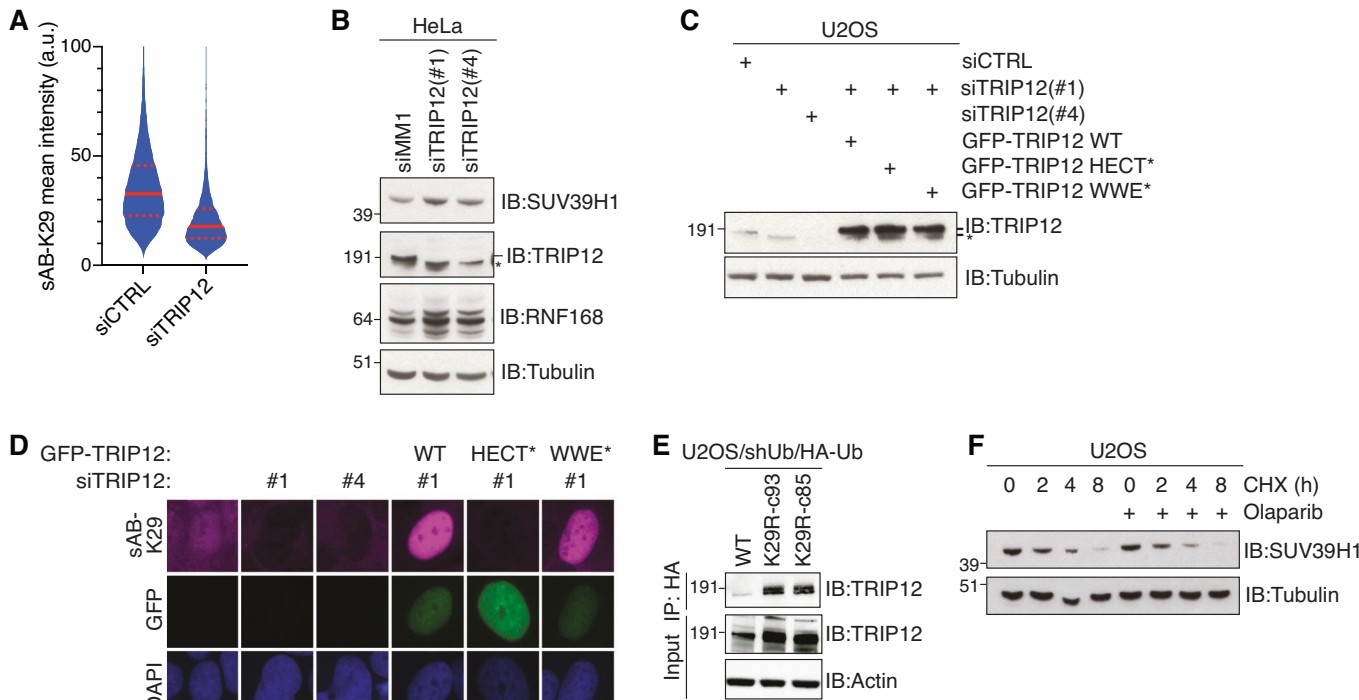

**Figure EV5.  Extended data related to Fig. 3.**

(A) QIBC analysis of HeLa cells transfected with indicated siRNAs (thick line, median; dotted lines, quartiles). Data were a single representative replicate from three independent experiments (>1000 cells analyzed per sample). (B) Immunoblot of HeLa cells transfected with indicated siRNAs. (C) Immunoblot analysis of U2OS cells sequentially transfected with indicated siRNA and siRNA-resistant GFP-TRIP12 expression plasmids. (D) Representative images of individual cells from data shown in Fig. 3J. Scale bar, 25 μm. (E) DOX-treated Ub(WT) and Ub(K29R) replacement cell lines were subjected to HA IP under denaturing conditions and immunoblotted with the indicated antibodies. (F) Immunoblot analysis of SUV39H1 in U2OS cells treated or not with cycloheximide (CHX) for the indicated times in the absence or presence of the PARP1 inhibitor Olaparib. Source data are available online for this figure.

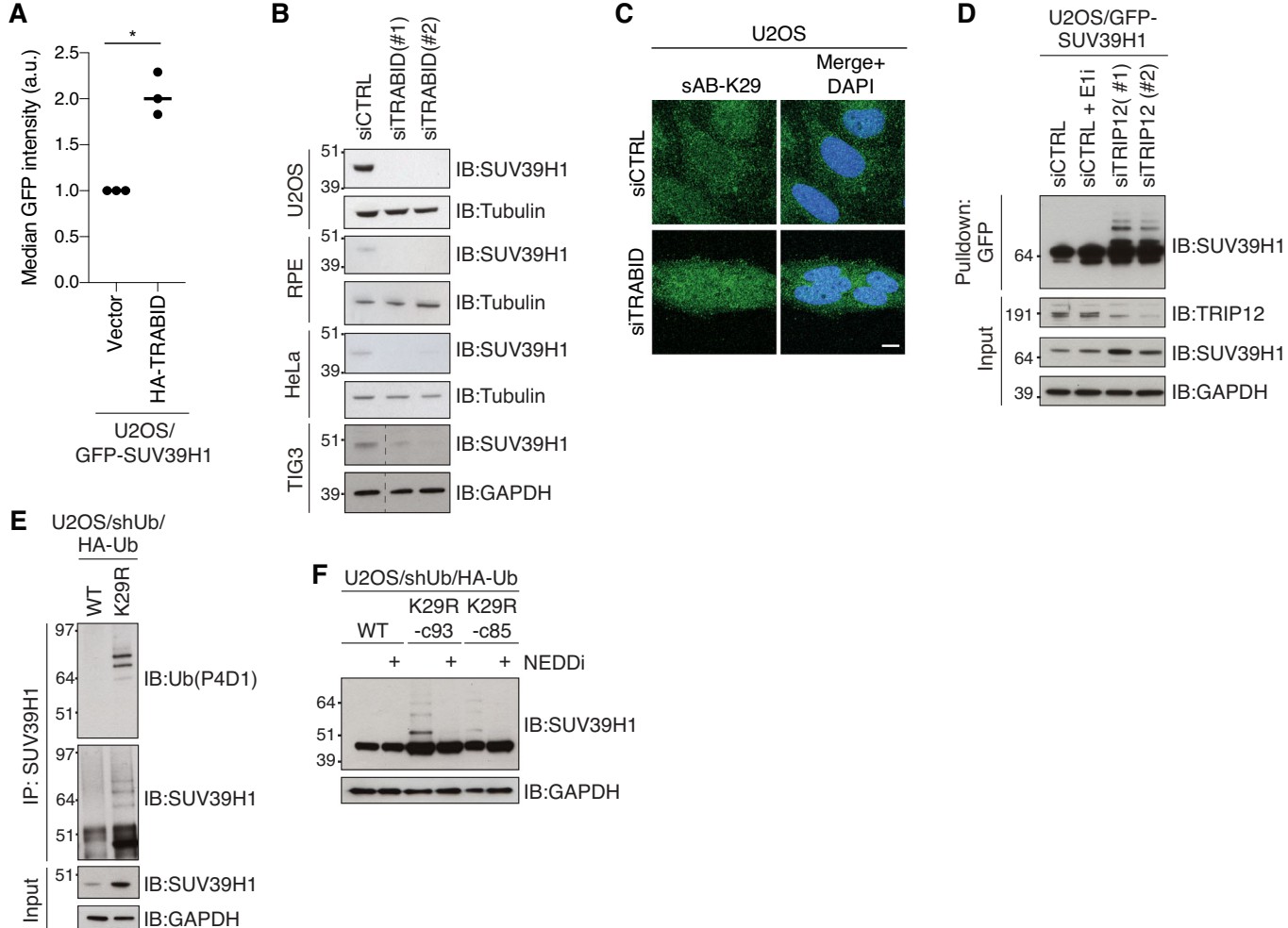

**Figure EV6. Extended data related to Fig. 4.**

(A) QIBC analysis of U2OS/GFP-SUV39H1 cells transfected or not with HA-TRABID expression construct. Data shown is fold change of mean of average nuclear GFP intensity relative to untransfected cells ($n = 3$ independent experiments; *$p < 0.05$, two-tailed $t$-test). (B) Immunoblot analysis of indicated cell lines transfected with non-targeting control (CTRL) or TRABID siRNAs. (C) Representative images of U2OS cells transfected with the indicated siRNAs and immunostained with sAB-K29. (D) U2OS/GFP-SUV39H1 cells transfected with siRNAs and treated with Ub E1 inhibitor, where indicated, were subjected to GFP pulldown under denaturing conditions and immunoblotted with indicated antibodies. (E) DOX-treated Ub(WT) and Ub(K29R) replacement cell lines were subjected to SUV39H1 IP and immunoblotted with indicated antibodies. (F) Immunoblot analysis of DOX-treated Ub(WT) and Ub(K29R) replacement cell lines treated or not with NEDDylation inhibitor for 4 h. Source data are available online for this figure.

**A**

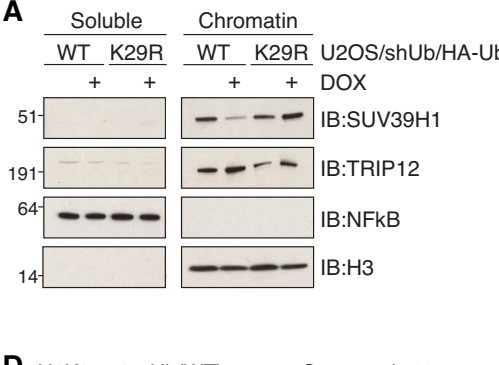

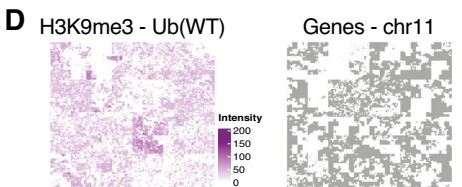

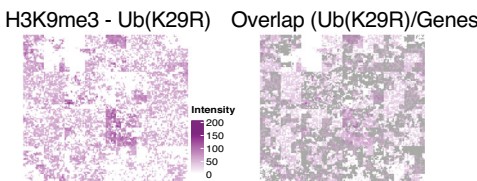

**D**

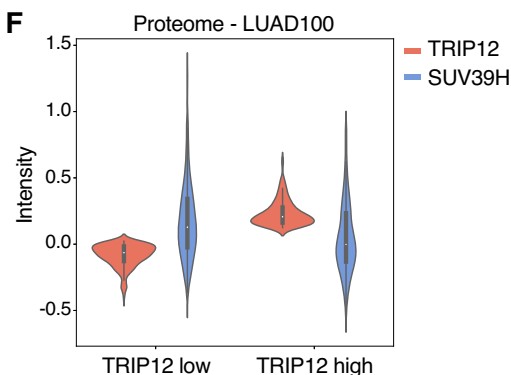

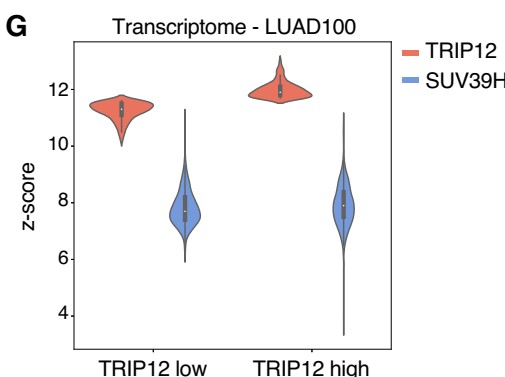

**F**

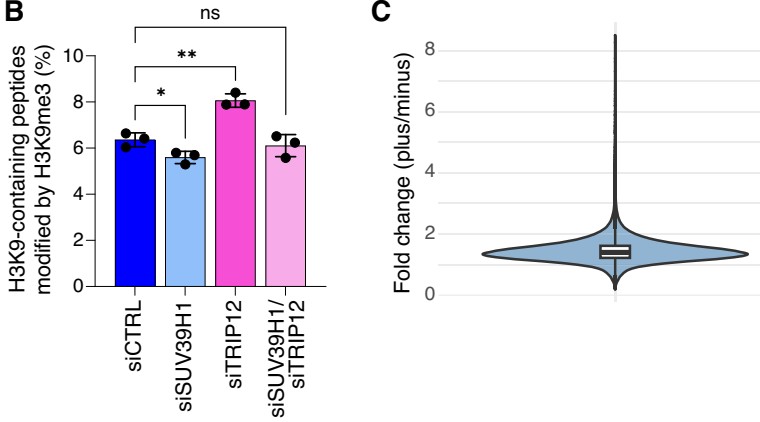

**B**

**C**

**E**

**G**

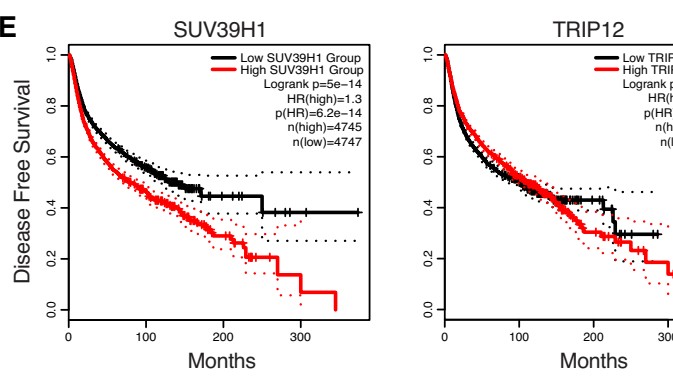

**H**

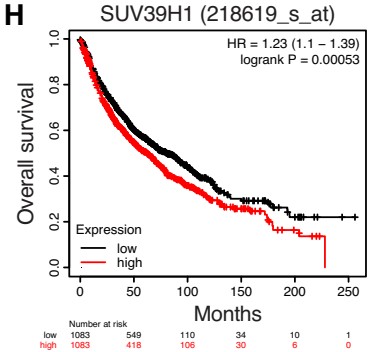

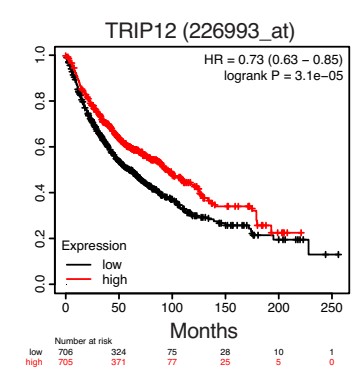

**Figure EV7.  Extended data related to Fig. 5.**

(A) Immunoblot analysis of soluble and chromatin-enriched fractions of Ub(WT) and Ub(K29R) replacement cell lines treated or not with DOX. (B) Quantitative proteomic analysis of H3K9me3 abundance (H3K9me3 peptides unmodified at K14) in U2OS cells treated with indicated siRNAs ($n = 3$ biological replicates; **$p < 0.01$; *$p < 0.05$; ns non-significant, unpaired two-tailed $t$-test). (C) Violin plot showing genome-wide distribution of H3K9me3 fold change values per 10 kb genomic bin, calculated from spike-in normalized H3K9me3 ChIP-seq signal in Ub(K29R)-replaced cells ($+$DOX) relative to non-replaced Ub(K29R) cells ($-$DOX). Each point represents a non-overlapping genomic bin, and fold changes were computed as the ratio of normalized signal between the two conditions. (D) Hilbert analysis depicting the spatial localization of H3K9me3 across chromosome 11 in DOX-treated Ub(WT) and Ub(K29R) replacement cell lines. Overlap panel depicts the signal from Ub(K29R) cells over genes on chromosome 11. (E) Kaplan–Meier analysis of disease-free survival stratified by expression of SUV39H1 (left) or TRIP12 (right) in a pan-cancer cohort. Dotted lines represent a 95% confidence interval. Analysis was performed using GEPIA2 on 33 cancer types. Hypothesis evaluation was performed with the log-rank test. Hazard ratios were calculated according to the Cox proportional-hazards model. (F) LUAD-100 proteome levels of indicated proteins stratified by relatively high and low TRIP12-expressing samples. Data were shown as the log2 ratio of reporter ion intensity. (G) LUAD-100 transcriptome levels stratified by relatively high and low TRIP12-expressing samples. (H) Kaplan–Meier analysis of overall survival in non-small cell lung cancer (NSCLC) stratified on SUV39H1 (left) or TRIP12 (right) expression. Analysis was performed using the KMPlotter tool. Hazard ratios and $p$ values were calculated through Cox regression. Source data are available online for this figure.

