## [Peer Review File · The EMBO Journal]

Functional landscape of ubiquitin linkages connects K29-linked ubiquitylation to epigenome integrity

Javier Arroyo-Gomez, Matthew Murray, Claire Guérillon, Juanjuan Wang, Ekaterina Isaakova, Nazaret Reveron-Gomez, Mikaela Koutrouli, Aldwin Suryo Rahmanto, Katrine Mitrofanov, Andreas Ingham, Sofie Schovsbo, Katrine Weischenfeldt, Fabian Coscia, Dimitris Typas, Moritz Völker-Albert, Victor Solis, Lars Jensen, Anja Groth, Andreas Mund, Petra Beli, Robert Shearer, and Niels Mailand

Corresponding author(s): Niels Mailand (niels.mailand@cpr.ku.dk) , Robert Shearer (r.shearer@unsw.edu.au)

Review Timeline:

Submission Date:	13th Dec 24
Editorial Decision:	31st Jan 25
Revision Received:	30th Jul 25
Editorial Decision:	16th Sep 25
Revision Received:	27th Sep 25
Accepted:	1st Oct 25

Editor: Hartmut Vodermaier

Transaction Report:

Prof. Niels Mailand
NNF Center for Protein Research, University of Copenhagen
Ubiquitin Signaling Group, Protein Signaling Program
Blegdamsvej 3B
Copenhagen DK-2200
Denmark

31st Jan 2025

Re: EMBOJ-2024-119909
Functional landscape of ubiquitin linkages couples K29-linked ubiquitylation to epigenome integrity

Dear Niels,

Thank you again for submitting your study on comprehensive ubiquitin replacement in cells, with particular emphasis on the roles of K29 linkages. After some delay caused by the busy period around the turn of the years, we have now received a complete set of reports from three expert referees, copied below for your information. As you will see, the reviewers all express interest in the work and appreciate its overall experimental quality. They nevertheless raise a number of presentational, experimental and conceptual issues that would need to be clarified prior to publication.

Should you be able to adequately address these points, we would be happy to consider a revised version further for The EMBO Journal. Since it may be helpful to discuss some of the noted concerns - including divergent referee views on restructuring/shifting emphasis of the paper, and conceptual queries such as those noted by referee 3 - and to agree on how to best incorporate them into a final manuscript, I would encourage you to contact me with a tentative response letter/revision plan already during the early stages of the revision work. Of course, we may also offer extension of the default three-months revision period if needed, with our 'scooping protection' (meaning that competing work appearing elsewhere in the meantime will not affect our considerations of your study) remaining valid during this time as well.

Detailed information on preparing, formatting and uploading a revised manuscript can be found below and in our Guide to Authors. Thank you again for the opportunity to consider this work for The EMBO Journal, and I look forward to hearing from you.

With kind regards,

Hartmut

9) To facilitate reproducibility and cross-laboratory adoption of methodologies, please structure the Materials & Methods section as outlined in our guide to authors, including a completed Reagents and Tools Table that can be downloaded from our author guidelines as well (<https://www.embopress.org/page/journal/14602075/authorguide#structuredmethods>).

10) Digital image enhancement is acceptable practice, as long as it accurately represents the original data and conforms to community standards. If a figure has been subjected to significant electronic manipulation, this must be clearly noted in the figure legend and/or the 'Materials and Methods' section. The editors reserve the right to request original versions of figures and the original images that were used to assemble the figure. Finally, we generally encourage uploading of numerical as well as gel/blot image source data; for details see: embopress.org/page/journal/14602075/authorguide#sourcedata

At EMBO Press, we ask authors to provide source data for the main manuscript figures. Our source data coordinator will contact you to discuss which figure panels we would need source data for and will also provide you with helpful tips on how to upload and organize the files.

In the interest of ensuring the conceptual advance provided by the work, we recommend submitting a revision within 3 months (1st May 2025). Please discuss the revision progress ahead of this time with the editor if you require more time to complete the revisions. Use the link below to submit your revision:

Link Not Available

Referee #1:

In the manuscript by Arroyo-Gomez, Niels Mailand and team, the authors describe a clever approach to reveal biological pathways linked to distinct Ub-linkage types. In particular, the pinpoint a biological role for the ubiquitin linkage K29 to be relevant in the context of epigenetic gene regulation. Specifically, the authors have explored a functional screen to explore what each ubiquitin linkage type might be involved in.

The authors report that K29-linked ubiquitin chain linkages are associated with chromosome associated biological processes involving the H3K9me3 methyltransferase SUV39H1 that appears to control this modification. K29-linked ubiquitylation catalyzed by TRIP12 and reversed by the deubiquitinase (DUB) TRABID modulates the degradation of SUV39H1. As a consequence, preventing K29-linked ubiquitylation-mediated control of SUV39H1 stability modulates the H3K9me3 landscape.

This is a manuscript that is technically very well put together at a high experimental standard. Also, novel K29-Ub linkage specific biology is described that merits sharing with the broader research community. However, in its current form, the

manuscript contains also a lot of biological information not only related to the K29-linked ubiquitylation dependent processes that is the main focus, but also on all other Ub-linkages, which somehow distracts the focus to the reported major findings highlighted in the abstract. Also, there are a number of issues that should be addressed by the authors before it can be recommended for publication as mentioned below.

Specific comments:

1. K48-, K63- and K27-linkages are instrumental for proliferation of human cells whereas K6-, K11-, K29- and K33-linked Ub polymers were found to be of lower critical importance for overall fitness of unperturbed cells. This is a relevant finding that is worth mentioning in the abstract.
2. The manuscript contains rich information about potential cellular pathways affected by distinct Ub-linkages. In its present form, this information is somewhat undervalued and may also merit to be mentioned in the abstract. Alternatively, all this data would clearly fill several other potential publications and could be trimmed down to a minimum to allow emphasis on K29-linked biological findings (see comment above).
3. For K29-linked Ub chain related cellular pathways, Fig 2M indicates links to nuclear processes. How did the authors pick out the particular proteins TRIP12, SUV39H1, RNF168 etc to be investigated further for K29-linked ubiquitination as described in Figure 3?
The authors should provide a clearer rationale and narrative that logically leads to the selection and further investigation of these particular targets.
4. Figure 5 shows effects on gene regions targeted by the H3K9 methylation status modulated by methyltransferase SUV39H1 - have you checked whether SUV39H1 incapacitation affects other histone marks more broadly?

Referee #2:

In this manuscript Arroyo-Gómez et al. use their previously developed system to replace endogenous ubiquitin to study the effects of blocking chain formation in the 7 internal lysines in ubiquitin. The authors extend their work and provide a very extensive resource on the proteins that are targeted by specific ubiquitin chains. Then the authors focus on SUV39H1 as a novel target of K29 ubiquitin chains, identifying E3 ubiquitin ligases and deubiquitylases that deposit and remove K29 chains in general and modify SUV39H1 in particular. They also provide an initial assessment of the relevance of SUV39H1 ubiquitylation by K29 chains in the control of its stability and the deposition of H3K9me3. They also make a correlative analysis of the expression of TRIP12 and SUV39H1 which is not very strong.

The work is very well designed, clearly presented and the experiments are well controlled and solid. The generation of these systems is a powerful tool and the resource provided in the manuscript constitute a very relevant advance in the field. The analysis of the ubiquitylation of SUV39H1 is thorough but raises some questions as there are some contradictory results. I think this is a solid and relevant work for publication but I would like the authors to address some questions:

1. Since the ubiquitin replacement is not complete (Figure S1A) it is important to address if the replacement with the different mutants is similar, understood as the ratio of HA-Ub versus remaining Ub. This could be addressed by mass spectrometry. Alternatively, the authors can follow the levels of total wild type ubiquitin and HA-ubiquitin after the treatment of cell extracts with USP2cc.
2. The data in Figure 2I shows that the replacement mutants also affect mono-ubiquitylation of substrates which is unexpected given that the mutants should only affect chain formation. These results imply that the K to R mutations might affect the recognition of ubiquitin by specific E2 and E3 enzymes or the activity of these enzymes with specific substrates. This is a potential limitation of the system and it should be discussed in the manuscript.
3. The analysis of SUV39H1 ubiquitylation makes use of GFP and FLAG tagged versions of the protein. However, these two tags seem to yield different results. Figure 3B shows that inducing the replacement with K29R reduces the ubiquitylation and the total levels of GFP-SUV39H1. Figure 3A and S3E show that the replacement with K29R increases the ubiquitylation and levels of endogenous or FLAG-SUV39H1. If possible, I would stick to the FLAG tagged version of SUV39H1 since it seems to better reflect the behavior of the endogenous protein. At least, it would be necessary to compare the levels of overexpression obtained with the GFP and FLAG tags to show that they are comparable.
4. Many of the experiments compare WT and K29R replacement (Figures 3A, 3D, 4E, S3E, S3F, S3L, S5E, S5F). Surprisingly, Figure 5B and S6A show that WT replacement leads to a reduction in the levels of SUV39H1 and an increase in TRIP12. Thus, the difference in ubiquitylation and stability of SUV39H1 between WT and K29R replaced cells could be the result of a lower stability of SUV39H1 upon WT replacement. Non replaced WT cells should be included as a control to confirm that the results are not due to an indirect effect of WT replacement.
5. The authors claim that K48 chains on SUV39H1 require K29 modification but Figure S4E shows an increase in K48 chains in K29R replaced cells. This could again be related to the effect of WT replacement and the authors need to address this

contradictory result.

6. The analysis of the control of SUV39H1 is quite preliminary. I understand that it is not the main focus of the work but some of the basic questions should be addressed better. It is not very clear why there is no reduction in H3K9me3 when the levels of SUV39H1 are reduced in WT replaced cells. The ChIP experiments need a deeper analysis. Is the increase in H3K9me3 similar in all the regions of the genome? What are the changes in pericentric heterochromatin, the region where H3K9me3 is mostly enriched?

7. The authors show that the K29R replacement changes H3K9me3 levels on chromatin. Further experiments are required to show that the changes in SUV39H1 are responsible for the alterations in H3K9me3. For instance, ChIP for SUV39H1 upon K29R replacement or the effects of TRIP12 depletion and/or TRABID overexpression on H3K9me3 and SUV39H1 on chromatin.

8. The analysis of the involvement of TRIP12 and SUV39H1 in cancer is very weak. As the authors point out the levels of mRNA are not very useful to address if the control of SUV39H1 stability by TRIP12 is relevant in cancer. The effect in Figure 5E for TRIP12 is very minor. The inverse correlation of TRIP12 and SUV39H1 protein levels in LUAD offers a hint of a potential coordinated regulation. I feel this last part does not offer relevant information and a more in-depth analysis of the control of H3K9me3 by the ubiquitylation of SUV39H1 would be more relevant for the manuscript.

Minor comment: the first paragraph in page 9 is a bit confusing since it is not easy to follow when the authors are talking about the whole proteome or the analysis of ubiquitylated proteins.

Referee #3:

In this manuscript, the authors utilise a cell-based ubiquitin replacement strategy to supplant endogenous ubiquitin with exogenous mutant proteins unable to form each of the seven lysine-based ubiquitin linkages in human cells. The authors use this system to profile the cellular impacts of abrogating individual polyubiquitin chain types and use their panel of cell-lines to assess the performance and specificity of commercially available ubiquitin linkage-selective affinity reagents. The authors choose to focus on K29-linked ubiquitination, showing that it is strongly associated with chromosome biology and that the histone methyltransferase SUV39H1 is a prominent substrate of proteasomal degradation mediated by K29-linked ubiquitination. Using an siRNA screen, the authors identify TRIP12 as an E3 ubiquitin ligase responsible for K29-linkage-dependent ubiquitination and degradation of SUV39H1. These results also implicate a role for Cullin-RING ubiquitin ligase activity in initiating ubiquitination of SUV39H1 to prime its subsequent TRIP12-dependent K29-Ub modification, as well as branching TRIP12-catalysed K29-linked ubiquitination with K48-linked ubiquitin chains. Abrogation of K29-linked ubiquitination stabilises SUV39H1 and deregulates the histone H3Kme3 landscape, while dysregulation of the SUV39H1 K29-Ub writer/reader machinery is associated with poorer prognosis in multiple cancer types.

This is a well-executed and comprehensive study, identifying a K29-Ub modified substrate, a responsible ligase, and probable deubiquitylase. The creation, validation, and characterisation of all seven Ub replacement cell lines is impressive, as is the system-wide proteomic profiling of ubiquitination changes within those cells. The authors do a good job of establishing SUV39H1 as a substrate of K29-linked ubiquitination, identifying TRIP12 as a key ligase catalysing this modification in cells and whose activity is required for SUV39H1's K29-linked ubiquitin dependent turnover. This is demonstrated using both the K29R ubiquitin replacement cells and in wild-type cells looking at the endogenous proteins. Perhaps slightly weaker are the experiments implicating TRABID as a DUB for ubiquitylated SUV39H1, but the authors do demonstrate a destabilisation of endogenous SUV39H1 following TRABID depletion in multiple cell lines.

My main concerns with the manuscript are the following:

It is unclear whether the levels of Ub in the replacement cells are the same as the levels in a "normal" cell, and whether the introduced mutations can be efficiently processed by the endogenous cellular ubiquitination and deubiquitination machinery. For instance, mutation of K33 to R blocks the ability of OTULIN to cleave linear chains containing this mutation. Hence it cannot be ruled out that the observed effects are not on account of a particular linkage type being made by the K-R substitution but due to some other effect like change in the kinetics of ubiquitination. Similarly, K27 faces into the hydrophobic core of ubiquitin and is not accessible. So the mutation K27R may affect the stability of Ub leading to altered dynamics. While it may be beyond the scope to test all of these possibilities, at a minimum these must be addressed for K29R, the focus of this manuscript and the rest stated as limitations of the approach.

On page 10, the sentence "..... while disrupting K27-linked ubiquitylation was specifically associated with enhanced ubiquitylation of mitochondrial proteins (Figure 2J-L; Figure S3)." Looking at the 2022 publication on K27 chains, this association of K27 linkage to mitochondrial proteins was not mentioned. Could the authors please comment on the reproducibility of the observations of the previous study?

The authors have tried to dissect the topology of the chains on SUV39H1 and suggest that it is first priming by CRLs, followed by K29 extension by TRIP12 and then extension/branching with K48 linkages again in a CRL-dependent manner. One possibility that the authors have not considered is that it is neddylation-dependent chain initiation (i.e. chain priming). The authors show that absolutely no Ub chains are present on SUV39H1 (and that SUV39H1 protein levels are stabilised during

CHX chase in cells) when neddylation is inhibited (Figs 4G, 4H & S5F) even though UbiCRest analysis suggests that CRL-dependent K48-linked ubiquitination of SUV39H1 mostly occurs distal to K29-linked ubiquitination (Fig 4F). Histone neddylation has previously been reported, although reports of non-cullin neddylation are admittedly a contentious issue and reports often rely heavily upon exogenous overexpression of NEDD8 protein. That said, RNF168, another proteolytic substrate of TRIP12, has been suggested to be a NEDD8 ligase that neddylates histones to regulate DNA damage repair and is itself reported to be a substrate for neddylation (Li et al. 2014, J Cell Sci 127, 2238-48). The authors' data is consistent with a model where the K29 Ub chains are attached to a priming neddylation of the SUV39H1 substrate rather than a priming ubiquitin. This is easily testable by blotting SUV39H1 pulldowns for NEDD8.

Minor points:

1. Figure 3A contains two GAPDH panels, one of which has an asterisk next to it. The NDC80 blot also has an asterisk next to its label. I assume that, like the asterisked text in Figure 4C, these samples were run on a separate gel. However, the figure legend does not mention this and the labels are not explained. The authors should update the legend to explain the significance of the asterisks.
2. In Figure S1A, it is unclear if the levels of replaced ubiquitin are the same as parental cells. Please show comparison with parental U2OS cells. Also incubation of cell lysates with a promiscuous DUB like USP2 will reveal if the ubiquitin levels are the same.
3. RNF168 is mentioned but not labelled in the plot shown in Figure 2E.
4. In Figure S4A, U2OS/shUb, U2OS/shUb WT and U2OS/shUb K29R at 24 h show different ubiquitylation levels. Furthermore, clone C93 seem to have a higher ubiquitylation signal than WT and clone C85. It is not clear if the main conclusions of the study were drawn from analyses of one or multiple clones. Please specify.

Point-by-point response to the referees' comments

We thank the referees for their insightful and constructive feedback on our study. In the revised version of our manuscript, we included the results of a range of new experiments performed on the basis of the reviewers' helpful suggestions, which we believe address all of their concerns. In the following, we provide a detailed point-by-point response to the referee reports. The reviewers' comments (reproduced in their entirety) are shown in bold and italicized text, while our responses are in plain text.

Referee #1:

In the manuscript by Arroyo-Gomez, Niels Mailand and team, the authors describe a clever approach to reveal biological pathways linked to distinct Ub-linkage types. In particular, the pinpoint a biological role for the ubiquitin linkage K29 to be relevant in the context of epigenetic gene regulation. Specifically, the authors have explored a functional screen to explore what each ubiquitin linkage type might be involved in.

The authors report that K29-linked ubiquitin chain linkages are associated with chromosome associated biological processes involving the H3K9me3 methyltransferase SUV39H1 that appears to control this modification. K29-linked ubiquitylation catalyzed by TRIP12 and reversed by the deubiquitinase (DUB) TRABID modulates the degradation of SUV39H1. As a consequence, preventing K29-linked ubiquitylation-mediated control of SUV39H1 stability modulates the H3K9me3 landscape.

This is a manuscript that is technically very well put together at a high experimental standard. Also, novel K29-Ub linkage specific biology is described that merits sharing with the broader research community. However, in its current form, the manuscript contains also a lot of biological information not only related to the K29-linked ubiquitylation dependent processes that is the main focus, but also on all other Ub-linkages, which somehow distracts the focus to the reported major findings highlighted in the abstract. Also, there are a number of issues that should be addressed by the authors before it can be recommended for publication as mentioned below.

Specific comments:

1. K48-, K63- and K27-linkages are instrumental for proliferation of human cells whereas K6-, K11-, K29- and K33-linked Ub polymers were found to be of lower critical importance for overall fitness of unperturbed cells. This is a relevant finding that is worth mentioning in the abstract.

We agree and have modified the abstract accordingly.

2. The manuscript contains rich information about potential cellular pathways affected by distinct Ub-linkages. In its present form, this information is somewhat undervalued and may also merit to be mentioned in the abstract. Alternatively, all this data would clearly fill several other potential publications and could be trimmed down to a minimum to allow emphasis on K29-linked biological findings (see comment above).

These are fair and constructive considerations. We opted to keep the proteomic analyses of global ubiquitylation changes resulting from abrogation of all individual Ub linkages in the manuscript, as they provide unique novel insights into cellular pathways impacted by linkage-specific Ub chains. We believe that this is a valuable data resource and that disseminating it to the research community holds promise to empower important advances in the functional understanding of linkage-specific ubiquitylation processes, particularly for low abundance Ub linkages whose functions remain technically challenging to decipher. At the same time, by revealing a strong connection between K29-linked ubiquitylation and chromosome-associated processes that clearly stands out among all lysine-based Ub chain types, we believe these proteomic analyses add weight to our discovery of an important role of K29-linked Ub chains in maintaining epigenome integrity via controlling SUV39H1 stability.

3. For K29-linked Ub chain related cellular pathways, Fig 2M indicates links to nuclear processes. How did the authors pick out the particular proteins TRIP12, SUV39H1, RNF168 etc to be investigated further for K29-linked ubiquitination as described in Figure 3? The authors should provide a clearer rationale and narrative that logically leads to the selection and further investigation of these particular targets.

As mentioned above, our GO term analysis of proteins whose ubiquitylation status is impacted by K29-linked ubiquitylation revealed a striking enrichment of chromosome-associated processes that was uniquely observed for this Ub linkage type (Figure 2M; Figure EV3). Factors involved in chromosome-associated processes were thus a natural focus of our targeted studies of regulatory roles of K29-linked ubiquitylation. SUV39H1, RNF168 and NDC80 all have crucial roles in different aspects of preserving chromosomal integrity, while TRIP12 is a known K29-Ub-catalyzing E3 ligase targeting several chromosome-associated proteins incl. RNF168, PARP1 and BRD4, hence we focused on these factors among the proteins whose ubiquitylation status is most significantly impacted by abrogating K29-linked ubiquitylation according to our proteomic profiling. We have amended the text to provide a more elaborate account of our rationale for focusing on these K29-Ub-regulated proteins (page 10-11).

4. Figure 5 shows effects on gene regions targeted by the H3K9 methylation status modulated by methyltransferase SUV39H1 - have you checked whether SUV39H1 incapacitation affects other histone marks more broadly?

Yes – these data were already included in the manuscript as a supplemental table (Dataset EV5) but might have been missed by the Reviewer. To this end, we performed quantitative mass spectrometry-based analysis of the impact of Ub(WT) or Ub(K29R) replacement on more than 80 different histone marks. This showed that Ub(K29R) replacement, but not Ub(WT) replacement, leads to a marked increase in H3K9me3 levels, consistent with the stabilization of SUV39H1 upon abrogation of K29-linked ubiquitylation (Figure 5A). Remarkably, this proteomic analysis additionally revealed that no non-H3K9 histone marks were significantly affected by Ub(K29R) replacement (Dataset EV5), underscoring a particular importance of K29-linked ubiquitylation in maintaining H3K9 methylation status among epigenetic marks.

Referee #2:

In this manuscript Arroyo-Gómez et al. use their previously developed system to replace endogenous ubiquitin to study the effects of blocking chain formation in the 7 internal lysines in ubiquitin. The authors extend their work and provide a very extensive resource on the proteins that are targeted by specific ubiquitin chains. Then the authors focus on SUV39H1 as a novel target of K29 ubiquitin chains, identifying E3 ubiquitin ligases and deubiquitylases that deposit and remove K29 chains in general and modify SUV39H1 in particular. They also provide an initial assessment of the relevance of SUV39H1 ubiquitylation by K29 chains in the control of its stability and the deposition of H3K9me3. They also make a correlative analysis of the expression of TRIP12 and SUV39H1 which is not very strong.

The work is very well designed, clearly presented and the experiments are well controlled and solid. The generation of these systems is a powerful tool and the resource provided in the manuscript constitute a very relevant advance in the field. The analysis of the ubiquitylation of SUV39H1 is thorough but raises some questions as there are some contradictory results. I think this is a solid and relevant work for publication but I would like the authors to address some questions:

1. Since the ubiquitin replacement is not complete (Figure S1A) it is important to address if the replacement with the different mutants is similar, understood as the ratio of HA-Ub versus remaining Ub. This could be addressed by mass spectrometry. Alternatively, the authors can follow the levels of total wild type ubiquitin and HA-ubiquitin after the treatment of cell extracts with USP2cc.

We performed quantitative proteomic analysis of diGly-modified Ub peptides to quantify the extent of disrupting Ub linkage formation and thus the efficiency of Ub replacement across our panel of Ub replacement cell lines. Importantly, this directly validated strong reduction of the linkage type directly impacted by the K-to-R mutation in the Ub(K6R), Ub(K11R), Ub(K27R), Ub(K33R), Ub(K48R) and Ub(K63R) replacement cell lines, to an extent that was broadly comparable across this panel of cell lines (new Figure 1H). Due to technical limitations K29 linkages could not be consistently quantified by this proteomic approach, as also observed in our previous study (Shearer *et al*, 2022). Instead, we used the highly specific K29-linkage binder sAB-K29 (Figure EV2) (Yu *et al*, 2021) to verify quantitative suppression of K29-linked ubiquitylation in Ub(K29R) cell lines (new Figure 1I).

2. The data in Figure 2I shows that the replacement mutants also affect mono-ubiquitylation of substrates which is unexpected given that the mutants should only affect chain formation. These results imply that the K to R mutations might affect the recognition of ubiquitin by specific E2 and E3 enzymes or the activity of these enzymes with specific substrates. This is a potential limitation of the system and it should be discussed in the manuscript.

We agree that this is an important consideration, which echoes Reviewer #3's point 1. In the revised manuscript, we now point out and discuss potential limitations of the Ub replacement approach (page 21-22).

3. The analysis of SUV39H1 ubiquitylation makes use of GFP and FLAG tagged versions of the protein. However, these two tags seem to yield different results. Figure 3B shows that

inducing the replacement with K29R reduces the ubiquitylation and the total levels of GFP-SUV39H1. Figure 3A and S3E show that the replacement with K29R increases the ubiquitylation and levels of endogenous or FLAG-SUV39H1. If possible, I would stick to the FLAG tagged version of SUV39H1 since it seems to better reflect the behavior of the endogenous protein. At least, it would be necessary to compare the levels of overexpression obtained with the GFP and FLAG tags to show that they are comparable.

This is a valid point. We concur that stabilization of GFP-SUV39H1 upon Ub(K29R) replacement was difficult to appreciate in Figure 3B in the original manuscript, likely due to a high level of GFP-SUV39H1 overexpression in this experiment. Indeed, the stabilization of SUV39H1 upon Ub(K29R) replacement was much more pronounced for FLAG-SUV39H1 (Figure S4E in the original manuscript), recapitulating the behavior of endogenous SUV39H1 and showing additionally that preventing K29-linked ubiquitylation increases SUV39H1 modification by K48 linkages. Accordingly, we removed the experiment using GFP-SUV39H1 from the revised study and now show the data on FLAG-SUV39H1 in the main figure (now Figure 3B), in line with the Reviewer's helpful suggestion.

4. Many of the experiments compare WT and K29R replacement (Figures 3A, 3D, 4E, S3E, S3F, S3L, S5E, S5F). Surprisingly, Figure 5B and S6A show that WT replacement leads to a reduction in the levels of SUV39H1 and an increase in TRIP12. Thus, the difference in ubiquitylation and stability of SUV39H1 between WT and K29R replaced cells could be the result of a lower stability of SUV39H1 upon WT replacement. Non replaced WT cells should be included as a control to confirm that the results are not due to an indirect effect of WT replacement.

We occasionally observe a moderate decrease in the level of SUV39H1 upon Ub(WT) replacement (e.g. Figure 5B), which might be due to the increased total level of Ub in Ub(WT)-replaced cells relative to non-replaced cells (Figure 1F). Importantly, however, we found that the observed half-life of SUV39H1 is similar in Ub(WT)-replaced cells and parental U2OS cells (compare Figure 3C and Figure 3E). Accordingly, considering the complete stabilization of SUV39H1 upon Ub(K29R) replacement (Figure 3D), the striking difference in SUV39H1 ubiquitylation and stability between Ub(WT)- and Ub(K29R)-replaced cells seems unlikely to reflect increased turnover of SUV39H1 in Ub(WT)-replaced cells, also considering the similar levels of ectopic Ub expression in Ub(WT)- and Ub(K29R)-replaced cells (Figure 1E-G). However, to more directly demonstrate this, we followed the Reviewer's helpful suggestion and performed new experiments comparing the stability and ubiquitylation status of SUV39H1 in replaced and non-replaced Ub(WT) and Ub(K29R) cells. We found that the half-life of SUV39H1 is very similar in non-replaced Ub(WT) and Ub(K29R) cells (new Figure EV4J). Importantly, consistent with the above findings, Ub(K29R) replacement led to full stabilization of SUV39H1 whereas Ub(WT) replacement had little effect (new Figure EV4J). These observations were paralleled by the ubiquitylation level of SUV39H1, which was very low in both non-replaced Ub(WT) and Ub(K29R) cells and increased substantially upon Ub(K29R) but not Ub(WT) replacement (new Figure EV4B). Altogether, these findings strongly suggest that the difference in SUV39H1 stability and ubiquitylation between Ub(WT)- and Ub(K29R)-replaced cells is a direct consequence of disrupting K29-linked ubiquitylation of SUV39H1 in the latter cells and does not reflect accelerated degradation of SUV39H1 upon Ub(WT) replacement.

5. The authors claim that K48 chains on SUV39H1 require K29 modification but Figure S4E shows an increase in K48 chains in K29R replaced cells. This could again be related

to the effect of WT replacement and the authors need to address this contradictory result.

We think this is a misunderstanding of our model, as we do not claim that K48-linked ubiquitylation of SUV39H1 requires its prior modification with K29-linked Ub chains. Using linkage-specific DUBs we find that most K48-linked Ub conjugates on SUV39H1 occur distal to K29-linked Ub linkages on SUV39H1 when cellular Ub status is unperturbed (Figure 4F). However, we also show that SUV39H1 remains permissive for K48-linked ubiquitylation when K29-linked ubiquitylation is abrogated (Figure 3B). Thus, when K29-linked Ub chains cannot be formed, SUV39H1 still undergoes K48-linked ubiquitylation, but these modifications are not sufficient to drive proteasomal degradation of SUV39H1, leading to accumulation of K48-linked Ub conjugates on SUV39H1. For clarity, these findings are summarized in the schematic model in Figure 4J.

6. The analysis of the control of SUV39H1 is quite preliminary. I understand that it is not the main focus of the work but some of the basic questions should be addressed better. It is not very clear why there is no reduction in H3K9me3 when the levels of SUV39H1 are reduced in WT replaced cells. The ChIP experiments need a deeper analysis. Is the increase in H3K9me3 similar in all the regions of the genome? What are the changes in pericentric heterochromatin, the region where H3K9me3 is mostly enriched?

As mentioned above (point 4), we occasionally observe a moderate decrease in SUV39H1 expression in Ub(WT)-replaced cells, however our quantitative mass spectrometry data shows that this is not sufficient to significantly reduce total H3K9me3 abundance (Figure 5A; Dataset EV5), possibly because the expression level of SUV39H2 remains unchanged under these conditions (Figure 5A). This suggests that epigenome integrity may be more sensitive to defective control of SUV39H1 turnover than moderate reductions in SUV39H1 expression.

In the revised manuscript, we provide further analysis of the H3K9me3 ChIP-seq data, showing that Ub(K29R) replacement leads to a striking global increase in H3K9me3 signal that is highly uniform across the genome and not confined to regions such as pericentric heterochromatin where H3K9me3 is most enriched (new Figure 5E; new Figure EV7C). These data further underscore the importance of controlling SUV39H1 stability via K29-linked ubiquitylation for maintaining H3K9me3 homeostasis. See also our response to the next point.

7. The authors show that the K29R replacement changes H3K9me3 levels on chromatin. Further experiments are required to show that the changes in SUV39H1 are responsible for the alterations in H3K9me3. For instance, ChIP for SUV39H1 upon K29R replacement or the effects of TRIP12 depletion and/or TRABID overexpression on H3K9me3 and SUV39H1 on chromatin.

We thank the Reviewer for the helpful suggestions. Despite careful efforts, the SUV39H1 antibodies we tried unfortunately were not sufficiently specific to allow reliable genome-wide mapping of SUV39H1 occupancy by ChIP-seq or CUT&RUN. Instead, we used a quantitative immunofluorescence approach to show that the marked increase in global H3K9me3 abundance induced by Ub(K29R) replacement that we also observed in proteomic and ChIP-seq analyses was substantially attenuated by SUV39H1 knockdown (new Figure 5C), suggesting that stabilization of SUV39H1 is the main determinant underlying increased H3K9me3 levels upon disruption of K29-linked ubiquitylation. Moreover, we used mass spectrometry-based quantification of histone modifications to show that knockdown of

TRIP12 significantly increases global H3K9me3 abundance, and that this increase is rescued by co-depletion of SUV39H1 (new Figure EV7B). These new findings strengthen the notion that deregulation of K29-Ub- and TRIP12-dependent SUV39H1 turnover is causative of increased H3K9me3 levels on chromatin in Ub(K29R)-replaced cells.

8. The analysis of the involvement of TRIP12 and SUV39H1 in cancer is very weak. As the authors point out the levels of mRNA are not very useful to address if the control of SUV39H1 stability by TRIP12 is relevant in cancer. The effect in Figure 5E for TRIP12 is very minor. The inverse correlation of TRIP12 and SUV39H1 protein levels in LUAD offers a hint of a potential coordinated regulation. I feel this last part does not offer relevant information and a more in-depth analysis of the control of H3K9me3 by the ubiquitylation of SUV39H1 would be more relevant for the manuscript.

We followed the Reviewer's suggestion to expand the focus on the role of SUV39H1 regulation by K29-linked ubiquitylation and TRIP12 in controlling cellular H3K9me3 status, as described in the preceding points. While our pharmacogenomic analyses are consistent with a potential physiological relevance of TRIP12-dependent regulation of SUV39H1 in cancer progression and prognosis that we believe merits inclusion in the manuscript, we de-emphasized this part by moving most of these data to Expanded View figures.

Minor comment: the first paragraph in page 9 is a bit confusing since it is not easy to follow when the authors are talking about the whole proteome or the analysis of ubiquitylated proteins.

We have amended this paragraph to make it more clear when impacts on total protein abundance or ubiquitylation status are described.

Referee #3:

In this manuscript, the authors utilise a cell-based ubiquitin replacement strategy to supplant endogenous ubiquitin with exogenous mutant proteins unable to form each of the seven lysine-based ubiquitin linkages in human cells. The authors use this system to profile the cellular impacts of abrogating individual polyubiquitin chain types and use their panel of cell-lines to assess the performance and specificity of commercially available ubiquitin linkage-selective affinity reagents. The authors choose to focus on K29-linked ubiquitination, showing that it is strongly associated with chromosome biology and that the histone methyltransferase SUV39H1 is a prominent substrate of proteasomal degradation mediated by K29-linked ubiquitination. Using an siRNA screen, the authors identify TRIP12 as an E3 ubiquitin ligase responsible for K29-linkage-dependent ubiquitination and degradation of SUV39H1. These results also implicate a role for Cullin-RING ubiquitin ligase activity in initiating ubiquitination of SUV39H1 to prime its subsequent TRIP12-dependent K29-Ub modification, as well as branching TRIP12-catalysed K29-linked ubiquitination with K48-linked ubiquitin chains. Abrogation of K29-linked ubiquitination stabilises SUV39H1 and deregulates the histone H3Kme3 landscape, while dysregulation of the SUV39H1 K29-Ub writer/reader machinery is associated with poorer prognosis in multiple cancer types.

This is a well-executed and comprehensive study, identifying a K29-Ub modified substrate, a responsible ligase, and probable deubiquitylase. The creation, validation, and

characterisation of all seven Ub replacement cell lines is impressive, as is the system-wide proteomic profiling of ubiquitination changes within those cells. The authors do a good job of establishing SUV39H1 as a substrate of K29-linked ubiquitination, identifying TRIP12 as a key ligase catalysing this modification in cells and whose activity is required for SUV39H1's K29-linked ubiquitin dependent turnover. This is demonstrated using both the K29R ubiquitin replacement cells and in wild-type cells looking at the endogenous proteins. Perhaps slightly weaker are the experiments implicating TRABID as a DUB for ubiquitylated SUV39H1, but the authors do demonstrate a destabilisation of endogenous SUV39H1 following TRABID depletion in multiple cell lines.

My main concerns with the manuscript are the following:

It is unclear whether the levels of Ub in the replacement cells are the same as the levels in a "normal" cell, and whether the introduced mutations can be efficiently processed by the endogenous cellular ubiquitination and deubiquitination machinery. For instance, mutation of K33 to R blocks the ability of OTULIN to cleave linear chains containing this mutation. Hence it cannot be ruled out that the observed effects are not on account of a particular linkage type being made by the K-R substitution but due to some other effect like change in the kinetics of ubiquitination. Similarly, K27 faces into the hydrophobic core of ubiquitin and is not accessible. So the mutation K27R may affect the stability of Ub leading to altered dynamics. While it may be beyond the scope to test all of these possibilities, at a minimum these must be addressed for K29R, the focus of this manuscript and the rest stated as limitations of the approach.

We agree that this is an important consideration, which echoes Reviewer #2's point 2. In the revised manuscript, we now point out and discuss potential limitations of the Ub replacement approach (page 21-22).

Our extensive analyses of Ub(K29R)-replaced cells collectively suggest that the Ub(K29R) mutant behaves indistinguishably from Ub(WT) in general ubiquitylation/deubiquitylation dynamics, underscored by the notion that Ub(K29R) replacement has little overt impact on cell fitness (Figure 1K). However, prompted by the Reviewer's concern we carried out a range of additional experiments to carefully evaluate any impacts the K29R mutation might have on Ub stability and conjugation/deconjugation kinetics in cells. Importantly, we found that the K29R mutation has no appreciable impact on the halflife of Ub as assessed by a Cycloheximide time course analysis (new Figure EV4C). Moreover, Ub(WT) and Ub(K29R) conjugates accumulated with similar kinetics upon proteasome inhibition (new Figure EV4D), and overall rates of deubiquitylation as evidenced by the kinetics of Ub conjugate clearance upon inhibiting all ubiquitylation reactions using an Ub E1 enzyme inhibitor did not differ between Ub(WT)- and Ub(K29R)-replaced cells (new Figure EV4E). Together, these findings strongly suggest that the K29R mutation does not appreciably impact Ub processing by the cellular ubiquitylation and deubiquitylation machineries, strengthening the notion that the block to Ub-dependent turnover of SUV39H1 in Ub(K29R)-replaced cells is a direct consequence of abrogating its K29-linked ubiquitylation.

On page 10, the sentence "..... while disrupting K27-linked ubiquitylation was specifically associated with enhanced ubiquitylation of mitochondrial proteins (Figure 2J-L; Figure S3)." Looking at the 2022 publication on K27 chains, this association of K27 linkage to mitochondrial proteins was not mentioned. Could the authors please comment on the reproducibility of the observations of the previous study?

As we point out in the manuscript (page 10), the most pronounced ubiquitylation changes we observe in Ub(K27R)-replaced cells generally recapitulate the results we previously published using Ub(K27R) replacement cells (Shearer *et al.*, 2022), supporting overall reproducibility of these data. However, it should be emphasized that the proteomic profiling of ubiquitylation changes in our previous and present studies were performed using different mass spectrometers, with the current work being carried out using a state-of-the-art Orbitrap Astral instrument, which displays far superior (up to 10-fold higher) sensitivity compared to the Q Exactive mass spectrometer used in our 2022 study. Furthermore, we used an optimized data acquisition strategy (narrow-window data-independent acquisition) achieving improved proteome coverage, resolution and mass accuracy relative to the conventional data-dependent acquisition approach used in our earlier work. For low-abundant species like Ub-modified proteins these technology developments substantially influence the number of detected proteins with significantly altered ubiquitylation status in Ub(K27R)-replaced cells (\log_2 fold change >0.5 ; p-value ($-\log_{10}$) >1), with 653 such proteins being identified in the present work vs. 136 proteins in our 2022 study. In our previous study of K27-linked ubiquitylation, we did not observe significant enrichment of GO terms related to mitochondria among proteins showing enhanced ubiquitylation in Ub(K27R)-replaced cells, likely reflecting the much higher sensitivity afforded by the proteomics setup used in our current study.

The authors have tried to dissect the topology of the chains on SUV39H1 and suggest that it is first priming by CRLs, followed by K29 extension by TRIP12 and then extension/branching with K48 linkages again in a CRL-dependent manner. One possibility that the authors have not considered is that it is neddylation-dependent chain initiation (i.e. chain priming). The authors show that absolutely no Ub chains are present on SUV39H1 (and that SUV39H1 protein levels are stabilised during CHX chase in cells) when neddylation is inhibited (Figs 4G, 4H & S5F) even though UbiCRest analysis suggests that CRL-dependent K48-linked ubiquitination of SUV39H1 mostly occurs distal to K29-linked ubiquitination (Fig 4F). Histone neddylation has previously been reported, although reports of non-cullin neddylation are admittedly a contentious issue and reports often rely heavily upon exogenous overexpression of NEDD8 protein. That said, RNF168, another proteolytic substrate of TRIP12, has been suggested to be a NEDD8 ligase that neddylates histones to regulate DNA damage repair and is itself reported to be a substrate for neddylation (Li et al. 2014, J Cell Sci 127, 2238-48). The authors' data is consistent with a model where the K29 Ub chains are attached to a priming neddylation of the SUV39H1 substrate rather than a priming ubiquitin. This is easily testable by blotting SUV39H1 pulldowns for NEDD8.

This is an interesting suggestion, which we tested by immunoblotting pulldowns of stably expressed GFP-SUV39H1 with a NEDD8 antibody, thereby avoiding NEDD8 overexpression that has been reported to drive spurious, non-physiological NEDDylation reactions (Hjerpe *et al.*, 2012). While we observed a single band in GFP-SUV39H1 IPs that was reactive with the NEDD8 antibody, this band was not mobility-shifted relative to unmodified GFP-SUV39H1 and was not affected by a NEDDylation inhibitor (NEDDi) that decreased SUV39H1 ubiquitylation and overall NEDDylation as expected (Reviewer Figure 1), and thus in all likelihood reflects cross-reactivity of the NEDD8 antibody with GFP-SUV39H1. Based on this and other experiments we performed we have been unable to obtain supportive evidence for direct NEDDylation of SUV39H1. Consistently, SUV39H1 was not among the NEDD8 target proteins identified in a proteome-wide study of NEDDylation sites in human cells (Lobato-Gil *et al.*, 2021).

Reviewer Figure 1.

U2OS cells and a derivative cell line stably expressing GFP-SUV39H1 treated or not with the NEDDylation inhibitor MLN-4924 (NEDDi) for 5 h were subjected to GFP pulldown, followed by immunoblotting with indicated antibodies.

Minor points:

1. Figure 3A contains two GAPDH panels, one of which has an asterisk next to it. The NDC80 blot also has an asterisk next to its label. I assume that, like the asterisked text in Figure 4C, these samples were run on a separate gel. However, the figure legend does not mention this and the labels are not explained. The authors should update the legend to explain the significance of the asterisks.

The Reviewer is correct, and we have updated the figure legend accordingly to explain the significance of the asterisks.

2. In Figure S1A, it is unclear if the levels of replaced ubiquitin are the same as parental cells. Please show comparison with parental U2OS cells. Also incubation of cell lysates with a promiscuous DUB like USP2 will reveal if the ubiquitin levels are the same.

To enable comparison of the levels of Ub in parental U2OS cells and the panel of Ub replacement cell lines, we added a blot showing that the level of endogenous Ub is indistinguishable between parental U2OS cells and the derivative U2OS/shUb cell line used to generate the Ub replacement cell lines (new Figure EV1B), extending the data in Figure EV1A. We also included new mass spectrometry data based on diGly antibody enrichment directly quantifying Ub linkage intensity and the extent of their disruption upon Ub replacement across our panel of Ub replacement cell lines, validating near-complete

reduction of the linkage type directly impacted by the K-to-R mutation in these Ub replacement cell lines (new Figure 1H).

3. RNF168 is mentioned but not labelled in the plot shown in Figure 2E.

Although we provide evidence from immunoblotting and IP experiments to show that the stability and ubiquitylation of RNF168 is clearly increased upon abrogation of K29-linked ubiquitylation (Figure 3A; Figure EV4J), these effects were somewhat variable between individual replicates of our mass spectrometry-based analysis and thus did not reach statistical significance, possibly due to RNF168 being a very low copy number protein. To avoid confusion, we amended the text describing RNF168 as one of the candidate targets of K29-Ub-regulated ubiquitylation we assessed in this study based on it being a known TRIP12 substrate.

4. In Figure S4A, U2OS/shUb, U2OS/shUb WT and U2OS/shUb K29R at 24 h show different ubiquitylation levels. Furthermore, clone C93 seem to have a higher ubiquitylation signal than WT and clone C85. It is not clear if the main conclusions of the study were drawn from analyses of one or multiple clones. Please specify.

All key results from Ub(K29R) replacement cells were validated in each of the two independent Ub(K29R) clones (c85 and c93) we generated (Figure 1I; Figure EV4F,G), and which behaved indistinguishably in all functional studies we carried out. For experiments involving only one of these clones the c93 clone was used. We now point this out in the Methods section (page 28).

References

- Hjerpe R, Thomas Y, Kurz T (2012) NEDD8 overexpression results in neddylation of ubiquitin substrates by the ubiquitin pathway. *J Mol Biol* 421: 27-29
- Lobato-Gil S, Heidelberger JB, Maghames C, Bailly A, Brunello L, Rodriguez MS, Beli P, Xirodimas DP (2021) Proteome-wide identification of NEDD8 modification sites reveals distinct proteomes for canonical and atypical NEDDylation. *Cell reports* 34: 108635
- Shearer RF, Typas D, Coscia F, Schovsbo S, Kruse T, Mund A, Mailand N (2022) K27-linked ubiquitylation promotes p97 substrate processing and is essential for cell proliferation. *EMBO J* 41: e110145
- Yu Y, Zheng Q, Erramilli SK, Pan M, Park S, Xie Y, Li J, Fei J, Kossiakoff AA, Liu L, Zhao M (2021) K29-linked ubiquitin signaling regulates proteotoxic stress response and cell cycle. *Nat Chem Biol* 17: 896-905

Prof. Niels Mailand
NNF Center for Protein Research, University of Copenhagen
Ubiquitin Signaling Group, Protein Signaling Program
Blegdamsvej 3B
Copenhagen DK-2200
Denmark

16th Sep 2025

Re: EMBOJ-2024-119909R
Functional landscape of ubiquitin linkages couples K29-linked ubiquitylation to epigenome integrity

Dear Niels,

Thank you for submitting your revised manuscript to The EMBO Journal, and apologies for its delayed re-assessment. Two of the original referees have now looked into it once more, and were fully satisfied with the revisions. After incorporation of their few remaining minor points (see below), we shall therefore be happy to accept the manuscript for publication. In addition, please also address the following remaining editorial issues at this stage:

- Please remove the EV Dataset legends from the main manuscript text, they are already included in Excel files correctly.
- In the Data Availability section, please remove the referee access information at this stage, and ensure that the datasets are now becoming openly available.
- Regarding your well-taken Sustainability statement: we don't really have a separate section for this, so I would suggest to include it as part of the Acknowledgement section.
- Please rename the Conflict of Interest section into "Disclosure and Competing Interests Statement", in accordance with our updated Guide to Authors (<https://www.embopress.org/competing-interests>)
- As we are switching from a free-text author contribution statement towards a more formal statement based on Contributor Role Taxonomy (CRediT) terms, please remove the present Author Contribution section and instead specify each author's contribution(s) directly in the Author Information page of our submission system during upload of the final manuscript. See <https://casrai.org/credit/> for more information.
- Finally, please provide suggestions for a short 'blurb' text prefacing and summing up the study in two sentences (max. 250 characters), followed by 3-5 one-sentence 'bullet points' with brief factual statements of key results of the paper; they will form the basis of an editor-written 'Synopsis' accompanying the online version of the article. Please also upload a synopsis image, which can be used as a "visual title" for the synopsis section of your paper. The image should be in PNG or JPG format with the modest dimensions of EXACTLY 550 pixels wide and between 300 and 600 pixels high.

I am returning the manuscript to you for a final round of minor revision, hoping you can quickly make these modifications and upload the revised files. Once we will have received them, we should be ready to swiftly proceed with formal acceptance and production of the manuscript.

With kind regards,

Hartmut

1) Every manuscript requires a Data Availability section (even if only stating that no deposited datasets are included). Primary datasets or computer code produced in the current study have to be deposited in appropriate public repositories prior to

resubmission, and reviewer access details provided in case that public access is not yet allowed. Further information: embopress.org/page/journal/14602075/authorguide#dataavailability

9) To facilitate reproducibility and cross-laboratory adoption of methodologies, please structure the Materials & Methods section as outlined in our guide to authors, including a completed Reagents and Tools Table that can be downloaded from our author guidelines as well (<https://www.embopress.org/page/journal/14602075/authorguide#structuredmethods>).

10) Digital image enhancement is acceptable practice, as long as it accurately represents the original data and conforms to community standards. If a figure has been subjected to significant electronic manipulation, this must be clearly noted in the figure legend and/or the 'Materials and Methods' section. The editors reserve the right to request original versions of figures and the original images that were used to assemble the figure. Finally, we generally encourage uploading of numerical as well as gel/blot image source data; for details see: embopress.org/page/journal/14602075/authorguide#sourcedata

In the interest of ensuring the conceptual advance provided by the work, we recommend submitting a revision within 3 months (15th Dec 2025). Please discuss the revision progress ahead of this time with the editor if you require more time to complete the revisions. Use the link below to submit your revision:

Link Not Available

Referee #2:

In the revised version of their manuscript, Arroyo-Gómez have satisfactorily addressed all my questions and comments. The new data further strengthen their conclusions. This is a high-quality manuscript relevant for a broad audience.

I only have one question related to Figure EV7B where the Y axis is labeled as "Occupied (%)". What do the authors refer to?

Referee #3:

The authors have addressed all the concerns raised and the revised manuscript is significantly improved. I am happy for the manuscript to be accepted for publication.

Minor point: In figure 3H, can the authors annotate the E3 ligase hits whose knockdown results in an increase in the amount of K29 linkages